# 🤖 *DRBench*: A Realistic Benchmark for Enterprise Deep Research

**Amirhossein Abaskohi**[1,2]     **Tianyi Chen**[1]     **Miguel Muñoz-Mármol**[1]     **Curtis Fox**[1,2]
**Amrutha Varshini Ramesh**[1,2]     **Étienne Marcotte**[1]     **Xing Han Lù**[3,4]     **Nicolas Chapados**[1]
**Spandana Gella**[1,3,4]     **Christopher Pal**[1,3,5,6]     **Alexandre Drouin**[1,3]     **Issam H. Laradji**[1,2]

[1]ServiceNow Research     [2]University of British Columbia     [3]Mila – Quebec AI Institute
[4]McGill University     [5]Polytechnique Montreal     [6]Canada CIFAR AI Chair

## Abstract

We introduce *DRBench*, a benchmark for evaluating AI agents on complex, open-ended deep research tasks in enterprise settings. Unlike prior benchmarks that focus on simple questions or web-only queries, *DRBench* evaluates agents on multi-step queries (for example, "What changes should we make to our product roadmap to ensure compliance with this standard?") that require identifying supporting facts from both the public web and private company knowledge base. Each task is grounded in realistic user personas and enterprise context, spanning a heterogeneous search space that includes productivity software, cloud file systems, emails, chat conversations, and the open web. Tasks are generated through a carefully designed synthesis pipeline with human-in-the-loop verification, and agents are evaluated on their ability to recall relevant insights, maintain factual accuracy, and produce coherent, well-structured reports. We release 100 deep research tasks across 10 domains, such as Sales, Cybersecurity, and Compliance. We demonstrate the effectiveness of *DRBench* by evaluating diverse DR agents across open- and closed-source models (such as GPT, Llama, and Qwen) and DR strategies, highlighting their strengths, weaknesses, and the critical path for advancing enterprise deep research. Code and data are available at https://github.com/ServiceNow/drbench.

## 1 Introduction

Organizations today face a strong need to find useful insights in a world full of overwhelming information. Valuable insights are often hidden in noisy data, which can contain many distracting or irrelevant details that obscure the insights that really matter. This challenge is present in enterprise settings, where data is spread across many applications and stored in different formats (e.g., PDFs, spreadsheets, emails, and internal tools) making extracting relevant information difficult. To uncover these hidden, valuable insights, one must conduct what is known as **deep research**. This task involves asking high-level strategic questions (e.g, "What changes should we make to our roadmap to remain compliant?"), planning sub-questions, retrieving and evaluating relevant materials, and producing a clear, actionable summary grounded in data sources (Zheng et al., 2025; Xu & Peng, 2025; Du et al., 2025). These tasks are typically performed by domain experts using a mix of search engines, communication platforms, and business applications in iterative, high-effort workflows (Mialon et al., 2024), which unfortunately require a significant amount of human effort.

One promising solution to reducing this human effort is agent-based deep research, which uses autonomous software agents to search, extract, and synthesize information across fragmented sources into an insightful report. Recently, LLM-based agents have emerged as promising assistants for deep research. Systems such as *Local Deep Researcher* (LearningCircuit, 2025), *Deep-Searcher* (Tech, 2024), and *DeepResearcher* (Zheng et al., 2025) propose modular agent pipelines that combine retrieval, reasoning, and summarization over documents and web sources. Architectures like OpenHands (All-HandsAI, 2024), OpenManus (FoundationAgents, 2024), and smolagents (HuggingFace, 2024) extend these capabilities to include collaboration, multi-modal search, and complex tool use in enterprise workflows (Xu & Peng, 2025). Despite these advances, evaluating such systems remains an open challenge.

Most existing benchmarks evaluate narrow aspects such as report factuality (Coelho et al., 2025), web-only synthesis (Bosse et al., 2025), or tabular analytics (Sahu et al., 2025), but they do not assess whether agents identify the most salient insights, remain faithful to retrieved evidence, or adapt to enterprise contexts.

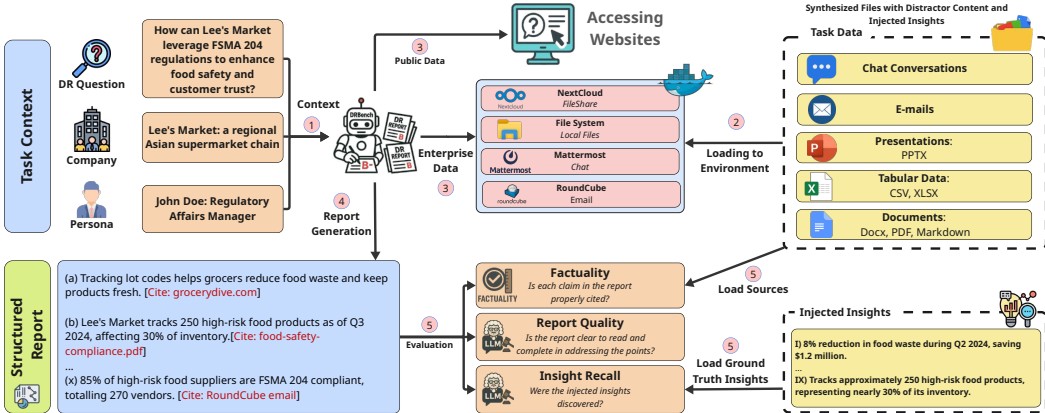

Figure 1: **DRBench** pipeline. ① The *Task Context* defines the deep research question grounded by the company and persona given to the agent. ② *Task Data*, including both distractor and injected groundtruth insights in different formats (PDFs, DOCX, PPTX, XLSX, chats, etc.) are loaded into the enterprise environment's applications. ③ The *DRBenchAgent* accesses both public web sources and local enterprise data to extract relevant insights for the research question. ④ It produces a structured research report, which is ⑤ evaluated for *Insight Recall* (detecting injected groundtruth insights), *Factuality* (verifying claims are correctly cited), and *Report Quality*.

To address these limitations, we introduce *DRBench*, a benchmark designed to evaluate LLM agents on open-ended, multi-step and long-horizon deep research tasks grounded in realistic enterprise contexts. As Figure 1 illustrates, *DRBench* includes a suite of queries grounded in user personas and organizational scenarios, requiring agents to search across real applications such as cloud file storage (Nextcloud), enterprise chat (Mattermost), and user file systems, and to reason over formats like spreadsheets, slide decks, and PDFs. Our evaluation framework introduces three scoring axes using LLM-as-a-judge methods inspired by G-Eval (Liu et al., 2023): (1) *Insight Recall and Distractor Avoidance*, which together evaluate whether the agent surfaces the most salient injected insights while avoiding distractor content; (2) *Factuality*, which uses a TREC-RAG pipeline (Wang et al., 2024) to verify whether claims are correctly grounded in their cited sources; and (3) *Report Quality*, which measures the coherence, completeness, and overall readability of the synthesized report. We conduct a comparative study of agent architectures inspired by recent work (Zheng et al., 2025; Xu & Peng, 2025; LearningCircuit, 2025; Zheng et al., 2025), analyzing how well they perform on *DRBench* across planning, insight identification, and grounding on facts. Our results show that while agents are competent at document retrieval and summarization, they often miss high-value insights, cite irrelevant evidence, or produce incoherent explanations, highlighting the limitations of current architectures and the need for more targeted innovation.

**Our contributions are as follows: (1)** We introduce *DRBench*, the first benchmark for evaluating LLM agents on complex enterprise deep research tasks combining public web sources with private organizational data; **(2)** We provide a suite of 100 high-level research tasks with 1093 sub-questions spanning 10 domains, including Sales, Cybersecurity, and Compliance, each grounded in realistic company contexts and personas; **(3)** We design a reproducible enterprise environment integrating realistic enterprise applications like chat, cloud storage, emails, and documents; **(4)** We propose a scalable pipeline that generates realistic research questions and insights by combining web facts with synthesized internal data; and **(5)** We develop an evaluation framework that scores agent reports on insight recall and distractor avoidance, factuality, and overall report quality.

## 2 RELATED WORK

**Deep Research Benchmarks.** With the growing capabilities of LLMs in research and reasoning tasks, several benchmarks have emerged, including Deep Research Bench (Bosse et al., 2025), DeepResearch Bench (Du et al., 2025), DeepResearchGym (Coelho et al., 2025), ResearcherBench (Xu et al., 2025b), Mind2Web2 (Gou et al., 2025), and GAIA (Mialon et al., 2024). As summarized in Table 1, these efforts primarily evaluate web-only retrieval or synthesis in controlled settings. A recent benchmark by Choubey et al. (2025) further emphasizes closed-form fact retrieval within engineering artifacts. In contrast,

Table 1: Comparison of deep research benchmarks (top) and AI agent benchmarks with a computer environment (middle). Columns report dataset size, whether both public and local data are required, the provided environment type, task domains, task description, and evaluation method. Unlike prior work, *DRBench* combines public web retrieval with local enterprise data in realistic enterprise applications and evaluates both insight recall, distractor avoidance and report quality. **Task Description**: types of tasks covered by the benchmark: **WR** for Web Research, **DR** for Deep Research with both public and local data, **CU** for Computer Use and/or Mobile Use. *DRBench* has 1093 total **# groundtruth** insights that need to be extracted to address the 100 DR Questions. Example groundtruth insights can be found at Table 8 in Appendix A.

| Benchmark | # groundtruth | Public & Local Data | Provides Env | Task Domain | Task Description | Main Evaluation Method |
|---|---|---|---|---|---|---|
| Deep Research Bench (Bosse et al., 2025) | 89 | ✗ | ✓ | Generic | WR & CU | Answer Accuracy |
| DeepResearch Bench (Du et al., 2025) | 100 | ✗ | ✗ | Generic | WR | Insight Recall |
| DeepResearchGym (Coelho et al., 2025) | 1,000 | ✗ | ✗ | Generic | WR | Document Retrieval |
| ResearcherBench (Xu et al., 2025b) | 65 | ✗ | ✗ | AI | WR | Insight Recall, Factuality |
| LiveDRBench (Java et al., 2025) | 100 | ✗ | ✗ | Generic | WR & CU | Insight Precision, Recall |
| BrowseComp-Plus (Chen et al., 2025) | 1,005 | ✗ | ✗ | Generic | WR | Answer Accuracy, URL Recall |
| Mind2Web 2 (Gou et al., 2025) | 130 | ✗ | ✗ | Generic | WR | Partial Completion |
| GAIA (Mialon et al., 2024) | 466 | ✗ | ✗ | Generic | WR | Answer Accuracy |
| GAIA2 (Andrews et al., 2025) | 963 | ✗ | ✓ | Generic | CU | Action Accuracy |
| TheAgentCompany (Xu et al., 2025a) | 175 | ✗ | ✓ | Enterprise | CU | Task Completion, Efficiency |
| OSWorld (Xie et al., 2024) | 369 | ✗ | ✓ | Generic | CU | Task Completion |
| ***DRBench*** | 1093 (100 tasks) | ✓ | ✓ | Enterprise | DR | Insight Recall |

*DRBench* is the first to combine web retrieval with local enterprise data, requiring multi-step deep research grounded in persona- and domain-specific contexts.

**Enterprise Environments.** Realistic enterprise environments have become an important testbed for evaluating agents in complex multi-application workflows. *CRMArena-Pro* (Huang et al., 2025a;b) targets sales and CPQ pipelines through persona-grounded dialogues, but is limited to conversational sales workflows. *OSWorld* (Xie et al., 2024) and *OSWorld-Gold* (Abhyankar et al., 2025) benchmark agents in general-purpose desktop environments, using applications such as Microsoft Word and Excel, yet their focus remains on computer task execution rather than enterprise deep research. *TheAgentCompany* (Xu et al., 2025a) evaluates collaboration among autonomous agents for programming, browsing, and communication, though the tasks are computer-use focused and do not assess deep research capabilities. *WorkArena* (Drouin et al., 2024; Boisvert et al., 2024) offers a realistic enterprise environment with knowledge work tasks for web agents, though it does not support evaluation of deep research capabilities. In contrast, *DRBench* offers a domain-grounded enterprise environment with applications that would realistically be encountered in organizations. Tasks are tied to concrete personas and roles, requiring agents to search, reason, and synthesize insights across diverse formats, including spreadsheets, PDFs, wikis, emails, and presentations, reflecting realistic enterprise deep research.

**Deep Research Agents.** A growing line of work explores agents for multi-step search and synthesis across diverse information sources. LangChain's *Local Deep Researcher* (LearningCircuit, 2025) and Zilliz's *Deep-Searcher* provide modular pipelines for iterative querying and summarization, while *DeepResearcher* (Zheng et al., 2025) uses RL to enable planning, cross-validation, and self-reflection. Commercial systems such as *Gemini Deep Research* and *Manus.ai* synthesize web-based reports with citations, and open-source frameworks like OpenHands (All-HandsAI, 2024), OpenManus (FoundationAgents, 2024), and smolagents (HuggingFace, 2024) offer alternative architectures. Recent work also introduces task-agnostic frameworks for long-form synthesis and evaluation paradigms such as Mind2Web 2 (Gou et al., 2025), which treat agents as judges of browsing trajectories. Building on these efforts, *DRBench* analyzes their strengths and limitations in enterprise contexts, showing that current agents still fall short in consistently extracting and grounding critical insights within complex, heterogeneous environments.

## 3 *DRBench* - AN ENTERPRISE DEEP RESEARCH BENCHMARK

To evaluate agents on complex, open-ended enterprise deep research tasks, we designed *DRBench* with three guiding principles: it requires agents to integrate both public web data and local enterprise documents, it involves both web search and enterprise application use, and it is hosted in an interactive and reproducible

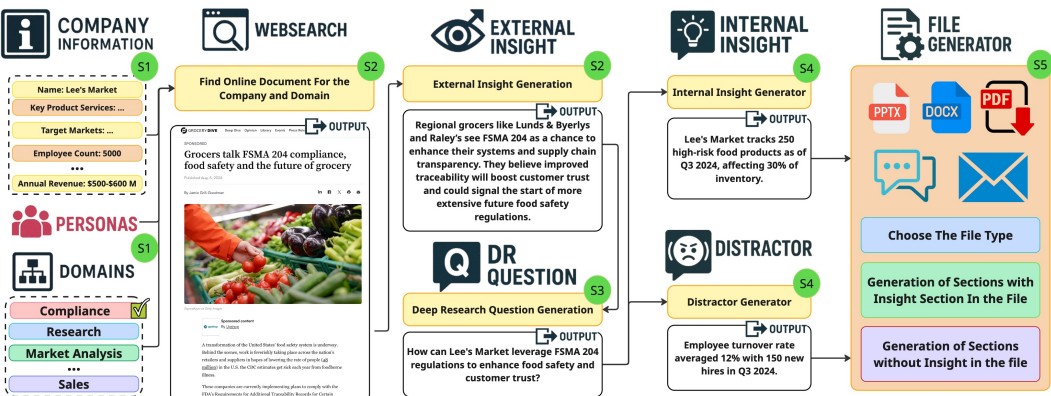

Figure 2: *DRBench* Task Generation Pipeline. The pipeline comprises five main stages during each LLMs generate candidate data such as company context, insights, and research questions, while human annotators verify quality and select the final version. Stages S1–S5 denote the five generation steps.

enterprise environment. These principles ensure that the benchmark reflects realistic enterprise workflows and provides a controlled yet challenging setting for research agents.

**Benchmark Scope.** *DRBench* evaluates document-centric deep research in enterprise settings, where agents must synthesize evidence across heterogeneous applications and sources, including both public web content and private enterprise documents. Tasks are cross-application by design and reflect realistic research workflows such as policy analysis, compliance assessment, market research, and strategic decision support that rely on reading, retrieving, and reasoning over unstructured enterprise data.

**The Enterprise Search Environment.** A unique aspect of *DRBench* is its realistic enterprise search environment. When addressing DR Questions like "What changes should we make to our product roadmap to ensure compliance with this standard?", the DR agents would need to navigate such environment and search across both public and private data sources to uncover relevant insights.

Public insights include information available on the open web or otherwise accessible to general users. Local insights, on the other hand, come from an enterprise's private systems. These insights are embedded within a vast search space that spans multiple data types (emails, slide decks, chat conversations, and Excel sheets) which reflect the complexity of real enterprise data ecosystems. This environment is populated with data from different applications, accessible to both web-based agents and API-calling agents. For example, an app like Mattermost can be used to host chat conversations (see Appendix E for examples of the applications). The goal of the DR Agent is to effectively navigate these public and private data sources to address complex, high-level DR questions. For the environment implementation details, please see Appendix D.

**Task Definition.** Each task is associated with a deep research question $Q_i$ and Task Context $C$ which includes company information and the user persona. Each task also has a corresponding set of groundtruth insights $I$ consisting of relevant private insights $I_l$ (we also refer to this as internal insights), distractor private insights $I_d$, and public insights $I_p$. Each private insight, whether relevant or a distractor, is embedded into a file $f_i$ which could take the form of a PDF, Excel sheet, slide deck, chat log, and so on. The agent's task is to generate a report by extracting the public insights $I_p$ from accessible sources such as the web, while also extracting the private insights $I_l$ from the files hosted in the enterprise environment. At the same time, the agent must avoid extracting the distractor insights $I_d$, which are not relevant to the DR question.

*DRBench* provides 100 realistic deep research tasks explicitly framed around enterprise environments. Each task is associated with public insights extracted form quality, time-invariant URLs and local insights embedded within synthetic enterprise data, typically spanning 2–4 applications and 3–16 supporting files (see Appendix B). Tasks are distributed across 10 enterprise domains (such as Sales and Compliance - the full list is in Appendix B) and divided between easy, medium, and hard categories that indicates the difficulty of addressing the DR Question. Finally, *DRBench* is fully self-hosted, with dated URLs and reproducible evaluation scripts to ensure stability and fair comparison across agent methods.

## 3.1 DATA GENERATION

To create realistic and reproducible deep research tasks, *DRBench* employs a five-stage pipeline (Figure 2) that combines large-scale LLM generation with human-in-the-loop verification. The pipeline helps us

generate candidate company contexts, personas, questions, insights, and supporting files using LLM Models such as Llama-3.1-8B-Instruct, Llama-3.1-405B (Dubey et al., 2024). Three human annotators then validate the generated content to ensure that they are realistic and plausible.

The pipeline has been used to generate 100 tasks with 1093 groundtruth insights across 10 enterprise domains, each grounded in realistic personas and company profiles. We control the difficulty of each task by setting the number of insights, file types and application types. The complete list of tasks is provided in Appendix B. Refer to Appendix I for details on the cost of using data generation.

**Stage 1: Company and Persona Generation.** This stage produces the synthetic company profile and user persona that form the Task Context $\mathcal{C}$. LLMs were used to generate company descriptions detailing the industry vertical, key products, market position, and competitive landscape. In parallel, they were used to create realistic personas across departments (e.g., a Regulatory Affairs Manager or a Market Research Analyst) that serve as the role grounding for the final deep research question. They were then refined by human experts. The prompts used for this stage are provided in Appendix P.1 and the list of companies are given in Appendix B.

**Stage 2: Public Source and Insight Collection.** Given the company and persona context from Stage 1, we have retrieved candidate URLs relevant to the specified domain and company background. To ensure quality, time-invariant insights, the search is restricted to dated, journal-based or industry-report websites that provide authoritative information. Thus, the collected URLs and their contents are expected to be stable in time. Human annotators then review the candidate URLs and select one that is both topically aligned and provides insights into the topic. The selected page becomes the *Task URL* included in $\mathcal{C}$. Its HTML content is parsed, and LLMs are prompted to extract business-relevant insights, which are subsequently filtered and validated by human reviewers for accuracy and contextual fit. The public insights $I_p$ derived from the Task URL are included in $\mathcal{C}$ and serves as a required piece of insight for the agent to retrieve during report generation. Prompts used for this stage and the list of urls are provided in Appendix P.3.

**Stage 3: Question Generation.** Given the Task Context, we generate the deep research question $Q$. The prompt (see Appendix P.2) is instantiated with the company profile and persona, the selected domain, the Task URL, and the public insight $I_p$. The LLM proposes several open-ended candidate questions grounded in this context. Human annotators then review these candidate DR Questions, selecting and refining one to align with the persona and company. They also ensure that the insights available in the provided URL can at least partially support answering the deep research question. For example, if the question concerns compliance with a specific regulation, the URL might include relevant insights, such as "groceries must have a traceability plan." While this doesn't fully resolve the question, it provides a foundation.The question should be high-level enough to allow us to synthesize additional supporting private/internal insights $I_l$ (such an insight could be "the cost of implementing such a plan is $X$ amount") which are needed to strengthen the report generated for the question. This requirement ensures that new internal insights can be generated, as discussed in Stage 4.

**Stage 4: Internal Insight Generation.** In this stage we generate the injected insights set $\mathcal{G} \subset \mathcal{I}$. Using the public insight $I_p$ and the deep research question $Q$, LLMs are used to create company-specific insights aligned with the organization's industry, priorities, customer segments, and business goals. These insights are designed to provide additional supporting facts that need to be extracted to create a report that better addresses the DR questions. Human annotators review and refine these insights for accuracy and alignment with the questions. In addition to relevant insights, we also produce distractor insights $I_d$, which are plausible but irrelevant statements that do not support resolving the DR Question. Prompt details are provided in Appendix P.4 and example internal insights are provided in Appendix B.

**Stage 5: File Mapping and Generation.** This stage produces the the set of files $\{f_i\}$ containing both the relevant and distractor private insights. First, each insight is assigned to a modality such as email, chat, pdf, docx, and so on. Then the file generation module follows the following three-step "needle-in-a-haystack" process: (1) create an outline of the file based on its modality(e.g., document structure or chat configuration), (2) insert the distractor or relevant insight into an appropriate section of the file, and (3) fill the remaining content with realistic but irrelevant information. Human annotators spot-check the generated files to ensure fidelity, coherence and no contradicting information. Prompts for file generation are provided in Appendix P.5 and screenshots of such generated files are in Appendix C.

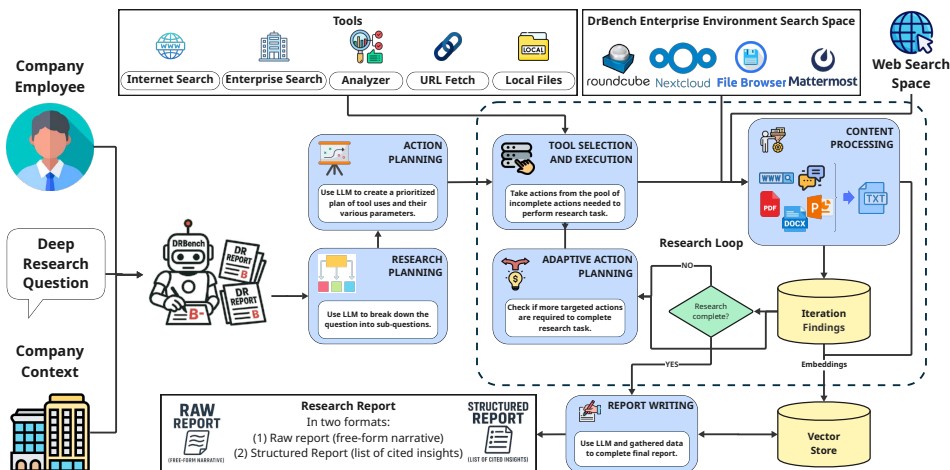

Figure 3: *DRBench* Agent architecture showing the enterprise research workflow from question submission through iterative research cycles to final report generation, using both enterprise and web search capabilities. Reports are generated in two formats: a raw report, consisting of free-form narrative text, and a structured report that lists the main insights with their corresponding citation(s).

## 4 *DRBench* AGENT

The *DRBench* baseline agent (DRBA) is the first agent built specifically for deep research in enterprise settings, designed to operate directly within the *DRBench* environment. Its multi-stage architecture systematically investigates research questions by iteratively retrieving, processing, and synthesizing knowledge from enterprise services and the web until completion or a maximum iteration limit is reached (Figure 3). The agent has access to app-specific API calling tools to access diverse information sources and a diverse toolset for analyzing retrieved results (Table 9, Appendix F.3). DRBA's architecture is organized into four main components: research planning, action planning, an adaptive research loop, and report writing. See Appendix see Appendix F for more details on the implementation details. Refer to Appendix I for details on the cost of using DRBA.

**Research Planning.** The agent decomposes research questions into structured research investigation areas to guide subsequent action generation. This initial decomposition lays out a strategy to systematically cover the research space while maintaining focus on the initial deep research question. The agent supports two research planning modes: (1) **Complex Research Planning (CRP)**, which generates a structured research plan with detailed investigation areas, expected information sources, and success criteria; and (2) **Simple Research Planning (SRP)**, which produces a lightweight decomposition of the main question into a small set of self-contained subqueries. See Appendix F.2 for detailed examples of both modes.

**Research Loop with Adaptive Action Planning (AAP).** The system iterates through (1) tool selection and execution based on action priorities, (2) content processing and storage in a vector store, (3) adaptive action generation to cover research gaps, and (4) iteration findings storage for traceability and assessment.

**Report Writing.** The report writing subsystem queries the vector store to synthesize the research findings and relevant retrieved content. This component generates a comprehensive report and uses its own citation tracking system to ensure proper attribution of claims. For our evaluation, we use the structured report format, a list of insights with their citations, rather than a free-form narrative response (raw report).

## 5 EXPERIMENTS AND RESULTS

In this section, we evaluate the DRBench Agent (DRBA) on both the full *DRBench* benchmark and a reduced subset for ablations. We consider (1) the **FullBenchmark**, covering all 100 tasks across 10 domains (see Table 7 for details of the tasks), and (2) the **MinEval** subset, restricted to 15 tasks for efficient ablation studies. Results are reported across four metrics: Insight Recall, Distractor Avoidance, Factuality, and Report Quality, explained in the next section. Implementation details and hyperparameters are in Appendix H, while the impact of the number of research loop iterations on the performance of DRBA is analyzed in Appendix O.

## 5.1 EVALUATION METRICS

Our evaluation intentionally operates at the level of *atomic insights* rather than treating the report as a single monolithic output. This design is diagnostic rather than purely outcome-oriented, as it enables partial credit when only a subset of key findings is recovered, allows localization of specific failure modes such as missing, unsupported, or irrelevant insights, and supports multi-axis analysis across recall, precision, and factual grounding. Compared to end-to-end correctness metrics, insight-level evaluation reveals how and where an agent succeeds or fails, which is especially important for long, compositional research reports.

**Insight Recall.** Our benchmark directly evaluates the set of atomic insights provided by the agent, *which was the case for all our experimental results*. If an agent provides the full report, it decomposes the report into atomic insights using an LLM (see Prompt 14). Each insight is then compared against the groundtruth set using an LLM Judge with Prompt 15. If a match is found, the insight is marked as *detected* and contributes to the insight recall score; otherwise, it is ignored. This metric thus measures recall rather than precision, since judging whether an unmatched insight is nonetheless *useful* for answering the deep research question is inherently subjective and difficult to automate. In practice, the computed insight recall functions as an **accuracy measure** in our setting, since it reflects the proportion of groundtruth insights tied to the research question that the agent successfully identifies. To prevent agents from trivially achieving 100% recall by copying all content into the generated report, the LLM Judge evaluates only the first $k$ insights, where $k$ equals the number of ground-truth insights plus five. This buffer ensures that reports are not penalized for including seemingly relevant insights that are not part of the groundtruth insight set. While this cutoff may seem arbitrary, we found it essential for preventing agents from gaming the metric by copying large portions of the source files. The +5 buffer allows space for a few reasonable additional insights without rewarding unrestricted copying. We acknowledge that this design limits our ability to measure the full breadth of useful discoveries, and we discuss its implications and alternatives in Appendix T. Our evaluation relies on short, atomic claims, which keeps LLM-judge variance minimal regardless of whether the judge is a closed or open model; refer to Table 20 in Appendix L for more details.

**Distractor Avoidance.** To measure precision, we track whether the agent's report includes distractor insights that are irrelevant to the research question. We compute *distractor recall* analogously to insight recall, and define *distractor avoidance* as $1-$ distractor recall.

**Factuality.** Using the same set of insights (that we used for Insight Recall), we follow the methodology of FactScore (Min et al., 2023). If an insight lacks a citation or references a non-existent source, it is labeled unfactual. Otherwise, we apply a retrieval-augmented system based on `text-embedding-3-large` (OpenAI, 2024) to fetch the top-5 most relevant chunks from the cited document (Appendix H). The LLM Judge with Prompt 16 then determines whether the cited evidence supports the claim. We also store justifications and model confidence scores for interpretability.

**Report Quality.** Inspired by prior work (Coelho et al., 2025; Abaskohi et al., 2025), we query the LLM Judge with Prompt 17 to assign a 1–10 rating across six dimensions: (1) depth and quality of analysis, (2) relevance to the research question, (3) persona consistency, (4) coherence and conciseness, (5) absence of contradictions, and (6) completeness and coverage. The final report quality score is obtained by averaging these six ratings.

## 5.2 MAIN RESULTS

We first evaluate our DRBA agent using GPT-4o as the backbone model, a maximum of 15 research loop iterations, and different combinations of planning modules: Simple Research Planning (SRP), Complex Research Planning (CRP), and Adaptive Action Planning (AAP) on the full benchmark. The results are reported in Table 2. Overall, the agent demonstrates moderate ability to ground its answers in factual evidence but struggles to consistently surface the main injected insights necessary for answering the deep research questions. In many cases, the agent relies on prior knowledge or external web content rather than integrating the crucial enterprise-specific information available in the files. By contrast, it is consistently strong in avoiding distractors, showing that the agent is robust against misleading or irrelevant information but less effective at prioritizing decision-critical insights. Note that our LLM Judge backbone is GPT-4o. It should be also mentioned that Insight Recall is robust to paraphrasing, with negligible variance across reformulations (see Appendix S).

As the results illustrates, SRP tends to produce more factually grounded answers, while CRP excels at filtering out distractors through structured decomposition. AAP, on the other hand, provides the largest

Table 2: DRBA performance with different planning configurations on *DRBench*(FullBenchmark). We compare the base agent with variants using Simple Research Planning (SRP), Complex Research Planning (CRP), Adaptive Action Planning (AAP), and their combinations. See Appendix K for the standard error across 3 runs on MinEval. Note that higher numbers correspond to better scores, and the best result on each metric is bolded.

| Configuration | Insight Recall | Factuality | Distractor Avoidance | Report Quality | Harmonic Mean |
|---|---|---|---|---|---|
| Base DRBA | 13.18 | 58.04 | 95.76 | 88.23 | 34.82 |
| + SRP | 13.42 | **62.11** | 96.62 | 89.74 | 35.68 |
| + CRP | 13.31 | 59.53 | **97.14** | 87.92 | 35.21 |
| + AAP | **15.97** | 60.37 | 96.48 | **90.08** | **39.74** |
| + SRP + AAP | 14.83 | 55.29 | 96.55 | 88.96 | 37.34 |
| + CRP + AAP | 14.19 | 52.08 | 96.47 | 87.54 | 35.89 |

Table 3: Performance of DRBA on the FullBenchmark subset using different backbone language models and planning strategies. Note that higher numbers correspond to better scores, and the best result on each metric is bolded. The full table with more models is given in Appendix M.

| DRBA Backbone Model | Planning | Insight Recall | Factuality | Distractor Avoidance | Report Quality | Harmonic Mean |
|---|---|---|---|---|---|---|
| GPT-5 | - | 36.52 | 72.11 | 93.22 | 93.41 | **63.81** |
| GPT-5 | SRP | 35.41 | 69.42 | 94.67 | **93.88** | 62.64 |
| GPT-5 | CRP | **37.48** | 62.33 | 91.71 | 92.03 | 62.02 |
| Llama-3.1-405B-Instruct | - | 16.1 | 69.3 | **95.2** | 88.6 | 40.68 |
| Llama-3.1-405B-Instruct | SRP | 15.7 | **70.4** | 94.6 | 89.7 | 40.15 |
| Llama-3.1-405B-Instruct | CRP | 18.33 | 65.72 | 95.04 | 89.01 | 43.70 |
| DeepSeek-V3.1 | - | 22.6 | 68.4 | 94.9 | 84.1 | 49.20 |
| DeepSeek-V3.1 | SRP | 23.1 | 69.2 | 94.1 | 84.9 | 49.91 |
| DeepSeek-V3.1 | CRP | 28.21 | 67.09 | 93.96 | 85.57 | 55.03 |
| Qwen-2.5-72B-Instruct | - | 22.8 | 63.2 | 95.4 | 86.9 | 48.98 |
| Qwen-2.5-72B-Instruct | SRP | 20.9 | 61.1 | 95.1 | 85.2 | 46.26 |
| Qwen-2.5-72B-Instruct | CRP | 24.39 | 55.74 | 95.12 | 87.51 | 49.46 |

improvements in both insight recall and report quality, suggesting that dynamically adapting the plan during execution helps the agent recover missed evidence and refine its use of sources. However, combining CRP or SRP with AAP does not yield clear gains, and in some cases reduces factuality, likely because overlapping strategies create redundant or unstable planning behavior. These findings indicate that adaptive mechanisms are key for improving coverage of injected insights, while lightweight planning is more effective for maintaining factual grounding, and that carefully balancing the two remains an open challenge. In particular, for stronger backbones such as GPT-5, complex planning promotes stepwise evidence aggregation across multiple files, which improves recall for insights that require combining numeric values with specific time periods or business details. See Appendix N for detailed results for each task.

## 5.3 ABLATION: EFFECT OF BACKBONE LANGUAGE MODEL ON DRBA

We evaluate the impact of backbone language models on DRBA using the FullBenchmark subset for controlled comparison. As the results in Table 3 shows, GPT-5 achieves the best balance of factual grounding, insight recall, and report quality. Open-source models show mixed results: Llama-3.1-405B excels in factuality but lags in recall, DeepSeek-V3.1 delivers balanced performance through targeted fine-tuning, and Qwen-2.5-72B is reliable but trails GPT-5. These results underline the importance of backbone choice; larger and more advanced models generally yield stronger overall performance, though some open-source options are competitive in specific metrics. In addition, our experiments reveal a significant limitation in agents' ability to retrieve critical insights from the open web. As shown in Table 22 in Appendix M, no agent successfully sourced external knowledge, highlighting the difficulty of extracting relevant information for deep research applications within an unboundedly large search space.

## 5.4 QUALITATIVE ANALYSIS

In Table 4 we show a sample of three groundtruth insights as well as the predicted insights from using both Llama 3.1 405B and GPT-5. We see that for the first insight, both models are able to effectively recover the groundtruth insight. For the second insight GPT-5 can extract the relevant time of year, where

as Llama-3.1 405B fails to do so. This possibly suggests that GPT-5 may be better at extracting fine details. These examples highlight that successful insight recall requires more than identifying salient numbers; it also requires correctly binding those values to their associated temporal or business context. Models that fail to perform this binding often produce superficially plausible but incomplete insights, which are scored as unsuccessful despite containing correct numeric fragments.

We observe systematic qualitative differences in how models recover injected insights. As illustrated in Table 4, larger backbones such as GPT-5 more reliably recover both numeric values and their associated contextual qualifiers (e.g., time period or business condition). In contrast, smaller or weaker models often restate isolated numeric facts without correctly attaching the relevant temporal or semantic context, resulting in lower insight recall despite surface-level correctness. Across all backbones, agents consistently fail to identify when required information is missing from private files and should instead be sourced from the web. For example, for the question "How can Lee's Market leverage FSMA 204 regulations to enhance food safety and customer trust?", FSMA 204 does not appear in the enterprise documents. However, agents generate broad queries such as "grocery store customer trust" or "food safety best practices", rather than targeted searches for FSMA 204 regulations. As a result, no agent successfully retrieves the relevant regulatory content, highlighting a limitation in detecting and acting upon missing-domain signals. This failure reflects limitations in problem scoping and query formulation rather than search execution itself, suggesting that robust missing-knowledge detection is a prerequisite for effective use of open-web retrieval in deep research settings.

Table 4: Insights Recall Improvement Areas (Task DR0002). We highlight in bold where each model was able to accurately find details relevant to the groundtruth insight. We also show the corresponding score where 1.0 is considered a successful recall and 0.0 an unsuccessful recall. The full table with all groundtruth insights and predicted insights is given in Appendix G.

| Groundtruth Insight | Insight Predicted by Llama 3.1 405B | Insight Predicted by GPT-5 |
|---|---|---|
| 45% of our online customers have interacted with personalized product recommendations, resulting in a 25% increase in average order value. | **45% of Lee's Market online customers engage with personalized product recommendations, resulting in a 25% increase in average order value.** (Score = 1.0) | **45% of online customers engaged with personalized product recommendations, and among those engagers average order value increased by 25%.** (Score = 1.0) |
| 85% of Lee's Market transactions are linked to customer loyalty accounts as of Q2 2024. | **85% of transactions are linked to loyalty accounts at Lee's Market**, providing a solid foundation for personalized marketing and improving customer engagement. (Score = 0.0) | **As of Q2 2024, 85% of transactions were linked to loyalty accounts**, leaving a 15% unlinked identity gap. (Score = 1.0) |

## 5.5 Performance of Web Agents on *DRBench*

We evaluated Generic WebAgents from AgentLab in a browser-only setting (without API access). The GPT-4.1-powered agent achieved only **1.11% insight recall**, **6.67% factuality**, and **33.07% report quality**. While the reports appeared well-structured, they lacked grounded insights, with most trajectories degenerating into repetitive clicks on irrelevant files or windows. This shows that browser-only agents are currently far from effective for deep research tasks. Further trajectory examples are shown in Appendix J.

## 5.6 App-based Environment vs Local Environment

In Table 5, we compare results across two settings in *DRBench*: (1) **local**, where all the task files (e.g., PDFs, PPTX, DOCX, XLSX, chats) are directly passed to the agent, and (2) **app-based**, where the same files must be retrieved through our standard enterprise environment and its apps, introducing additional interaction complexity. We find that OpenAI's Deep Research (GPT-5) achieves the highest scores across all metrics. Our agent with GPT-5 and DeepSeek backbones achieves similar performance to Perplexity in the local-only setting, but lags behind OpenAI and Gemini. In the app-based setting, performance declines across both backbones, highlighting the added difficulty of navigating multi-application environments. This gap underscores that the environment in *DRBench* is intentionally challenging, enabling a more realistic evaluation of model capabilities in enterprise research scenarios.

We observe that weaker backbones experience substantially larger performance drops when transitioning from the local to the app-based environment, particularly in insight recall and factuality. This degradation reflects the increased difficulty of multi-step navigation and cross-application context switching. In

Table 5: Model Performance Comparison Across Local or App-based Environments on the FullBenchmark. Note that higher numbers correspond to better scores, and the best result on each metric is bolded.

| Model | Env | Insight Recall | Factuality | Distractor Avoidance | Report Quality | Harmonic Mean |
|---|---|---|---|---|---|---|
| DRBA (GPT-5) | App | 36.52 | 72.11 | 93.22 | 93.41 | 63.81 |
| DRBA (GPT-5) + CRP | App | 37.48 | 62.33 | 91.71 | 92.03 | 62.02 |
| DRBA (DeepSeek-V3.1) | App | 22.6 | 68.4 | 94.9 | 84.1 | 49.20 |
| DRBA (DeepSeek-V3.1) + CRP | App | 28.21 | 67.09 | 93.96 | 85.57 | 55.03 |
| DRBA (GPT-5) | Local | 38.91 | 75.84 | 95.46 | 92.18 | 66.43 |
| DRBA (GPT-5) + CRP | Local | 39.74 | 77.02 | 95.21 | 93.37 | 67.39 |
| DRBA (DeepSeek-V3.1) | Local | 31.02 | 72.31 | 95.64 | 86.94 | 58.80 |
| DRBA (DeepSeek-V3.1) + CRP | Local | 31.88 | 73.42 | 95.27 | 87.86 | 59.82 |
| Perplexity | Local | 34.21 | 76.08 | 96.12 | 87.41 | 62.29 |
| OpenAI Deep Research (GPT-5) | Local | **41.96** | **83.64** | **96.88** | **93.02** | **70.35** |
| Gemini | Local | 40.88 | 81.32 | 96.41 | 91.54 | 68.90 |

contrast, stronger models such as GPT-5 exhibit more stable performance, indicating greater robustness to interaction complexity.

## 6  HUMAN EVALUATION

**Quality of Deep Research Questions.**    We evaluated the quality of the deep research questions in *DRBench* through a human study with five expert annotators across the first 15 tasks. Each task was judged on three criteria: (1) grounding in the external website, (2) relevance to the domain and company context, and (3) alignment with associated insights. Annotators provided binary ratings plus optional feedback. Results show strong quality: 12 tasks received unanimous approval, while only three (tasks DR1, DR11, and DR13) received a single negative vote due to minor issues with specificity or distractor difficulty. This corresponds to a 96% approval rate (72/75 votes).

**Correlation of Used Metrics with Human Preference.**    We collected human preference on a subset of 11 tasks[1]. Each annotator was shown a groundtruth insight with aligning insights from two models[2] and asked to choose which they preferred, or label both as good/bad. Missing alignments were shown as empty strings. We compared agents with AAP no RP against GPT-5 and Llama-3.1-405B-Instruct. The Fleiss $\kappa$ (Fleiss, 1971) across five annotators was 0.67. Most outputs were judged *both bad* due to missing alignments, but when preferences were expressed, GPT-5 was favored 61.1% over Llama-405B-Instruct, consistent with our metric-based findings in Section 5.3. Additional analyses are in Appendix R.

**Human Validation of the LLM-as-a-Judge Evaluation**    To validate the reliability of our LLM-as-a-judge evaluation protocol, we conducted an additional human study. Specifically, we recruited four human evaluators and asked them to determine, for each predicted insight, whether it appeared in the corresponding ground truth insight list. We then compared the human judgments against the LLM-as-a-judge decisions used in our benchmark. Across **264** evaluated insights, we observed over **91.3% agreement** between the human annotators and the LLM-based judge. We further measured inter-rater alignment using Cohen's $\kappa$ (Cohen, 1960), obtaining a score of **0.683**, which indicates *substantial agreement* (Gwet, 2001). These results confirm that our LLM-as-a-judge setup closely aligns with human judgment and provides a reliable and scalable evaluation mechanism for insight recall.

## 7  CONCLUSION

In this work, we introduced *DRBench*, a benchmark for evaluating AI agents on complex enterprise deep research tasks that require reasoning over both public and private data. *DRBench* provides 100 persona-grounded tasks situated in realistic enterprise environments, integrating heterogeneous data formats and real-world applications. We also presented DRBench Agent (DRBA) as a strong baseline and analyzed its behavior across planning strategies and backbone models. Our results show that while agents effectively avoid distractors and produce structured reports, they still struggle to reliably extract decision-critical insights. Adaptive planning improves insight recall, whereas simpler strategies better preserve factual accuracy, highlighting a fundamental trade-off between exploration and reliability.

---

[1]We selected tasks with fewer than 8 insights for a reasonable amount of manual work.

[2]The gold-prediction alignment is provided by the insight recall metric.

ETHICS STATEMENT

This work raises important considerations around data privacy, fairness, and potential misuse. Although *DRBench* simulates enterprise research environments with private data, all datasets are synthetically generated or drawn from public, time-invariant web sources. No personal or sensitive user data is included. The synthetic personas and companies are fictional, designed to prevent any risk of harm or re-identification. We highlight that agents evaluated on *DRBench* must handle sensitive-like contexts (e.g., healthcare, compliance, cybersecurity), which underscores the importance of designing systems that prioritize data protection and avoid exposing private enterprise content. Human annotators were involved in validating task quality; they were compensated at fair rates and gave informed consent.

Large Language Models (LLMs) were used solely to assist with polishing the writing of this paper, such as improving readability and clarity of exposition. All ideas, experimental designs, implementations, analyses, and conclusions are original contributions of the authors.

REPRODUCIBILITY STATEMENT

We have taken multiple steps to ensure reproducibility. The *DRBench* benchmark, including all generated tasks, data generation scripts, supporting files, and evaluation scripts, will be released under a permissive license. Each task is fully self-contained with dated URLs for public insights and synthetic enterprise files for private insights, ensuring stability over time. Detailed descriptions of the task generation pipeline, environment implementation, evaluation prompts, and cost considerations are included in the supplementary materials. We provide open-source code for running agents in the *DRBench* environment and for reproducing all reported results. Hyperparameter settings, backbone models, and planning strategies are documented. Together, these design choices make our benchmark transparent, reproducible, and extensible for future research.

LIMITATIONS

While DRBench captures realistic enterprise deep research workflows, it has several limitations that reflect deliberate design trade-offs. First, the benchmark currently covers a finite set of enterprise task families and domains, prioritizing depth and realism over exhaustive breadth; expanding to additional task archetypes (e.g., cross-team decision making, longitudinal audits, or incident response) remains an important direction for future work. Second, our evaluation operates at the level of atomic insights rather than span- or token-level grounding, which limits fine-grained attribution analysis but enables scalable, robust assessment of long-horizon research outputs; incorporating more granular grounding signals is a promising extension. Finally, despite strong human validation results, our evaluation pipeline retains a residual dependence on LLM-as-a-judge methods, which may introduce subtle biases; future iterations will explore hybrid human–automatic evaluation and alternative grounding-aware metrics to further strengthen reliability.

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

Table 6: Comparison of Deep Research Tasks of Different Benchmarks.

| Benchmark | Sample Question |
|---|---|
| DeepResearchGym (Coelho et al., 2025) | Is the COVID vaccine dangerous |
| Deep Research Bench (Bosse et al., 2025) | Find a reliable, known number on the internet. The total number of FDA Class II Product Recalls of medical devices. |
| DeepResearch Bench (Du et al., 2025) | While the market features diverse quantitative strategies like multi-factor and high-frequency trading, it lacks a single, standardized benchmark for assessing their performance across multiple dimensions such as returns, risk, and adaptability to market conditions. Could we develop a general yet rigorous evaluation framework to enable accurate comparison and analysis of various advanced quant strategies? |
| ResearcherBench (Xu et al., 2025b) | Compare the Transformer and Mamba model architectures, analyzing their performance and technical characteristics in different application scenarios. Based on the latest research, discuss the advantages and disadvantages of both models and their applicable scenarios. |
| LiveDRBench (Java et al., 2025) | For complex reasoning tasks (e.g., tasks involving multiple citations or extended reasoning chains), what are the strengths of current agent technologies, and what are their limitations? Please analyze this in the context of research since June 2024. |
| BrowseComp-Plus (Chen et al., 2025) | Identify the title of a research publication published before June 2023, that mentions Cultural traditions, scientific processes, and culinary innovations. It is co-authored by three individuals: one of them was an assistant professor in West Bengal and another one holds a Ph.D. |
| GAIA2 (Andrews et al., 2025) | Update all my contacts aged 24 or younger to be one year older than they are currently. |
| *DRBench* | How can Lee's Market leverage FSMA 204 regulations to enhance food safety and customer trust? |

## A    COMPARISON OF DEEP RESEARCH BENCHMARKS AND AI AGENT BENCHMARKS WITH A COMPUTER ENVIRONMENT

In Table 1, we compare existing deep research benchmarks and AI agent benchmarks that provide a computer environment with *DRBench*. While the questions in existing benchmarks focus on public interest topics and require generic web search and computer use, *DRBench* provides realistic questions that real personas in organizations need to resolve.

## B    *DRBench* TASKS

As shown in Tables 7, 29, 30, 31, 32, 33, and 34 *DRBench* contains 100 tasks in total, covering 3 industries (retail, healthcare and electric vehicles), 10 task domains (compliance, sales, customer relationship management, market analysis, customer service management, IT service management, cyber security, marketing, quality assurance, and research), and 3 difficulty levels (easy, medium, hard). In addition, we generate the following 3 companies (one for each industry type): (1) a supermarket chain called Lee's Market, (2) a virtual healthcare company called MediConn Solutions, and (3) an electric vehicle company called Elexion Automotive.

Table 8 presents a deep research question from *DRBench* and its supporting groundtruth insights. We also visualize the DR Question and all QA pairs by embedding them with OpenAI's `text-embedding-3-large` model and projecting into 2D using t-SNE in Figure 4. The plot shows that injected supporting insights lie closer to the DR Question, while distractors appear farther away, confirming that our injected insights are semantically aligned with the research objective.

## C    *DRBench* EXAMPLES OF INJECTED INSIGHTS

As shown in Figure 2, supporting documents are generated with enterprise insights injected. In Figure 5, we show two examples of a generated files (PPTX and Mattermost chat) with their embedded insights.

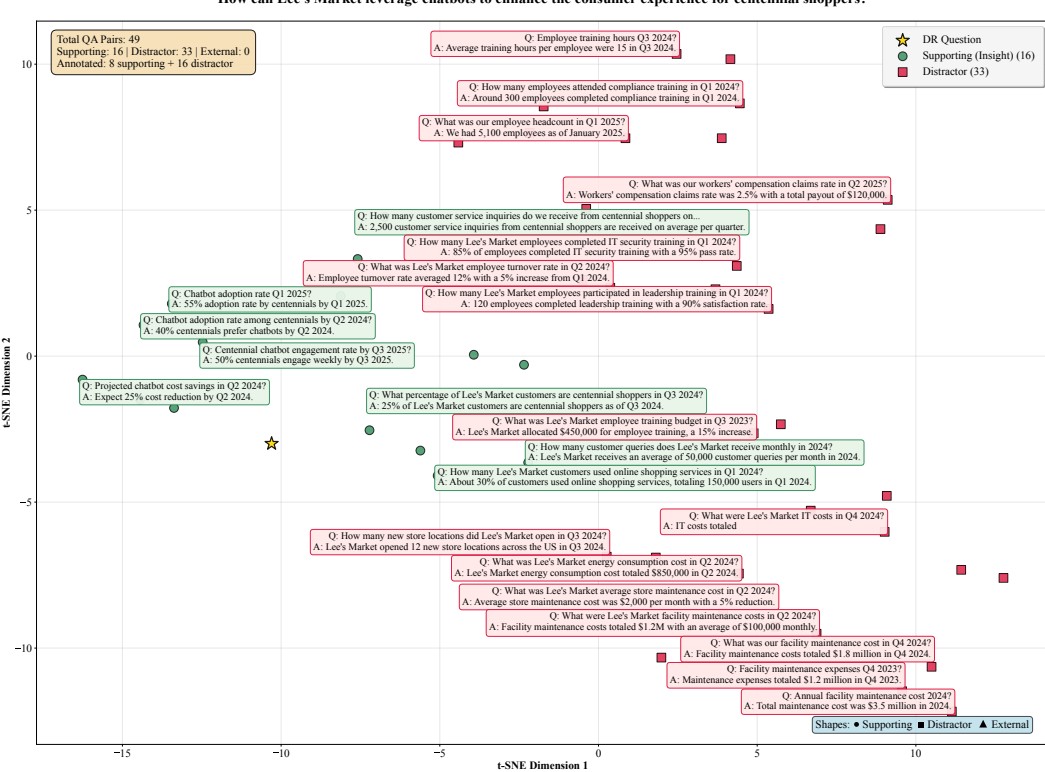

Figure 4: t-SNE visualization of QA pairs for the DR Question in Task DR0005. The plot shows the distribution of annotated pairs across **Supporting Insights** (green), **Distractors** (red), and the central **Deep Research (DR) Question** (gold star). Out of 49 pairs, 16 correspond to supporting insights and 33 are distractors. The visualization illustrates how relevant insights cluster separately from distractors, highlighting the challenge of retrieving salient information in a distractor-heavy environment.

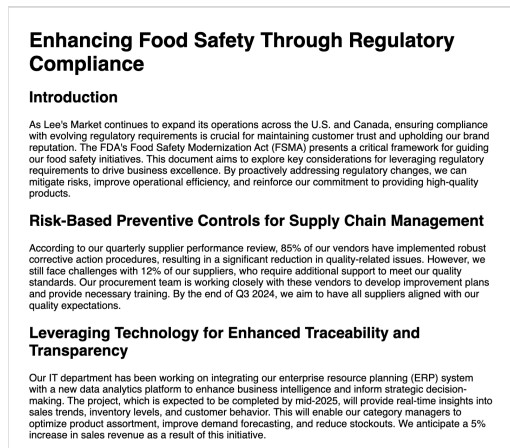

(a) Example supporting file named *food-safety-regulatory-compliance.pdf* with an injected insight "*Lee's Market reduced food waste by 8% in Q2 2024, saving $1.2M.*"

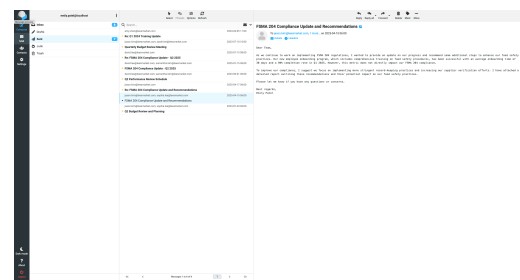

(b) Example supporting email in the Sent mailbox with an injected insight "*85% of high-risk food suppliers are FSMA 204 compliant, totaling 270 vendors.*"

Figure 5: Example files with injected insights in *DRBench*.

Table 7: *DRBench* Questions and Statistics. **Industry**: target industry of the deep research question. **Domain**: domain of the deep research task. **DR Question**: question of the deep research task. **Difficulty**: difficulty of the task defined based on the rubric mentioned in Section 3. **# Applications**: the number of total applications in the task environment. **# Insights**: the number of relevant insights to the deep research question. **# Distractors**: the number of non-supportings documents that do not contain relevant insights. Please refer to Tables 29, 30, 31, 32, 33and 34 for details on the details on the remaining 60 tasks.

| Industry | Domain | DR Question | Difficulty | # Applications | # Insights | # Distractors |
|---|---|---|---|---|---|---|
| Retail | Compliance | How can Lee's Market leverage FSMA 204 regulations to enhance food safety and customer trust? | easy | 2 | 3 | 7 |
| Retail | Sales | How can personalization drive sales in the retail industry and what strategies can be used for Lee's Market in action? | easy | 2 | 4 | 10 |
| Retail | CRM | How can we leverage data-driven loyalty programs to enhance customer engagement? | medium | 4 | 7 | 15 |
| Retail | Market Analysis | What are the new trends in the grocery retail market and what strategies can Lee's Market adopt to remain competitive? | medium | 3 | 6 | 14 |
| Retail | CSM | How can Lee's Market leverage chatbots to enhance the consumer experience for centennial shoppers? | hard | 4 | 16 | 33 |
| Healthcare | Compliance | What are the key factors influencing MediConn Solutions' decision to accept insurance for telehealth providers, considering HIPAA compliance and state-specific data privacy regulations? | easy | 3 | 4 | 8 |
| Healthcare | ITSM | How can we leverage ISTM and AI-driven analytics to minimize IT service desk workload and improve response times in MediConn Solutions? | easy | 3 | 6 | 12 |
| Healthcare | Cybersecurity | What is the impact of third-party data breaches on MediConn's virtual healthcare platforms and patient data, and what new regulations can be implemented to defend against these breaches? | medium | 4 | 7 | 14 |
| Healthcare | CRM | How can MediConn Solutions leverage trendy new CRM solutions to improve patient engagement and retention, and what new CRM solutions are expected in 2025? | medium | 4 | 12 | 24 |
| Healthcare | Marketing | What are the most critical elements of a robust digital presence for a telehealth provider such as MediConn Solutions, and how can we optimize our website and content marketing strategy to attract digital-first patients? | hard | 4 | 15 | 30 |
| Electric Vehicle | Compliance | How can we balance the need for durability and warranty guarantees for EV batteries with evolving regulatory requirements, especially ACC regulations (ACC II), while staying on track with our production timelines through 2035? | easy | 2 | 3 | 6 |
| Electric Vehicle | Quality Assurance | How can Elexion Automotive's quality assurance processes be optimized to address the unique challenges of electric vehicle production, such as software and user experience issues, compared to gasoline cars? | easy | 1 | 3 | 6 |
| Electric Vehicle | Cybersecurity | How can Elexion Automotive effectively implement a cybersecurity strategy for its electric vehicles, considering the risks and challenges posed by connected and autonomous technologies? | medium | 3 | 6 | 12 |
| Electric Vehicle | Research | Can we leverage AI-enhanced battery management to improve EV battery lifespan by 15%? | medium | 3 | 7 | 14 |
| Electric Vehicle | CSM | How can Elexion Automotive increase customer trust through after-sales support while balancing the need for exceptional customer care with efficient and cost-effective service? | hard | 4 | 15 | 30 |

# D  DRBENCH ENTERPRISE ENVIRONMENT

The *DRBench* Enterprise Environment provides a containerized simulation of realistic enterprise research settings where employees access confidential company information, personal files, and internal communications for comprehensive report generation. The environment simulates both a user's local machine filesystem and provides password-protected access to enterprise services.

To emulate realistic enterprise research settings, *DRBench* provides a self-contained Docker environment that integrates commonly used applications: Nextcloud for shared documents, Mattermost for internal chat, an IMAP server and Roundcube open-source client for emails, and Filebrowser to emulate local files. Each task is initialized by distributing its data across these services, enabling agents to retrieve, analyze, and cite information through enterprise-like interfaces rather than static files. This design ensures realistic interaction while maintaining reproducibility and controlled evaluation.

Table 8: Example Deep Research Question and Supporting Groundtruth Insights

| Deep Research Question | Supporting groundtruth insight | Insight Category |
|---|---|---|
| How can Lee's Market leverage FSMA 204 regulations to enhance food safety and customer trust? | U.S. grocers are working to meet the FDA's FSMA 204 traceability rules by January 2026, which require tracking lot codes and key data for high-risk foods to expedite recalls. This compliance is viewed as an "evolutionary step" to modernize grocery operations and enhance food safety. | External |
| | By capturing detailed traceability data, such as lot codes, at every step, retailers can meet regulations and gain inventory benefits. This allows grocers to know exact expiration dates by lot, enabling them to discount items before they expire, thus reducing food waste and keeping products fresher. | External |
| | Regional grocers like Lunds & Byerlys and Raley's see FSMA 204 as a chance to enhance their systems and supply chain transparency. They believe improved traceability will boost customer trust and could signal the start of more extensive future food safety regulations. | External |
| | Lee's Market tracks 250 high-risk food products as of Q3 2024, affecting 30% of inventory. | Internal |
| | Lee's Market reduced food waste by 8% in Q2 2024, saving $1.2M. | Internal |
| | 85% of high-risk food suppliers are FSMA 204 compliant, totaling 270 vendors. | Internal |

## D.1 ARCHITECTURE AND SERVICES

The environment implements a **multi-service architecture** within a single Docker container. This design prioritizes deployment simplicity and cross-platform compatibility while maintaining service isolation. The container orchestrates the following enterprise services:

- **Nextcloud**: Open-source file sharing and collaboration platform analogous to Microsoft SharePoint or Google Drive, providing secure document storage with user authentication.

- **Mattermost**: Open-source team communication platform simulating internal company communications similar to Microsoft Teams or Slack, with teams, channels, and persistent chat history.

- **FileBrowser**: Web-based file manager providing access to the container's local filesystem, simulating employee desktop environments and local document access.

- **Email System**: Roundcube webmail interface with integrated SMTP (postfix) and IMAP (dovecot) services for enterprise email communication simulation.

- **VNC/NoVNC Desktop**: Protocol and browser-based VNC access providing full desktop environment interaction within the container for comprehensive enterprise workflow simulation.

## D.2 TASK LOADING AND DATA DISTRIBUTION

At initialization, the environment processes task configuration files (`env.json`) and distributes data across services through automated Python scripts and it makes sure that this source data is only accessible through the intended applications:

- **File Distribution**: Documents are placed in appropriate Nextcloud user folders and FileBrowser directories based on task specifications

- **Communication Import**: Chat histories and team conversations are imported into Mattermost channels with proper user attribution

- **Email Integration**: Email conversations are loaded into the mail system with realistic threading and metadata

- **User Provisioning**: Enterprise users are automatically created across all services with consistent authentication credentials

```python
from drbench import drbench_enterprise_space, task_loader

# Load task configuration
task = task_loader.get_task_from_id(task_id)

# Initialize environment with automatic port allocation
env = drbench_enterprise_space.DrBenchEnterpriseSearchSpace(
    task=task.get_path(),
    start_container=True,
    auto_ports=True  # Prevents port conflicts in parallel execution
)

# Environment provides service discovery
available_apps = env.get_available_apps()
# Returns: {'nextcloud': {'port': 8081, 'credentials': {...}}, ...}

# Pass relevant information to the agent

# Cleanup when research complete
env.delete()
```

Listing 1: DrBench Environment Usage

### D.3 PYTHON INTEGRATION

The `DrBenchEnterpriseSearchSpace` class provides programmatic container management with the following capabilities: **container lifecycle management**, **service access information**, **task-specific data loading**, and **automatic cleanup**. The typical usage pattern shown in Listing 1 demonstrates these integrated capabilities.

### D.4 ENTERPRISE SERVICE APIS

Each service exposes both **web interfaces** for human and web-agent interaction, and **programmatic APIs** for agent access:

- **Nextcloud**: WebDAV API for file operations, sharing, and metadata retrieval
- **Mattermost**: REST API for message history, channel management, and user interactions
- **Email**: IMAP/SMTP protocols for message retrieval and sending
- **FileBrowser**: HTTP API for filesystem operations and file management

This dual-access model enables both agent-driven research and human verification of enterprise scenarios, supporting comprehensive evaluation of research capabilities across realistic enterprise information architectures.

## E   *DRBench* EXAMPLES OF APPLICATION SCREENSHOTS

Figures 6 and 7 show the applications provided in *DRBench* environment: File Browser, Mattermost, Roundcube, and Nextcloud.

## F   *DRBench* AGENT IMPLEMENTATION DETAILS

### F.1 DETAILED WORKFLOW

As depicted in Figure 3, the workflow begins with a Company Employee submitting an enterprise Deep Research Question along with Company Context. The *DRBench* agent processes this input through several key stages:

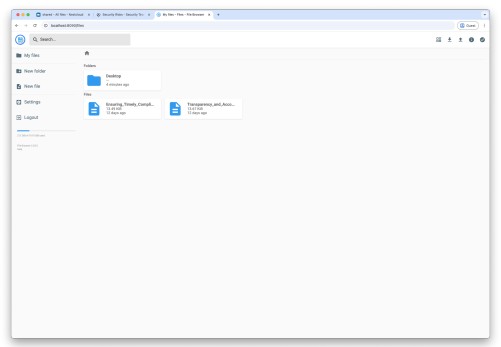

(a) Screenshot of the File Browser interface, displaying organized files and folders within the system.

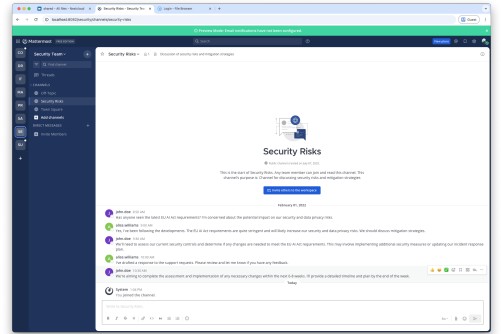

(b) Screenshot of the Mattermost communication platform, showing a discussion channel and user interface elements.

Figure 6: Screenshots of Applications in *DRBench* environment (Part 1).

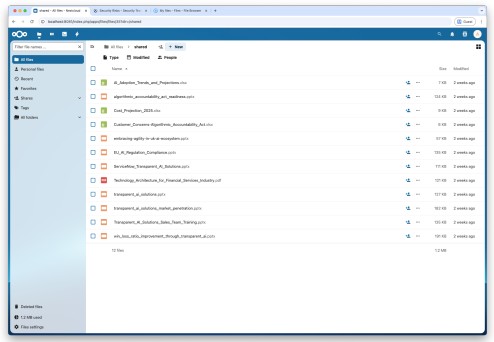

(a) Screenshot of the Nextcloud file management system, illustrating the file list view with various document types.

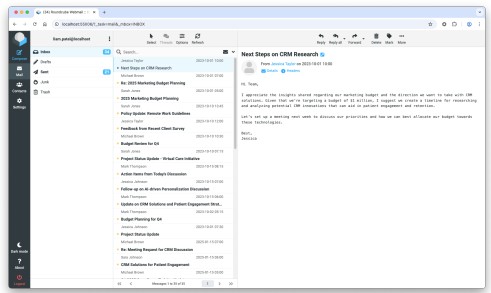

(b) Screenshot of Roundcube, an email client, it shows an open email in the user's inbox.

Figure 7: Screenshots of Applications in *DRBench* environment (Part 2).

**Stage 1: Research Planning.** The agent decomposes the research question into structured research investigation areas to guide subsequent action generation. This initial decomposition lays out a strategy to systematically cover the research space while maintaining focus on the initial deep research question.

**Stage 2: Action Planning.** The Action Planning stage translates the research objectives into executable actions through a planning subsystem. This component uses an LLM to create a prioritized sequence of actions. Each action is parameterized with specific tool selection and execution parameters, its dependencies to other actions in the plan, and a priority score.

**Stage 3: Research Loop with Adaptive Execution.** The Research Loop iterates over the following sub-stages until completion: (1) Tool Selection and Execution: The tool selection and execution subsystem implements a sophisticated priority-based selection of actions from the plan at each research iteration step and proceeds to execute it with the current research context. (2) Content Processing: If necessary, the action will make use of the content processing subsystem to extract, synthesize, and store retrieved documents and websites into a vector store to form a task-specific knowledge base that will grow on each iteration. (3) Adaptive Action Planning: After each execution round, the agent analyzes the most recent findings; if coverage gaps are detected, new actions are created and added to the plan at this point. This ensures that newly discovered knowledge is taken into account to answer the research question. (4) Iteration Findings Storage: Results from each iteration are stored in the vector store with rich metadata for traceability and later assessment.

**Stage 4: Convergence and Completion.** The research loop continues until all actions in the plan have been completed or the maximum iteration limit is reached.

**Stage 5: Report Writing.** The report writing subsystem queries the vector store to synthesize the research findings and relevant retrieved content. This component generates a comprehensive report and uses its own citation tracking system to ensure proper attribution of claims within the report.

The vector store serves as the main knowledge integration component, maintaining embeddings of all processed content and enabling semantic retrieval at the report generation stage. This component is crucial in a deep research setting to prevent information loss from early research stages in arbitrarily long research sessions.

### F.2 RESEARCH PLANNING IMPLEMENTATION

The Research Planning subsystem offers three operational modes to evaluate the impact of structured planning on research effectiveness:

- **Complex Mode:** Generates a comprehensive research plan with detailed investigation areas. These areas contain details about the specific research focus, expected information sources, and success criteria, among others. Each area includes an importance level and specific business intelligence objectives (see Listing 2).

- **Simple Mode:** Creates focused question decompositions with 4-10 self-contained subqueries derived directly from the main research question. Uses straightforward decomposition without the complex enterprise research structure of complex mode. See examples in Listing 4 and Listing 3 for comparison of different planning modes.

- **None:** Bypasses structured planning entirely, proceeding directly to action generation based on the original question. This mode serves as a baseline to measure the added-value of explicit planning stages.

The planning process begins with the enriched query (Prompt 1) and uses the research planning prompt (Prompt 2) to generate structured outputs. In Complex Mode, the system creates detailed investigation areas with enterprise-focused metadata, while Simple Mode produces straightforward question decompositions similar to existing multi-step reasoning approaches. The resulting plan structure directly feeds into the Action Planning System (Appendix F.3) for executable action generation.

```
{
  "area_id": 1,
  "research_focus": "Core strategic
    domain, market segment, or business hypothesis to investigate",
  "information_needs": [
    "What specific intelligence is required for strategic decisions"
  ],
  "knowledge_sources": ["internal", "external", "both"],
  "research_approach
    ": "competitive_analysis | market_research | strategic_assessment
    | trend_analysis | risk_analysis | performance_benchmarking",
  "key_concepts": ["concept1", "concept2"],
  "business_rationale": "Why this investigation
    area is critical for enterprise strategy and decision-making",
  "expected_insights": "What strategic
    understanding or competitive intelligence this area should provide",
  "stakeholder_impact": "Which
    business units or decision-makers will benefit from these insights",
  "importance_level": "critical | important | supplementary"
}
```

Listing 2: Investigation Area Structure for Full Planning Mode

```
{
  "query": "What are the new trends in the grocery retail market
    and what strategies can Lee's Market adopt to remain competitive?",
  "plan": {
    "research_investigation_areas": [
      {
```

```
      "area_id": 1,
      "research_focus": "Current trends in grocery retail market",
      "information_needs": [
      "Latest consumer preferences",
      "Emerging technologies influencing grocery shopping",
      "Sustainability practices in grocery retail",
      "Changes in supply chain dynamics"
      ],
      "knowledge_sources": ["external"],
      "research_approach": "trend_analysis",
      "key_concepts
    ": ["e-commerce growth", "sustainability in supply chains"],
      "business_rationale": "Understanding consumer
     trends and technological advancements that shape shopper behavior is
     critical for adapting offerings and enhancing customer engagement.",
      "expected_insights": "Identify specific trends affecting customer
     buying decisions, including the rise of online grocery shopping
     and preferences for sustainable, local, or organic products.",
      "stakeholder_impact
    ": "Marketing, Product Development, Supply Chain Management",
      "importance_level": "critical"
    },
    {
      "area_id": 2,
      "research_focus": "Competitive analysis of grocery retailers",
      "information_needs": [
      "Market share analysis",
      "Competitor strengths and weaknesses",
      "Innovative strategies adopted by competitors"
      ],
      "knowledge_sources": ["external"],
      "research_approach": "competitive_analysis",
      "
    key_concepts": ["market positioning", "competitive differentiation"],
      "business_rationale": "A comprehensive understanding of
     competitors allows for strategic positioning and the identification
     of innovative practices that can be adopted or improved upon.",
      "expected_insights": "Detailed profiles of key
     competitors, including strategic moves they are making to capture
    market share, which can inform Lee's Market's competitive strategy.",
      "stakeholder_impact
    ": "Executive Leadership, Strategic Planning, Marketing",
      "importance_level": "critical"
    },
    ...
    ]
  }
}
```

Listing 3: Complex Mode Research Plan Example

```
{
  "query": "How can we leverage
     data-driven loyalty programs to enhance customer engagement?",
  "plan": {
    "mode": "simple",
    "subqueries": [
    "What are the key features
     of successful data-driven loyalty programs in the retail industry?",
    "How can data analytics be used to personalize rewards
    and incentives in loyalty programs to increase customer engagement?",
    "What types of customer data should be collected and analyzed
     to optimize loyalty programs for a company like Lee's Market?",
```

```
        "Which
        technology platforms and tools are most effective for implementing
        and managing data-driven loyalty programs in the retail sector?",
        "How can Lee's Market measure the success of their
        loyalty program in terms of customer engagement and sales growth?",
        "What are the best practices for integrating a loyalty program
        with existing marketing strategies to enhance customer experience?",
        "How can Lee's Market ensure customer data
      privacy and security while leveraging data-driven loyalty programs?",
        "What are the potential
        challenges and limitations of implementing a data-driven loyalty
      program for a retail company with Lee's Market's size and resources?"
        ],
        "research_methodology": {
        "overall_approach": "Query decomposition into focused subqueries"
        }
    }
}
```

Listing 4: Simple Mode Research Plan Example

## F.3    ACTION PLANNING SYSTEM

The Action Planning System translates research objectives into executable actions through an intelligent planning subsystem that manages tools, prioritization, and dependencies.

**Available Tools**    Table 9 summarizes the available tools organized by category and their primary purposes.

Table 9: DrBench Agent Tool Categories and Purposes

| Category | Tool | Purpose |
|---|---|---|
| Information Retrieval | Internet Search | External market research, competitive intelligence, and public data analysis. Ideal for market trends, competitor analysis, industry reports, news articles. |
| | Enterprise API | Access to proprietary internal data through extensible adapters (Nextcloud, Mattermost, email, FileBrowser). Ideal for internal metrics, communications, confidential documents. |
| | URL Fetch | Direct content extraction from specific URLs. Ideal for deep analysis of reports, whitepapers, case studies, competitor websites. |
| Analysis | Analyzer | AI-powered synthesis and analysis using vector search. Ideal for cross-referencing findings, identifying patterns, generating insights. |
| Local Processing | Local Document Search | Semantic search within locally ingested documents. Ideal for targeted retrieval from local files with source references. |

**Priority Scoring and Dependencies**    Actions are assigned priority scores (0.0-1.0 scale) based on strategic importance and expected information value. The priority assignment follows enterprise research principles:

- **Source Type Prioritization:** Enterprise and local sources receive higher priority than external sources, reflecting the strategic value of proprietary information in competitive analysis.

- **Query Specificity:** Targeted queries addressing specific investigation areas score higher than broad exploratory searches, ensuring focused research execution.

- **Dependency Management:** Actions can specify prerequisite relationships where certain information gathering must precede analysis or synthesis tasks. The scheduler respects these dependencies while maximizing parallel execution within each iteration.

### F.4 ENTERPRISE INTEGRATION ARCHITECTURE

**Service Adapters** The system implements extensible adapters for enterprise services including Nextcloud file server, Mattermost chat, IMAP email systems, and FileBrowser interfaces. Each adapter handles service-specific authentication, data retrieval, and metadata preservation for proper citation attribution.

**Source Prioritization Strategy** Enterprise and local sources receive priority multipliers in action scoring, reflecting the strategic value of proprietary information. The system maintains source type classification throughout the pipeline to ensure internal intelligence drives analytical conclusions while external sources provide context and validation.

### F.5 TOOL SELECTION AND EXECUTION

The Tool Selection stage implements a priority-based action selection and tool invocation within each research iteration to execute the action plan.

**Action Selection Process** At each iteration, the system selects executable actions based on priority scores and dependency satisfaction:

- **Priority-Based Scheduling:** Actions are ranked by their priority scores, with enterprise and local sources prioritized over external sources to maximize the value of specific private information.
- **Dependency Validation:** The scheduler checks that all prerequisite actions have completed before making an action available for execution.
- **Sequential Execution:** Actions execute one at a time in priority order, maintaining research coherence and enabling each action to build upon previous findings.

Once selected, actions execute through a standardized interface and results are integrated into the research context, informing the following stage of adaptive action planning.

### F.6 ADAPTIVE PLANNING

The Adaptive Planning system enables dynamic evolution of the action plan by analyzing results after each iteration to generate extra actions addressing information gaps.

It starts by analyzing the most recently completed actions and performs these two substages:

**Source analysis and gap classification.** The system evaluates the possible imbalances in the action completion if information came from internal or external sources and identifies possible scenarios to cover.

**Dynamic action generation.** After analyzing the sources and results from the previous actions, the system makes an LLM call to generate 1-5 extra candidate actions with a specific prioritization. After candidate actions are generated, they go through a deduplication process to make sure the plan didn't cover them already and incorporates the final subset into the priority action plan so they can be considered by the scheduler in the following iteration.

### F.7 CONTENT PROCESSING AND VECTOR STORE

The Content Processing system implements a pipeline for unified ingestion of documents in multiple formats (PDF, docx, HTML, JSON, plain text formats, etc.) that normalizes and cleans text inside the documents and websites retrieved during the research. Content is then deduplicated and chunkized, and embeddings are computed for each of these chunks.

The Vector Store implements the storage and retrieval of the content via JSON metadata and NumPy embedding metrics, enabling semantic similarity and keyword based searches

## F.8 REPORT GENERATION

**Multi-Stage Synthesis Pipeline**   The Report Generation stage implements a four-stage synthesis approach: (1) thematic content clustering via vector searches, (2) source prioritization and deduplication, (3) LLM Synthesis with source tracking, and (4) Final report writing and citation resolution.

The system generates targeted search queries based on the research plan, including specific to ensure that a predefined set of themes or sections (background, analysis, implementation, trends) are retrieved and written and prevents redundant analyses.

**Unified Citation System**   The citation system implements deferred resolution to keep consistency of citations from section to section by referencing document IDs in the Vector Store and carrying them over for each piece of synthesized text. A final citation resolution stage will assign the correct numbering to each document in the final report.

## F.9 DRBA PROMPTS

The *DRBench* agent relies on carefully designed prompts to orchestrate enterprise research workflows. These prompts implement the core architectural principles: enterprise context enrichment, structured planning decomposition, priority-based action generation, quantitative synthesis requirements, and adaptive research capabilities. The following five prompts represent the critical LLM interactions that enable systematic enterprise research with proper source prioritization and citation tracking:

- **Enriched Query Generation** (Prompt 1): Transforms basic research questions into enterprise-contextualized queries by incorporating company information, stakeholder personas, and business context to guide subsequent research activities.
- **Research Planning** (Prompt 2): Decomposes complex research questions into structured investigation areas with defined information needs, knowledge sources, and business rationales, enabling systematic coverage of the research space.
- **Action Generation** (Prompt 3): Converts research objectives into prioritized executable actions with tool specifications and dependency relationships, emphasizing enterprise source prioritization over external sources.
- **Adaptive Action Generation** (Prompt 4): Analyzes research progress to identify coverage gaps and source imbalances, generating complementary actions that enhance research depth and cross-validate critical findings.
- **Report Synthesis** (Prompt 5): Orchestrates quantitative-first content synthesis with strict citation requirements, ensuring numerical data leads analytical paragraphs and all claims are properly attributed to source documents.

---

**⚡ Query Generation**

```
Research Question:  {dr_question}

Company Context:  {company_name} is {company_desc}.

Persona Context:  This analysis is requested by {name}, {role} in
{department}.

Their responsibilities include:  {responsibilities}.
```

Prompt 1: Query Enrichment with Enterprise Context.

## G QUALITATIVE RESULTS

Shown below are some illustrative examples of how metrics are computed for different scenarios on a given test task.

> **⚡ Research Planning (Complex Mode)**
>
> ```
> Design a comprehensive enterprise research strategy for:
> "{question}"
>
> {tools_section}
>
> As a senior enterprise researcher with deep business
> intelligence expertise, create a thorough investigation
> plan that combines rigorous research methodology with
> strategic business analysis.  Your goal is to provide insights
> that drive informed decision-making in complex enterprise
> environments.
>
> Enterprise Research Design Principles:
>
> - Leverage proprietary internal data as competitive advantage while
> ensuring external market context
> - Design investigations that directly inform strategic decisions and
> business outcomes
> - Prioritize research areas that maximize ROI and strategic value to the
> organization
> - Balance comprehensive analysis with focused insights relevant to
> enterprise objectives.
>
> Generate a JSON object with strategic research investigation
> areas:
>
> {output_structure}
> ```

.

Prompt 2: Enterprise Research Planning with Investigation Areas. See example of the output structure in Listing 2

### G.1   INSIGHTS RECALL

We showcase the insights recall metric using Task DR0002, whose DR Question is "How can personalization drive sales in the retail industry and what strategies can be used for Lee's Market in action?" (also shown in Table 7), which evaluates sales challenges and competitive landscape analysis.

Table 10 shows the overall performance comparison between using Llama 3.1 405B and GPT-5 in our agent. Using GPT-5 in our agent results in increasing the insights recall score from 0.14 to 0.43, successfully answering 3 out of 7 questions compared to Llama's 1 out of 7.

Table 10: Insights Recall Performance Comparison: Llama 3.1 405B vs GPT-5 (Task DR0002). We summarize the number of questions answered successfully and unsuccessfully as well as the overall insights recall score for the given task.

| Metric | Llama 3.1 405B | GPT-5 | Improvement |
|---|---|---|---|
| Insights Recall Score | 0.14 | 0.43 | **+0.29** |
| Questions Answered Successfully | 1/7 | 3/7 | **+2** |
| Questions Failed | 6/7 | 4/7 | **-2** |

The question-by-question breakdown in Table 11 reveals the specific questions where each approach succeeded or failed. Both models successfully identified the insight related to online customer engagement, but only GPT-5 was able to identify the number of loyalty program members and the customer data collection rate. Neither model successfully answered the remaining 4 questions, indicating these insights may not have been readily available in the source materials or that agents struggled to find the right insights.

---

**⚡ Action Generation**

```
Generate specific executable actions for this research investigation
area with SOURCE PRIORITIZATION:

Research Focus: {research_focus}

Information Needs: {information_needs}

Knowledge Sources: {knowledge_sources}

Research Approach: {research_approach}

Available Tools: {available_tool_names}

Tool Selection Guidelines: {tool_guidelines}

Return JSON array of actions with:
- "type": Action type (web_search, enterprise_api, url_fetch, analyzer,
local_search)
- "description": Clear description of what this action will accomplish
- "parameters": Tool-specific parameters including query, search_type,
etc.
- "priority": Float 0.0-1.0 (enterprise sources: 0.7-1.0, external:
0.4-0.7)
- "expected_output": What information this action should provide
- "preferred_tool": Specific tool class name to use
```

Prompt 3: Priority-Based Action Generation with Source Awareness

Table 11: Question-by-Question Insights Recall Analysis (Task DR0002). We breakdown the results question by question for the given task, highlighting specifically which question is answered correctly or incorrectly for each model.

| Question | Llama 3.1 405B | GPT-5 | Δ |
|---|---|---|---|
| Online Customer Engagement with Personalized Recommendations | 1.0 | 1.0 | 0.0 |
| Number of Loyalty Program Members | 0.0 | 1.0 | +1.0 |
| Customer Data Collection Rate | 0.0 | 1.0 | +1.0 |
| Online Sales Growth | 0.0 | 0.0 | 0.0 |
| Effectiveness of Personalized Marketing | 0.0 | 0.0 | 0.0 |
| Personalized Promotions vs Mass Promotions | 0.0 | 0.0 | 0.0 |
| Retail Media Growth | 0.0 | 0.0 | 0.0 |
| **Total Insights Recall Score** | **1.0/7.0** | **3.0/7.0** | **+2.0** |

In Table 12 we extend Table 4 and show all the groundtruth insights as well as each of the predicted insights from using both Llama 3.1 405B and GPT-5. As before, we highlight in bold where the model was able to accurately find details relevant to the expected insight, and show all the corresponding scores as given in Table 11.

## G.2 FACTUALITY

The factuality metric evaluation uses the same Task DR0002 to assess the accuracy and reliability of generated content. Table 13 presents the factuality performance comparison, showing that while using Llama 3.1 405B achieved 0.41 factuality (7 factual claims out of 17 total claims), where as using GPT-5 reached 0.65 factuality (13 factual claim out of 20 total claims). This represents a significant improvement

⚡ **Adaptive Action Generation**

```
Based on the research progress so far, suggest new actions with
INTELLIGENT SOURCE COMPLEMENTARITY:

Original Research Query: {research_query}

Completed Actions Summary: {completed_actions_summary}

Recent Findings Analysis: {findings_json}

Source Composition Analysis: {internal_findings}

Available Tools: {available_tool_names}

Generate 1-5 new actions that:
1.  **Address gaps** in current research coverage
2.  **Balance source types** - if findings are heavily external,
prioritize internal sources and vice versa
3.  **Build on discoveries** - leverage new information to explore
related areas
4.  **Enhance depth** - dive deeper into promising
findings
5.  **Cross-validate** - verify critical findings through alternative
sources

Each action should have:
- Strategic rationale for why this action is needed
 now
- Clear connection to research gaps or promising
 leads
- Appropriate priority based on strategic value and source
 balance
- Specific parameters that build on existing knowledge
```

Prompt 4: Gap-Driven Adaptive Action Generation

in content reliability. This also highlights that GPT-5 is much better prepared to make accurate claims that are sustained by evidence.

Table 14 provides a detailed breakdown of factual versus unfactual claims. The agent using GPT-5 generated 6 additional factual claims while producing 3 fewer unfactual claims, resulting in a net improvement in accuracy percentage. This demonstrates that GPT-5 may generate a higher proportion of factual information than Llama 3.1 405B.

The impact of these factuality improvements of using GPT-5 over Llama 3.1 405B is summarized in Table 15. The 24.0 percentage point improvement in factuality represents enhanced content quality and research reliability. The increase in 3 total claims shows that GPT-5 can generate more content overall.

## H  EXPERIMENTAL SETTINGS

All experiments were conducted on a cluster of 8 NVIDIA A100 GPUs (80GB each). For file generation and task construction, we primarily used the `Llama-3.1-405B` model, with decoding performed using nucleus sampling at a temperature of 0.7 unless otherwise specified. For larger-scale evaluations of the DRBench Agent, we also used closed-source models such as GPT-4o and GPT-5, alongside DeepSeek models, to enable comparison across open- and closed-source backbones.

To ensure reproducibility, the DRBench environment was deployed as a self-contained Docker container with all supporting applications (Nextcloud, Mattermost, and Filebrowser) pre-configured. Each task was

Table 12: Insights Recall Improvement Areas (Task DR0002). We highlight in bold where each model was able to accurately find details relevant to the groundtruth insight. We also show the corresponding score where 1.0 is considered a successful recall and 0.0 an unsuccessful recall.

| Groundtruth Insight | Insight Predicted by Llama 3.1 405B | Insight Predicted by GPT-5 |
|---|---|---|
| 45% of our online customers have interacted with personalized product recommendations, resulting in a 25% increase in average order value. | **45% of Lee's Market online customers engage with personalized product recommendations, resulting in a 25% increase in average order value.** (Score = 1.0) | **45% of online customers engaged with personalized product recommendations, and among those engagers average order value increased by 25%**. (Score = 1.0) |
| As of Q3 2024, Lee's Market has 1.2 million loyalty program members. | 85% of transactions are linked to loyalty accounts at Lee's Market, providing a solid foundation for personalized marketing and improving customer engagement. (Score = 0.0) | From Q2 2024 to **Q3 2024, loyalty members increased from 1,050,000 to 1,200,000 (+150,000; +14.29%)**, average spend per member rose from 24 to 25 (+4.17%), and total member spend increased from 25,200,000 to 30,000,000 (+19.05%). (Score = 1.0) |
| 85% of Lee's Market transactions are linked to customer loyalty accounts as of Q2 2024. | **85% of transactions are linked to loyalty accounts at Lee's Market**, providing a solid foundation for personalized marketing and improving customer engagement. (Score = 0.0) | As of Q2 2024, **85% of transactions were linked to loyalty accounts**, leaving a 15% unlinked identity gap. (Score = 1.0) |
| Lee's Market online sales grew 12% in Q2 2024 compared to Q2 2023. | 45% of Lee's Market online customers engage with personalized product recommendations, resulting in a 25% increase in average order value. (Score = 0.0) | A naive blended online AOV upside of approximately 11.25% is derived from 45% engagement multiplied by a 25% AOV lift among engagers. (Score = 0.0) |
| Retailers excelling in personalized marketing are growing revenues about 10 percentage points faster than their peers, according to BCG. By effectively using first-party customer data, these leaders could unlock an estimated $570 billion in additional sales, highlighting the importance of data-driven sales strategies for growth. | Retailers have seen a consistent 25% increase in revenue due to advanced **personalization capabilities**. (Score = 0.0) | Grocers running **data-driven loyalty campaigns** have realized an average 3.8% like-for-like sales uplift. (Score = 0.0) |
| Personalized promotions can deliver returns three times higher than mass promotions, yet many retailers allocate under 5% of their promo budgets to personalization. One major chain increased its personalized promo spend from 1% to 10% by establishing a "customer investment council," resulting in $250 million in incremental sales. | Retailers have seen a consistent 25% increase in revenue due to advanced **personalization capabilities**. (Score = 0.0) | External sources indicate **POS-enabled personalization** can lift revenue 5%-15% and advocate personalized e-receipts with relevant offers and coupons to extend post-purchase engagement. (Score = 0.0) |
| Retail media networks are expanding rapidly, with retail media growing at approximately 25% annually, offering retailers a profitable revenue stream to reinvest in technology, data, and personnel. By integrating loyalty data, retailers like Sephora, which links 95% of transactions to loyalty accounts, enhance precision in product recommendations and provide a seamless omnichannel experience, boosting conversion rates and customer lifetime value. | 85% of transactions are linked to **loyalty accounts** at Lee's Market, **providing a solid foundation for personalized marketing and improving customer engagement**. (Score = 0.0) | As of Q2 2024, 85% of transactions were linked to **loyalty accounts**, leaving a 15% unlinked identity gap. (Score = 0.0) |

Table 13: Factuality Performance Comparison: Llama 3.1 405B vs GPT-5 (Task DR0002). We show the number of factual and unfactual claims made by each model, as well as the overall factuality score for the given task.

| Metric | Llama 3.1 405B | GPT-5 | Improvement |
|---|---|---|---|
| Factuality Score | 0.41 | 0.65 | **+0.24** |
| Factual Claims | 7 | 13 | **+6 claims** |
| Unfactual Claims | 10 | 7 | **-3 claims** |

---

**⚡ Report Synthesis**

```
As an expert research analyst, synthesize the following content into
a coherent, insightful, and well-supported analysis for the theme:
"{theme}" directly related to the overarching research question:
"{original_question}"

Source Priority Guidelines:

1. **"internal"**:  Highest priority (internal company
documents, proprietary files, confidential reports,
enterprise chat messages, local documents, CRM data,
internal APIs, project management tools).  Insights from
these sources should form the primary foundation of the
analysis.
2. **"external"**:  Medium priority (public web sources,
academic papers, industry reports, news articles).  Use these
to provide broader context, external validation, or contrasting
perspectives.

**Synthesis Requirements:**
* **QUANTITATIVE PRIORITY:** Lead with numerical data, calculations, and
aggregations
* Extract ALL percentages, costs, metrics, and performance
data
* Perform mathematical operations:  aggregate percentages, calculate
increases, sum totals
* Example:  "Finance customers (35%) combined with healthcare
(40%) represent 75% of regulated industry concerns
[DOC:doc_1][DOC:doc_2]"

* **FACT VERIFICATION (CRITICAL):**
* ONLY state what documents explicitly contain - no inference or
extrapolation
* Use exact quotes for key numerical claims:  "As stated in the
document:  '[exact quote]' [DOC:doc_id]"

* **Citation Usage (Critical):**
* **Format:** Reference sources by their document ID: "Internal review
shows 15% increase [DOC:doc_079c2e0f_1752503636]"
* **NEVER HALLUCINATE CITATIONS:** Only use provided doc_id
values
* **Cite every numerical claim and calculation with source
documents**

Generate 2-4 paragraphs of synthesized analysis with proper inline
citations.
```

Prompt 5: Report Section Synthesis with Citation Requirements

executed by instantiating a fresh container to avoid state leakage across runs. We capped the number of agent iterations according to the settings described in Section 5, with each iteration limited by a fixed computational budget.

For model outputs, we standardized all prompts and evaluation pipelines across backbones, using identical research questions, company contexts, and injected insight sets. To avoid stochastic variability, we repeated generation three times per task and reported averaged scores.

Finally, all supporting scripts, environment configurations, and evaluation code are fully containerized, enabling consistent replication of our reported results across hardware setups.

Table 14: Factuality Content Analysis (Task DR0002). We show the number of factual and unfactual claims made by each model, highlighting the factuality accuracy of each model for the given task.

| Agent | Factual | Unfactual | Accuracy |
|---|---|---|---|
| Llama-3.1-405B | 7 claims | 10 claims | 41.0% |
| GPT-5 | 13 claim | 7 claims | 65.0% |
| **Change** | **+6** | **-3** | **+24.0%** |

Table 15: Factuality Summary (Task DR0002). We summarize the factuality result improvements made by using GPT-5 over Llama 3.1 405B.

| Impact Category | Value |
|---|---|
| Factuality Improvement | +24.0 percentage points |
| Claims Added | 3 total claims |
| Accuracy Enhancement | From 41.0% to 65.0% |
| Content Quality | More grounded information |
| Task Domain | Sales |
| Task Industry | Retail |

## I    DATA SYNTHESIS AND DRBA COST DETAILS

To generate the DRBench tasks, we combined external insight extraction with internal file synthesis. Each task included an average of 10 supporting files spanning heterogeneous formats (PDF, DOCX, PPTX, XLSX, and JSONL), with each file containing roughly 5 paragraphs of content. Files were designed to embed injected insights while mixing in distractor material, ensuring realistic enterprise complexity without exceeding practical runtime or storage budgets.

Data synthesis was primarily powered by GPT-4o. During task construction, GPT-4o was responsible for (1) extracting structured insights from public web sources, (2) adapting these insights into enterprise-grounded interpretations, (3) generating persona-specific deep research questions, and (4) producing file-level content with a balanced mix of insights and distractors. For evaluation, the DRBench Agent (DRBA) used GPT-4o as its backbone, with each task typically requiring 15 iterations and approximately 120–150 model calls.

In terms of cost, GPT-4o-based synthesis of a single task (10 files, 5 paragraphs each, plus metadata) consumed about 30k–40k tokens, while DRBA execution required an additional 50k–70k tokens per task. At current GPT-4o API pricing ($5 per million input tokens and $15 per million output tokens), this corresponds to a per-task cost of approximately $1.5–$3.5 depending on the mix of input/output tokens and the iteration budget. This makes large-scale benchmarking feasible at moderate cost, while still being significantly cheaper than manual authoring or annotation.

We also note that smaller open-source models such as Llama-3.1-8B-Instruct perform well for file generation. Unlike GPT-4o, which requires API usage, Llama-3.1-8B can be hosted locally and runs efficiently on a single NVIDIA A100 40GB GPU. This provides a cost-effective alternative for generating large numbers of supporting documents, especially when full closed-source quality is not required.

## J    WEB AGENTS FOR DEEP RESEARCH TASKS

Since each of the environments can be access directly through a web user interface (UI), we also experimented with an agent that can directly interact with the webpages through common browser actions like `click`, `input` and `scroll`, which are executed through *playwright*[3]. We implement our web agent using the AgentLab and BrowserGym frameworks (Chezelles et al., 2025) with a GPT-4.1[4] backbone. Our agent is implemented from AgentLab's *GenericAgent*, which achieves respectable performance when used

---

[3]https://playwright.dev
[4]https://openai.com/index/gpt-4-1/

with GPT-4o[5] as a backbone; it completes 45.5% of the tasks in WorkArena (Drouin et al., 2024), 31.4% in WebArena (Zhou et al., 2024) and achieves a step-level reward of 13.7% on WebLINX (Lù et al., 2024).

**Hyperparameters and Prompt**    We present the hyperparameters for the agent in tables 16 and 17, which are in majority set to the default hyperparameters, except for the maximum number of input tokens (bound to a reasonable maximum length) and a higher maximum number of steps (to allow the agent to perform more actions required to write the report). We further update the agent's action space on the last step to only allow it to reply to the user with a report, ensuring that each trajectory terminates with a report. To ensure that the agent is aware of the tools it can use, we modify the default system prompt (see prompt 6). Additionally, each task intent is provided alongside information about the user and company (see prompt 7).

**Results**    We find that the GPT-4.1-powered web agent achieves an insights recall and factuality of 1.11% and 6.67% respectively and a report quality score of 33.07%. Although the high report quality indicates that the agent can properly formulate a report, the insights quality is severely limited, with none of the claims being backed by useful sources. For example, a DRBench Agent powered by GPT-5 may answer the question *What is Lee's Market's current food waste reduction rate as of Q2 2024?* with *An 8% reduction in food waste in Q2 2024 saved Lee's Market $1.2 million, indicating that better inventory control can yield both safety and financial benefits.*, which achieves a score of 1.0 for the question. On the other hand, a GPT-4.1-powered agent will provide an unsatisfactory answer, thus achieving an insights recall of 0.0. The most likely cause of this poor performance is the model's limited capability to properly interact with web interfaces when encountering unfamiliar tools. For instance, the agent may be unfamiliar with the VNC and file browser applications, making it harder for it to correctly select the file it needs to use. Moreover, whenever the agent ends up performing an ineffective action (e.g. click on an element that does not trigger any change to the page), it tends to persist by reiterating the same action (see Table 18), or the same sequence of ineffective actions, despite not achieving anything in the previous steps. As a result, despite a large number of steps, most of the agent's actions are not helpful towards solving the task.

Table 16: Web Agents Boolean Hyperparameters

| Value | Flags |
|-------|-------|
| **True** | `vision_support, use_ax_tree, use_tabs, use_focused_element, use_error_logs, use_history, use_action_history, use_screenshot, use_som, extract_visible_tag, extract_clickable_tag, use_thinking, use_concrete_example, use_abstract_example, use_hints, be_cautious, add_missparsed_messages` |
| **False** | `use_html, use_past_error_logs, use_think_history, use_diff, filter_visible_elements_only, filter_with_bid_only, filter_som_only, multiaction, strict, long_description, individual_examples, use_plan, use_criticise, use_memory, enable_chat` |

## K    STANDARD ERROR

Restricting to the MinEval subset, we average the results on each task across 3 different runs in Table 19. We give both the means and standard errors for the insight recall, factuality, distractor avoidance, and report quality.

## L    EVALUATING LLM JUDGE SENSITIVITY ACROSS MODEL TYPES

Across models, replacing the GPT-4o judge with Llama-3.1-405B yields only minor differences in evaluation outcomes (Table 20). This stability arises from the design of our evaluation protocol: almost all metrics, insight recall, factuality, and distractor avoidance—are computed through a claim-based marking strategy after breaking the model output into short, atomic statements. Large language models are highly

---

[5]https://openai.com/index/hello-gpt-4o/

⚡ **Web Agents System Prompt**

```
You are an agent trying to solve a web task based on the content of the
page and user instructions.  You can interact with the page and explore,
and send messages to the user.  Each time you submit an action it will
be sent to the browser and you will receive a new page.
You will be solving tasks that involve Deep Research, by
navigating websites and web apps with information useful
solving the task.  You will need to gather insights from
data contained in the services provided and the internet
to complete your report.  You have a maximum of 50 steps
to complete the task.  Before the end, you must use the
action {send_msg_to_user}, which should contain a final Deep
Research report detailing everything the user needs to
know.

You must sustain your claims in files, chats, or emails
from the enterprise environment or in websites you
searched.  You must provide a citation (or an inline
citation, that works too) with the source of those claims
(e.g.  << add citation examples from the documentation
>>).  Do not make up citations if you haven't retrieved its
content.

Here are some expected agent behavior:

**Enterprise Environment Interaction:**
- Access Nextcloud files, Mattermost chats, emails, VNC desktop,
etc.
- Extract relevant information from multiple sources
- Navigate complex enterprise data landscapes

**Report Requirements:**
- Synthesize findings into comprehensive research
report
- Include proper citations for all claims (flexible format -
auto-normalized)
- Draw meaningful insights and conclusions
- Ground all statements in available evidence

**Citation Format (Flexible - Auto-Normalized):**
- Files:  `quarterly_report.pdf`, `shared/budget-analysis.xlsx`,
`Analysis document (reports/analysis.docx)`
- URLs:  Direct links or `[Article Title](https://example.com)`
- Emails:  `Email from alice@company.com on Jan 20,
2025`
- Chat:  `Mattermost message from john.doe in Compliance team, General
channel`

If you need navigate the internet (outside of the designated websites),
you can use the browser inside noVNC Desktop.
```

Prompt 6: Extended instructions given to the Deep Research web agent.

Table 17: Web Agents Non-Boolean Hyperparameters

| Parameter | Value |
|---|---|
| chat_model.model_name | gpt-4.1 |
| chat_model.max_total_tokens | 32768 |
| chat_model.max_input_tokens | 28672 |
| chat_model.max_new_tokens | 4096 |
| chat_model.temperature | 0 |
| action.action_set.subsets | webarena |
| action.action_set.retry_with_force | true |
| flags.max_prompt_tokens | 28672 |
| flags.max_trunc_itr | 20 |
| env.max_steps | 50 |

---

**⚡ Web Agents Task Intent Prompt**

```
How can Lee's Market leverage FSMA 204 regulations to enhance food
safety and customer trust?

Here is some information about the user:
ID: MNG0003
First name:  John
Last name:  Doe
...
Justification:  As a regulatory affairs manager, John
...

Here is some information about the company:
Name:  Lee's Market
Annual revenue:  $500M - $600M
...
Target markets:  Asian communities in the U.S. and ...
```

Prompt 7: Web Agents Task Intent Prompt

Table 18: Web Agents tends to get stuck on cycles of actions, and are unable to backtrack or to restart with a different application.

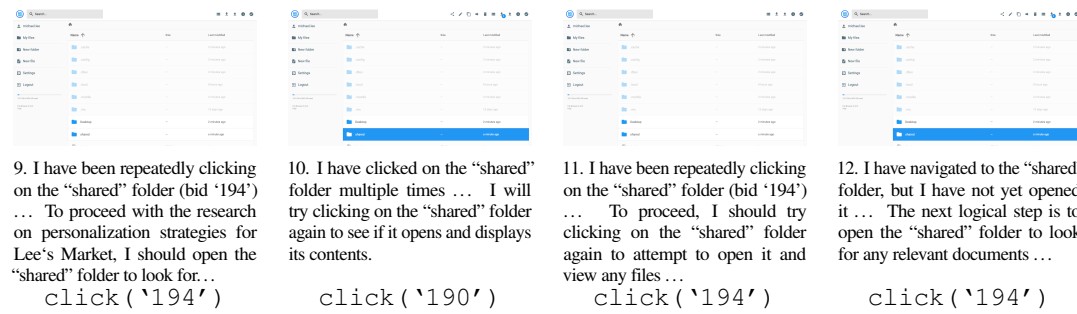

9. I have been repeatedly clicking on the "shared" folder (bid '194') … To proceed with the research on personalization strategies for Lee's Market, I should open the "shared" folder to look for…

click('194')

10. I have clicked on the "shared" folder multiple times … I will try clicking on the "shared" folder again to see if it opens and displays its contents.

click('190')

11. I have been repeatedly clicking on the "shared" folder (bid '194') … To proceed, I should try clicking on the "shared" folder again to attempt to open it and view any files …

click('194')

12. I have navigated to the "shared" folder, but I have not yet opened it … The next logical step is to open the "shared" folder to look for any relevant documents …

click('194')

consistent on such simple yes/no judgments, leading to minimal variance across judges. The only metric with larger fluctuations is report quality, which is the only non-binary, holistic score. Overall, these results indicate that our findings are robust to the choice of LLM judge.

Table 19: DRBA performance with different planning configurations on MinEval. We compare the base agent with variants using Simple Research Planning (SRP), Complex Research Planning (CRP), Adaptive Action Planning (AAP), and their combinations. Scores are reported for insight recall, factuality, distractor avoidance, report quality, and the overall harmonic mean.

| Configuration | Insight Recall (%) | Factuality (%) | Distractor Avoidance (%) | Report Quality (%) | Harmonic Mean (%) |
|---|---|---|---|---|---|
| Base DRBA | $18.8 \pm 2.2$ | $66.5 \pm 5.2$ | $98.1 \pm 0.6$ | $91.2 \pm 0.2$ | 44.7 |
| + SRP | $13.7 \pm 2.3$ | $62.2 \pm 6.3$ | $\mathbf{100.0 \pm 0.0}$ | $90.1 \pm 0.3$ | 36.3 |
| + CRP | $15.4 \pm 1.7$ | $\mathbf{70.5 \pm 2.9}$ | $\mathbf{100.0 \pm 0.0}$ | $91.7 \pm 0.2$ | 40.0 |
| + AAP | $\mathbf{19.7 \pm 1.7}$ | $60.4 \pm 4.6$ | $99.5 \pm 2.3$ | $\mathbf{92.8 \pm 0.3}$ | $\mathbf{45.4}$ |
| + SRP + AAP | $14.2 \pm 2.3$ | $50.4 \pm 5.8$ | $99.0 \pm 2.3$ | $90.6 \pm 0.4$ | 35.9 |
| + CRP + AAP | $18.8 \pm 2.9$ | $69.1 \pm 1.7$ | $\mathbf{100.0 \pm 0.0}$ | $92.3 \pm 0.3$ | 45.2 |

Table 20: Comparison of MinEval results when switching the LLM judge from GPT-4o to Llama-3.1-405B. Metrics show minimal variance in factuality and distractor avoidance, moderate variance in insight recall, and largest variance in report quality.

| Model | Judge | Insight Recall | Factuality | Distractor Avoidance | Report Quality |
|---|---|---|---|---|---|
| GPT-5 | GPT-4o | 39.63 | 65.17 | 92.86 | 93.42 |
| | Llama-3.1-405B | 38.92 | 65.11 | 92.40 | 91.83 |
| DeepSeek Chat 3.1 | GPT-4o | 30.26 | 70.27 | 96.67 | 86.88 |
| | Llama-3.1-405B | 29.54 | 69.98 | 96.22 | 85.10 |
| Qwen 2.5 72B | GPT-4o | 26.82 | 58.35 | 97.65 | 89.64 |
| | Llama-3.1-405B | 26.11 | 58.12 | 97.20 | 87.45 |
| GPT-4o | GPT-4o | 17.31 | 60.84 | 98.33 | 91.62 |
| | Llama-3.1-405B | 16.85 | 60.31 | 98.05 | 89.82 |
| Llama 3.1 405B | GPT-4o | 20.16 | 69.75 | 97.90 | 91.26 |
| | Llama-3.1-405B | 19.72 | 69.42 | 97.55 | 89.51 |

## M    COMPLEX MODEL ABLATION RESULTS

Extending our discussion in Section 5.3 to more GPT and Llama models in Table 21, we see that the smaller GPT-5-mini model lags behind but still outperforms earlier closed-source backbones such as GPT-4o and GPT-4o-mini, particularly in terms of harmonic mean. In addition, smaller variants of Llama degrade metric results further. This further substantiates the claim that larger and more advanced models tend to offer a better balance between recall, factuality, and overall report quality.

## N    QUANTITATIVE RESULTS PER TASK

We show a detailed breakdown of the insights recall in Table 23, factuality in Table 24, distractor avoidance in Table 25 and report quality in Table 26 on the MinEval subset for a variety of models.

## O    EFFECT OF NUMBER OF ITERATIONS

We next analyze the effect of varying the number of research loop iterations when using DRBA with GPT-5 as the backbone language model. Results for both the baseline configuration without explicit planning and the complex planning setup are shown in Table 27. Overall, increasing the iteration budget does not guarantee consistent improvements. With no planning, performance initially drops when the agent executes more iterations, as additional exploration often introduces noise and distracts from key insights. However, with a larger budget the agent partially recovers, suggesting that a small number of additional iterations can help refine factual grounding, while excessive exploration reduces focus.

For the complex planning setting, higher iterations improve certain metrics such as factuality, but this comes at the cost of lower insight recall and reduced overall balance. This indicates that while more steps allow the agent to verify citations more carefully, they can also lead to fragmented reasoning and overfitting to peripheral evidence. The best overall performance emerges at moderate iteration counts, highlighting the importance of carefully tuning the iteration budget rather than simply scaling up the number of steps.

Table 21: Performance of DRBA on the FullBenchmark subset using different backbone language models and planning strategies. Scores are reported for insight recall, factuality, distractor avoidance, report quality, and harmonic mean. Note that higher numbers corresponds to better scores, and the best result on each metric is bolded.

| Model | Planning | Insight Recall | Factuality | Distractor Avoidance | Report Quality | Harmonic Mean |
|---|---|---|---|---|---|---|
| GPT-5 | None | 36.52 | **72.11** | 93.22 | 93.41 | **63.81** |
| GPT-5 | SRP | 35.41 | 69.42 | 94.67 | **93.88** | 62.64 |
| GPT-5 | CRP | **37.48** | 62.33 | 91.71 | 92.03 | 62.02 |
| GPT-5-mini | None | 22.37 | 54.06 | 94.93 | 81.24 | 46.49 |
| GPT-5-mini | SRP | 24.58 | 54.31 | 94.18 | 82.43 | 48.87 |
| GPT-5-mini | CRP | 22.96 | 47.19 | 93.06 | 82.71 | 45.67 |
| GPT-4o | None | 16.23 | 61.44 | 95.86 | 89.94 | 40.22 |
| GPT-4o | SRP | 18.81 | 58.74 | 95.37 | 90.58 | 43.61 |
| GPT-4o | CRP | 16.04 | 58.61 | 95.63 | 89.22 | 39.58 |
| GPT-4o-mini | None | 11.86 | 42.24 | 95.79 | 81.08 | 30.59 |
| GPT-4o-mini | SRP | 11.78 | 50.13 | 94.92 | 81.97 | 31.35 |
| GPT-4o-mini | CRP | 11.31 | 43.97 | 94.56 | 82.16 | 29.87 |
| GPT-OSS-120B | None | 19.26 | 26.14 | 94.78 | 80.93 | 35.37 |
| GPT-OSS-120B | SRP | 15.17 | 24.54 | 94.66 | 80.98 | 30.87 |
| GPT-OSS-120B | CRP | 16.04 | 33.02 | 95.18 | 81.81 | 34.67 |
| Llama-3.1-405B-Instruct | None | 16.1 | 69.3 | 95.2 | 88.6 | 40.68 |
| Llama-3.1-405B-Instruct | SRP | 15.7 | 70.4 | 94.6 | 89.7 | 40.15 |
| Llama-3.1-405B-Instruct | CRP | 18.33 | 65.72 | 95.04 | 89.01 | 43.70 |
| Llama-3.1-70b-Instruct | None | 16.03 | 55.04 | 95.64 | 81.96 | 38.76 |
| Llama-3.1-70b-Instruct | SRP | 14.22 | 58.66 | **95.93** | 81.57 | 36.35 |
| Llama-3.1-70b-Instruct | CRP | 14.71 | 49.91 | 95.22 | 84.38 | 36.24 |
| DeepSeek-V3.1 | None | 22.6 | 68.4 | 94.9 | 84.1 | 49.20 |
| DeepSeek-V3.1 | SRP | 23.1 | 69.2 | 94.1 | 84.9 | 49.91 |
| DeepSeek-V3.1 | CRP | 28.21 | 67.09 | 93.96 | 85.57 | 55.03 |
| Qwen-2.5-72B-Instruct | None | 22.8 | 63.2 | 95.4 | 86.9 | 48.98 |
| Qwen-2.5-72B-Instruct | SRP | 20.9 | 61.1 | 95.1 | 85.2 | 46.26 |
| Qwen-2.5-72B-Instruct | CRP | 24.39 | 55.74 | 95.12 | 87.51 | 49.46 |

Table 22: Total average Insight Recall scores per model and insight source type computed on all the results available for each model running in Complex Research Plan mode on the FullBenchmark. Insights embedded in enterprise sources are more easily retrieved by DRBA in all the models.

| Model | Enterprise Fact | External Fact |
|---|---|---|
| GPT-5 | 0.597 | 0.0 |
| DeepSeek-V3.1 | 0.472 | 0.0 |
| GPT-5-mini | 0.444 | 0.0 |
| Qwen-2.5-72B-Instruct | 0.417 | 0.0 |
| Llama-3.1-405B-Instruct | 0.347 | 0.0 |
| GPT-OSS-120B | 0.333 | 0.0 |
| GPT-4o-mini | 0.194 | 0.0 |
| Llama-3.1-70B-Instruct | 0.194 | 0.0 |
| GPT-4o | 0.182 | 0.0 |

# P   DATA GENERATION PROMPTS

## P.1   COMPANY AND PERSONA DATA GENERATION

In this section we give the prompts used for company generation 8 and persona generation 9.

## P.2   QUESTION GENERATION

In this section we give the prompt used to generate our deep research questions 10.

Table 23: Mean and standard error of the insight recall metric for the first five tasks, obtained from three runs of on *DRBench* our agent (DRBA) using 15 iterations across different backbone models.

| Configuration | Plan | DR0001 | DR0002 | DR0003 | DR0004 | DR0005 |
|---|---|---|---|---|---|---|
| GPT-5 | - | .222 ± .056 | .429 ± .000 | .467 ± .033 | .519 ± .098 | .281 ± .035 |
| GPT-5 | SRP | .222 ± .056 | .381 ± .048 | .533 ± .033 | .370 ± .098 | .386 ± .046 |
| GPT-5 | CRP | .278 ± .056 | .381 ± .126 | .567 ± .067 | .370 ± .037 | .386 ± .046 |
| GPT-5-mini | - | .111 ± .056 | .286 ± .000 | .367 ± .033 | .222 ± .064 | .298 ± .018 |
| GPT-5-mini | SRP | .222 ± .056 | .286 ± .082 | .333 ± .033 | .296 ± .074 | .281 ± .063 |
| GPT-5-mini | CRP | .000 ± .000 | .381 ± .126 | .367 ± .067 | .370 ± .098 | .211 ± .053 |
| GPT-4o | - | .111 ± .056 | .238 ± .048 | .167 ± .033 | .185 ± .074 | .175 ± .018 |
| GPT-4o | SRP | .167 ± .000 | .333 ± .048 | .300 ± .000 | .148 ± .037 | .070 ± .018 |
| GPT-4o | CRP | .111 ± .056 | .238 ± .095 | .300 ± .100 | .111 ± .000 | .105 ± .000 |
| GPT-4o-mini | - | .000 ± .000 | .095 ± .048 | .267 ± .033 | .185 ± .037 | .140 ± .035 |
| GPT-4o-mini | SRP | .056 ± .056 | .190 ± .048 | .167 ± .067 | .148 ± .074 | .123 ± .018 |
| GPT-4o-mini | CRP | .000 ± .000 | .190 ± .048 | .267 ± .033 | .074 ± .037 | .123 ± .018 |
| GPT-OSS-120B | - | .111 ± .111 | .143 ± .000 | .433 ± .033 | .222 ± .000 | .211 ± .030 |
| GPT-OSS-120B | SRP | .056 ± .056 | .143 ± .000 | .367 ± .033 | .148 ± .098 | .158 ± .061 |
| GPT-OSS-120B | CRP | .000 ± .000 | .190 ± .048 | .333 ± .067 | .111 ± .064 | .281 ± .063 |
| Llama-3.1-405B-Instruct | - | .167 ± .000 | .143 ± .000 | .233 ± .088 | .185 ± .074 | .140 ± .035 |
| Llama-3.1-405B-Instruct | SRP | .222 ± .056 | .190 ± .048 | .167 ± .033 | .111 ± .064 | .158 ± .053 |
| Llama-3.1-405B-Instruct | CRP | .111 ± .056 | .238 ± .048 | .300 ± .058 | .148 ± .098 | .211 ± .030 |
| Llama-3.1-70B-Instruct | - | .167 ± .000 | .381 ± .048 | .200 ± .000 | .074 ± .037 | .105 ± .030 |
| Llama-3.1-70B-Instruct | SRP | .056 ± .056 | .286 ± .082 | .200 ± .058 | .185 ± .098 | .088 ± .046 |
| Llama-3.1-70B-Instruct | CRP | .167 ± .000 | .238 ± .048 | .267 ± .033 | .074 ± .074 | .105 ± .000 |
| DeepSeek-V3.1 | - | .167 ± .000 | .286 ± .082 | .300 ± .058 | .259 ± .037 | .246 ± .046 |
| DeepSeek-V3.1 | SRP | .278 ± .056 | .238 ± .048 | .333 ± .033 | .148 ± .074 | .281 ± .046 |
| DeepSeek-V3.1 | CRP | .167 ± .000 | .095 ± .048 | .600 ± .058 | .370 ± .037 | .281 ± .035 |
| Qwen-2.5-72B-Instruct | - | .222 ± .056 | .238 ± .048 | .400 ± .058 | .259 ± .037 | .158 ± .061 |
| Qwen-2.5-72B-Instruct | SRP | .111 ± .056 | .333 ± .048 | .333 ± .088 | .259 ± .037 | .123 ± .018 |
| Qwen-2.5-72B-Instruct | CRP | .167 ± .096 | .143 ± .082 | .400 ± .058 | .333 ± .000 | .298 ± .076 |

⚡ **Company Generation Prompt**

```
Generate a realistic company structure for {company_name} in the
{industry} industry.

Company size: {size} ({employee_range} employees)

The company should focus on this domain {domain}

EXTERNAL INSIGHTS: {external_insights}

Return ONLY a valid JSON object with this structure:
{output_structure}

Make it realistic for the {industry} industry.
```

Prompt 8: Company Generation Prompt Template.

Table 24: Mean and standard error of the factuality metric for the first five tasks, obtained from our agent (DRBA) using 15 iterations across different backbone models.

| Configuration | Plan | DR0001 | DR0002 | DR0003 | DR0004 | DR0005 |
|---|---|---|---|---|---|---|
| GPT-5 | - | .761 ± .072 | .504 ± .075 | .866 ± .007 | .833 ± .019 | .762 ± .077 |
| GPT-5 | SRP | .714 ± .050 | .384 ± .076 | .848 ± .034 | .812 ± .070 | .846 ± .029 |
| GPT-5 | CRP | .730 ± .060 | .291 ± .094 | .782 ± .012 | .782 ± .064 | .674 ± .121 |
| GPT-5-mini | - | .585 ± .045 | .297 ± .119 | .705 ± .039 | .704 ± .067 | .647 ± .098 |
| GPT-5-mini | SRP | .647 ± .120 | .299 ± .056 | .624 ± .041 | .694 ± .028 | .683 ± .020 |
| GPT-5-mini | CRP | .309 ± .126 | .381 ± .161 | .699 ± .047 | .692 ± .111 | .472 ± .114 |
| GPT-4o | - | .792 ± .150 | .490 ± .110 | .827 ± .056 | .570 ± .058 | .593 ± .204 |
| GPT-4o | SRP | .485 ± .262 | .512 ± .131 | .813 ± .041 | .693 ± .139 | .614 ± .121 |
| GPT-4o-mini | SRP | .475 ± .166 | .653 ± .097 | .704 ± .037 | .542 ± .110 | .406 ± .020 |
| GPT-4o | CRP | .828 ± .043 | .265 ± .133 | .800 ± .000 | .690 ± .128 | .459 ± .235 |
| GPT-4o-mini | - | .611 ± .056 | .429 ± .092 | .622 ± .062 | .481 ± .209 | .191 ± .046 |
| GPT-4o-mini | CRP | .557 ± .030 | .324 ± .169 | .580 ± .075 | .642 ± .119 | .344 ± .144 |
| GPT-OSS-120B | - | .144 ± .099 | .150 ± .035 | .386 ± .040 | .337 ± .117 | .445 ± .051 |
| GPT-OSS-120B | SRP | .074 ± .074 | .128 ± .072 | .410 ± .090 | .311 ± .155 | .451 ± .080 |
| GPT-OSS-120B | CRP | .368 ± .061 | .178 ± .078 | .564 ± .064 | .400 ± .076 | .435 ± .190 |
| Llama-3.1-405B-Instruct | - | .852 ± .087 | .726 ± .158 | .803 ± .028 | .820 ± .066 | .745 ± .022 |
| Llama-3.1-405B-Instruct | SRP | .802 ± .125 | .638 ± .202 | .800 ± .074 | .892 ± .035 | .832 ± .083 |
| Llama-3.1-405B-Instruct | CRP | .789 ± .053 | .392 ± .154 | .792 ± .055 | .771 ± .073 | .745 ± .100 |
| Llama-3.1-70B-Instruct | - | .618 ± .109 | .431 ± .160 | .684 ± .104 | .812 ± .021 | .687 ± .073 |
| Llama-3.1-70B-Instruct | SRP | .608 ± .173 | .681 ± .069 | .800 ± .069 | .826 ± .067 | .557 ± .143 |
| Llama-3.1-70B-Instruct | CRP | .588 ± .082 | .286 ± .108 | .686 ± .011 | .522 ± .270 | .559 ± .240 |
| DeepSeek-V3.1 | - | .860 ± .014 | .518 ± .085 | .818 ± .041 | .679 ± .095 | .757 ± .057 |
| DeepSeek-V3.1 | SRP | .696 ± .041 | .531 ± .086 | .922 ± .056 | .769 ± .035 | .754 ± .082 |
| DeepSeek-V3.1 | CRP | .581 ± .042 | .657 ± .024 | .838 ± .050 | .774 ± .053 | .662 ± .098 |
| Qwen-2.5-72B-Instruct | - | .674 ± .077 | .493 ± .109 | .866 ± .002 | .741 ± .060 | .696 ± .060 |
| Qwen-2.5-72B-Instruct | SRP | .806 ± .049 | .540 ± .174 | .741 ± .074 | .724 ± .101 | .550 ± .148 |
| Qwen-2.5-72B-Instruct | CRP | .626 ± .114 | .396 ± .056 | .723 ± .053 | .587 ± .139 | .586 ± .055 |

---

**⚡ Persona Generation Prompt**

```
Generate {persona_count} diverse employee personas for {company_name} in
the {industry} industry.

The personas should focus on this domain:  {domain}

Create diverse roles across seniority levels:  Junior, Mid, Senior,
Executive

Return ONLY a valid JSON array with this exact format:
{output_structure}

Make personas realistic with appropriate responsibilities for their
roles.
```

Prompt 9: Persona Generation Prompt.

## P.3 PUBLIC SOURCE AND INSIGHT COLLECTION

In this section we give the prompt used to generate external insights 11. The URLs used for external insight extraction and deep research question creation can be found in Table 28.

Table 25: Mean and standard error of the distractor avoidance metric for the first five tasks, obtained from three runs of on *DRBench* our agent (DRBA) using 15 iterations across different backbone models.

| Configuration | Plan | DR0001 | DR0002 | DR0003 | DR0004 | DR0005 |
|---|---|---|---|---|---|---|
| GPT-5 | - | .857 ± .000 | .900 ± .058 | 1.00 ± .000 | 1.00 ± .000 | 1.00 ± .000 |
| GPT-5 | SRP | .857 ± .000 | 1.00 ± .000 | 1.00 ± .000 | 1.00 ± .000 | 1.00 ± .000 |
| GPT-5 | CRP | .905 ± .048 | .900 ± .058 | .933 ± .000 | .905 ± .024 | 1.00 ± .000 |
| GPT-5-mini | - | .905 ± .048 | .967 ± .033 | .978 ± .022 | .976 ± .024 | 1.00 ± .000 |
| GPT-5-mini | SRP | .857 ± .000 | .933 ± .033 | 1.00 ± .000 | 1.00 ± .000 | 1.00 ± .000 |
| GPT-5-mini | CRP | .857 ± .000 | .867 ± .133 | 1.00 ± .000 | 1.00 ± .000 | 1.00 ± .000 |
| GPT-4o | - | .952 ± .048 | 1.00 ± .000 | 1.00 ± .000 | 1.00 ± .000 | 1.00 ± .000 |
| GPT-4o | SRP | .952 ± .048 | 1.00 ± .000 | 1.00 ± .000 | .976 ± .024 | 1.00 ± .000 |
| GPT-4o | CRP | .952 ± .048 | 1.00 ± .000 | 1.00 ± .000 | .964 ± .036 | 1.00 ± .000 |
| GPT-4o-mini | - | .952 ± .048 | 1.00 ± .000 | 1.00 ± .000 | 1.00 ± .000 | 1.00 ± .000 |
| GPT-4o-mini | SRP | .857 ± .000 | 1.00 ± .000 | 1.00 ± .000 | 1.00 ± .000 | 1.00 ± .000 |
| GPT-4o-mini | CRP | .857 ± .000 | 1.00 ± .000 | 1.00 ± .000 | 1.00 ± .000 | 1.00 ± .000 |
| GPT-OSS-120B | - | .857 ± .000 | 1.00 ± .000 | 1.00 ± .000 | 1.00 ± .000 | 1.00 ± .000 |
| GPT-OSS-120B | SRP | .857 ± .000 | 1.00 ± .000 | 1.00 ± .000 | 1.00 ± .000 | 1.00 ± .000 |
| GPT-OSS-120B | CRP | .905 ± .048 | 1.00 ± .000 | 1.00 ± .000 | 1.00 ± .000 | 1.00 ± .000 |
| Llama-3.1-405B-Instruct | - | 1.00 ± .000 | 1.00 ± .000 | 1.00 ± .000 | 1.00 ± .000 | 1.00 ± .000 |
| Llama-3.1-405B-Instruct | SRP | .905 ± .048 | 1.00 ± .000 | 1.00 ± .000 | 1.00 ± .000 | 1.00 ± .000 |
| Llama-3.1-405B-Instruct | CRP | .952 ± .048 | .967 ± .033 | 1.00 ± .000 | .976 ± .024 | 1.00 ± .000 |
| Llama-3.1-70B-Instruct | - | .905 ± .048 | 1.00 ± .000 | .978 ± .022 | .952 ± .048 | 1.00 ± .000 |
| Llama-3.1-70B-Instruct | SRP | .905 ± .048 | 1.00 ± .000 | 1.00 ± .000 | .976 ± .024 | 1.00 ± .000 |
| Llama-3.1-70B-Instruct | CRP | .905 ± .048 | 1.00 ± .000 | 1.00 ± .000 | 1.00 ± .000 | 1.00 ± .000 |
| DeepSeek-V3.1 | - | .905 ± .048 | .967 ± .033 | 1.00 ± .000 | 1.00 ± .000 | 1.00 ± .000 |
| DeepSeek-V3.1 | SRP | .857 ± .000 | 1.00 ± .000 | 1.00 ± .000 | .976 ± .024 | 1.00 ± .000 |
| DeepSeek-V3.1 | CRP | .905 ± .048 | 1.00 ± .000 | 1.00 ± .000 | .929 ± .000 | 1.00 ± .000 |
| Qwen-2.5-72B-Instruct | - | 1.00 ± .000 | 1.00 ± .000 | 1.00 ± .000 | .905 ± .024 | 1.00 ± .000 |
| Qwen-2.5-72B-Instruct | SRP | .952 ± .048 | 1.00 ± .000 | 1.00 ± .000 | .952 ± .024 | 1.00 ± .000 |
| Qwen-2.5-72B-Instruct | CRP | .952 ± .048 | 1.00 ± .000 | .978 ± .022 | .952 ± .048 | 1.00 ± .000 |

## P.4 INTERNAL INSIGHT GENERATION

In this section we give the prompts to generate both internal insights (Prompt 12) and internal distractors (Prompt 13).

## P.5 FILE GENERATION

In this section we give the prompts used for generating each of the file types used in our tasks, which we list as follows:

- **PDF**: Prompts 18, 19, and 20

- **Excel**: Prompts 21, 22, and 23

- **Powerpoint**: Prompts 24, 25, and 26

- **Email**: Prompts 27, 28, and 29

- **Chat**: Prompts 30, 31, 29, and 33

Table 26: Mean and standard error of the report quality metric for the first five tasks, obtained from three runs of *DRBench* with 15 iterations across different backbone models.

| Configuration | Plan | DR0001 | DR0002 | DR0003 | DR0004 | DR0005 |
|---|---|---|---|---|---|---|
| GPT-5 | - | .936 ± .008 | .918 ± .004 | .924 ± .005 | .909 ± .001 | .927 ± .009 |
| GPT-5 | SRP | .942 ± .000 | .927 ± .008 | .936 ± .006 | .915 ± .002 | .933 ± .007 |
| GPT-5 | CRP | .948 ± .007 | .921 ± .001 | .940 ± .008 | .922 ± .009 | .929 ± .008 |
| GPT-5-mini | - | .892 ± .002 | .884 ± .007 | .879 ± .004 | .891 ± .000 | .886 ± .005 |
| GPT-5-mini | SRP | .901 ± .009 | .889 ± .002 | .887 ± .001 | .895 ± .008 | .892 ± .003 |
| GPT-5-mini | CRP | .896 ± .001 | .882 ± .006 | .884 ± .003 | .889 ± .009 | .890 ± .001 |
| GPT-4o | - | .927 ± .000 | .911 ± .003 | .903 ± .001 | .918 ± .009 | .909 ± .002 |
| GPT-4o | SRP | .934 ± .008 | .919 ± .000 | .911 ± .009 | .923 ± .001 | .916 ± .000 |
| GPT-4o | CRP | .929 ± .009 | .914 ± .002 | .905 ± .000 | .920 ± .008 | .913 ± .009 |
| GPT-4o-mini | - | .886 ± .004 | .874 ± .008 | .861 ± .007 | .872 ± .002 | .879 ± .006 |
| GPT-4o-mini | SRP | .893 ± .002 | .881 ± .005 | .867 ± .004 | .878 ± .001 | .884 ± .003 |
| GPT-4o-mini | CRP | .889 ± .003 | .877 ± .006 | .864 ± .005 | .875 ± .000 | .882 ± .004 |
| GPT-OSS-120B | - | .872 ± .007 | .861 ± .001 | .849 ± .009 | .858 ± .006 | .866 ± .008 |
| GPT-OSS-120B | SRP | .878 ± .006 | .867 ± .009 | .854 ± .008 | .863 ± .004 | .872 ± .007 |
| GPT-OSS-120B | CRP | .874 ± .007 | .863 ± .000 | .851 ± .008 | .860 ± .005 | .869 ± .006 |
| Llama-3.1-405B-Instruct | - | .914 ± .000 | .903 ± .003 | .897 ± .001 | .909 ± .008 | .902 ± .002 |
| Llama-3.1-405B-Instruct | SRP | .921 ± .008 | .910 ± .001 | .904 ± .000 | .915 ± .009 | .908 ± .000 |
| Llama-3.1-405B-Instruct | CRP | .917 ± .009 | .906 ± .002 | .899 ± .001 | .911 ± .008 | .905 ± .001 |
| Llama-3.1-70B-Instruct | - | .889 ± .003 | .877 ± .007 | .869 ± .005 | .881 ± .001 | .873 ± .004 |
| Llama-3.1-70B-Instruct | SRP | .895 ± .002 | .883 ± .005 | .874 ± .004 | .886 ± .000 | .878 ± .003 |
| Llama-3.1-70B-Instruct | CRP | .891 ± .003 | .879 ± .006 | .871 ± .005 | .883 ± .001 | .875 ± .004 |
| DeepSeek-V3.1 | - | .884 ± .004 | .872 ± .008 | .864 ± .006 | .876 ± .002 | .869 ± .005 |
| DeepSeek-V3.1 | SRP | .890 ± .003 | .878 ± .006 | .870 ± .005 | .881 ± .001 | .874 ± .004 |
| DeepSeek-V3.1 | CRP | .886 ± .004 | .874 ± .007 | .866 ± .006 | .878 ± .002 | .871 ± .005 |
| Qwen-2.5-72B-Instruct | - | .901 ± .001 | .889 ± .005 | .881 ± .002 | .893 ± .009 | .885 ± .003 |
| Qwen-2.5-72B-Instruct | SRP | .908 ± .000 | .896 ± .003 | .888 ± .001 | .899 ± .008 | .891 ± .002 |
| Qwen-2.5-72B-Instruct | CRP | .904 ± .001 | .892 ± .004 | .884 ± .002 | .895 ± .009 | .888 ± .003 |

Table 27: Effect of the number of research loop iterations on the performance of DRBA with GPT-5 as the backbone model, on MinEval.

| Planning Method | # Iterations | Insight Recall | Factuality | Distractor Avoidance | Report Quality | Harmonic Mean |
|---|---|---|---|---|---|---|
| No Planning | 15 | 39.45 | 72.65 | 93.14 | 94.56 | 66.20 |
| No Planning | 30 | 28.80 | 69.03 | 98.57 | **96.12** | 57.34 |
| No Planning | 50 | 37.10 | 78.84 | **100.00** | 93.48 | 66.30 |
| Complex Research Planning (CRP) | 15 | **44.44** | 62.51 | 90.95 | 93.12 | **66.41** |
| Complex Research Planning (CRP) | 30 | 31.61 | **73.94** | 94.38 | 94.76 | 60.32 |
| Complex Research Planning (CRP) | 50 | 38.16 | 66.05 | 94.38 | 92.64 | 63.76 |

## Q  EVALUATION PROMPTS

In this section we give the prompts for decomposing reports into atomic insights 14, computing insight recall 15, computing factuality 16, and computing report quality 17. These prompts are discussed in detail in Section 5.1.

## R  HUMAN PREFERENCE EVALUATION

We calculate a human score for model $a$ task $t$ as:

---

**⚡ Deep Research Question Generation Prompt**

```
Generate 3 Deep Research (DR) questions for the following business
context:

Persona: {persona_name} – {persona_role}
Department: {persona_department}
Responsibilities: {persona_responsibilities}
Company: {company_name} ({company_industry})
Domain: {domain}

External Insights: {external_insights}

Generate 3 Deep Research questions that:
1.  Are appropriate for the persona's role and department
2.  Require analysis of the provided internal insights
3.  Consider the external market context ...

Each question should be 1 sentence of 15 words max, in
plain english, and end with a question mark.  Do not
include any preamble or explanation – return only the JSON
array.

Return ONLY a valid JSON array with this structure:
{output_structure}
```

Prompt 10: Deep Research Question Generation Prompt Template. Subquestions are generated to help human annotators select good DR questions.

---

**⚡ External Insight Extraction Prompt**

```
You will be given a report (with url, and date).  Based
on the report, generate 3 external insights that summarize
important findings, trends, or takeaways from the
report.

Output Format
{output_structure}

Url: {url}
Industry: {industry}
Domain: {domain}
Company Information: {company_information}

Important notes
Focus only on insights that are external and grounded in the report.
Insights should be concise, factual, and directly tied to the retail
industry context.
```

Prompt 11: External Insight Extraction Prompt.

$$S_{a,t} = \frac{1}{n}\sum_{i=1}^{n} s_{a,i}, \text{ where } s_{a,i} = \begin{cases} 1, & \text{if human\_choice is } a \text{ or "both good"} \\ 0, & \text{otherwise} \end{cases}$$

, where $n$ is the number of groundtruth insights in task $t$, $s_{a,i}$ is the human score of model $a$ on insight $i$. Figure 8 shows that the insight recall metric is on par with human decision.

Table 28: Public URLs For Deep Research Task Creation.

| Industry | Domain | Reference |
|---|---|---|
| Retail | Compliance | Grocers on FSMA-204 Compliance (GroceryDive) ⬀ |
| Retail | CRM | Grocery Loyalty & Inflation (EagleEye) ⬀ |
| Retail | Market Analysis | Grocery Trends Outlook 2025 (GroceryDive) ⬀ |
| Retail | ITSM | Retail IT Optimization (Thirdera) ⬀ |
| Retail | CSM | Chatbots & Grocery Interactions (GroceryDoppio) ⬀ |
| Retail | Knowledge Mgmt | Retail Knowledge Management (Knowmax) ⬀ |
| Retail | Sales | Personalization in Action (BCG) ⬀ |
| Retail | Cybersecurity | Retail Cybersecurity Threats (VikingCloud) ⬀ |
| Retail | Public Relations | Walmart CSR Strategy (SunriseGeek) ⬀ |
| Healthcare | Compliance | Telehealth Regulations (HealthcareDive) ⬀ |
| Healthcare | CRM | Future of Healthcare CRM (WTT Solutions) ⬀ |
| Healthcare | Market Analysis | Future of Telehealth (CHG Healthcare) ⬀ |
| Healthcare | ITSM | Healthcare ITSM (Topdesk) ⬀ |
| Healthcare | CSM | Patient Engagement Tech (TechTarget) ⬀ |
| Healthcare | Knowledge Mgmt | Knowledge Mgmt in Healthcare (C8Health) ⬀ |
| Healthcare | Sales | Sales for Digital Health (Medium) ⬀ |
| Healthcare | Marketing | Marketing Telehealth Services (MarketingInsider) ⬀ |
| Healthcare | Cybersecurity | Healthcare Cybersecurity 2024 (AHA) ⬀ |
| Automobiles | Compliance | Evolving EV Regulations (WardsAuto) ⬀ |
| Automobiles | CRM | Salesforce Automotive Cloud (TechTarget) ⬀ |
| Automobiles | CSM | EV Aftersales Support (EVReport) ⬀ |
| Automobiles | Sales | Tesla vs Dealerships (TheWeek) ⬀ |
| Automobiles | Research | AI for EV Optimization (Here.com) ⬀ |
| Automobiles | Cybersecurity | Cybersecurity Risks in Cars (HelpNetSecurity) ⬀ |
| Automobiles | Quality Assurance | EV Quality Issues (GreenCars) ⬀ |
| Automobiles | Asset Mgmt | Digital Twins in Autos (RTInsights) ⬀ |
| Automobiles | Market Analysis | Global EV Outlook 2024 (IEA) ⬀ |

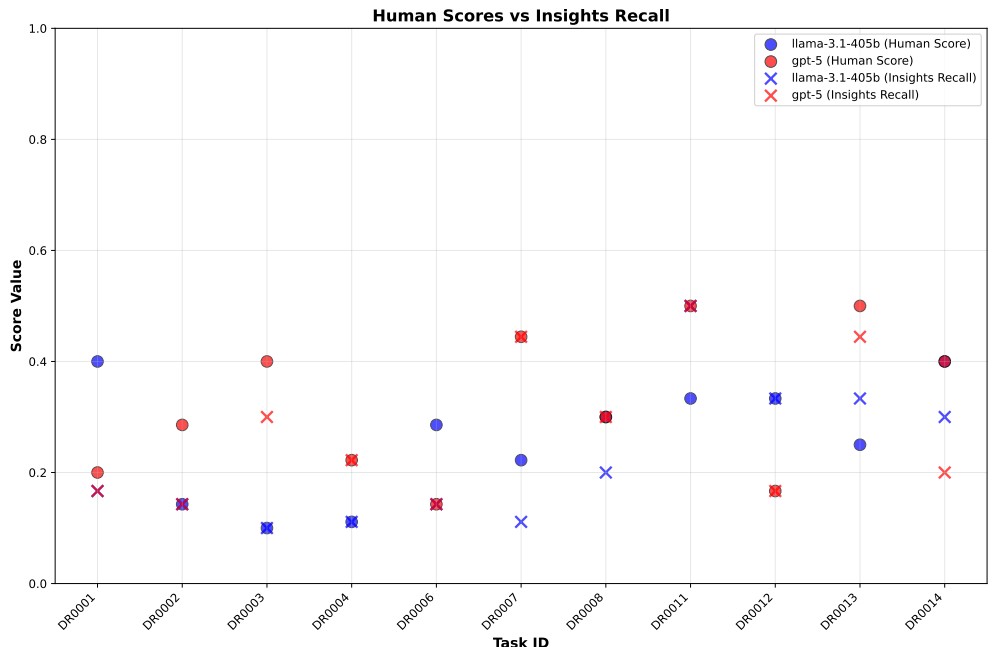

Figure 8: Comparison of Human Scores and Insight Recall Scores. As can be seen the human evaluation results are aligned with our automated evaluation.

> **⚡ Internal Supporting Insight Generation Prompt**
>
> ```
> Based on the Deep Research (DR) Question, external
> market insights, company context, and previous internal
> insights, generate 3 specific QA pair that an expert data
> scientist would need to get in order to address the DR
> Question.
>
> Company: {company_name} – {company_description}
> Industry: {industry}
> Company Size: {company_size} ({employee_count})
> Annual Revenue: {annual_revenue}
> Persona Context: {persona_context}
> DR Question: {dr_question}
> External Market Context (for inspiration): {external_context}
> QA History: {qa_list}
>
> Insight Requirements:
> – Use the DR Question as the central theme for the
> answer.
> – Draw inspiration and supporting details from the other internal
> insights and external insights provided.
> – Include QUANTITATIVE DATA in the answer: metrics, percentages, dollar
> amounts, timeframes, KPIs.
>
> Specific Question Instructions
> – the specific question should be a question that would be a step
> towards resolving the DR Question
> {additional_question_instructions}
>
> Answer Instructions
> – the answer should be 12 words max and minimum 5
> words
> {additional_answer_instructions}
>
> Justification Instructions
> – the justification should directly explain how the specific_question,
> answer pair help address the DR Question in 15 words
> max
>
> Misc Instructions
> – the filename should be 3 words max with dashes
> in between, do not mention the file type in the
> filename
> – use the example below as inspiration but do not use it
> directly
>
> Return ONLY a valid JSON object with this exact structure:
> {output_structure}
> ```

Prompt 12: Internal Supporting Insight Generation Prompt Template. *specific_questions* and *justification* help human annotators to select good insights.

## S    ROBUSTNESS OF INSIGHT RECALL TO PARAPHRASING

A potential concern is whether our Insight Recall metric is overly sensitive to wording differences and fails to recognize paraphrased or partially reworded insights. Our evaluation protocol, however, is designed to focus on the presence of key factual elements rather than exact lexical similarity. The LLM-as-a-judge prompt explicitly assesses whether the *core facts* of an insight appear in the predicted report, independent of phrasing.

⚡ **Internal Distractor Insight Generation Prompt**

```
Based on the Deep Research (DR) Question, external
market insights, company context, and previous internal
insights, generate 3 specific QA pairs that are DISTRACTOR
questions
- these should be questions that an expert data scientist might
ask about the company but are NOT relevant to addressing the DR
Question.

Company: {company_name} - {company_description}
Industry: {industry}
Company Size: {company_size} ({employee_count})
Annual Revenue: {annual_revenue}
Persona Context: {persona_context}
DR Question: {dr_question}
External Market Context (for inspiration): {external_context}
QA History: {qa_list}

DISTRACTOR Requirements:
- Generate questions that are plausible for this
company and industry but DO NOT help address the DR
Question.
- The questions should be about the company's operations, metrics, or
business but tangential to the DR Question.
- Include QUANTITATIVE DATA in the answer: metrics, percentages, dollar
amounts, timeframes, KPIs.
- Focus on different business areas that are NOT central to the
DR Question (e.g. if DR Question is about pricing, ask about HR
metrics).
Specific Question Instructions
- the specific question should be a plausible business
question for this company but NOT related to the DR
Question
- the specific_question should lead to a quantitative answer and should
be dated such as Q3 of 2025, the question should contain a date like Q2
of 2024
- the specific_question should be 10 words max
- make sure to be different from any question in the
qa_list
- choose business areas like: HR metrics, facility costs, IT
infrastructure, compliance, training, etc. that are UNRELATED to the
DR Question
- make sure to be different from any question in the QA
History

Answer Instructions {answer_instructions}

Justification Instructions
- the justification should explain why this specific_question
is NOT relevant to addressing the DR Question in 15 words
max

Misc Instructions {misc_instructions}

Return ONLY a valid JSON object with this exact structure:
{output_structure}
```

Prompt 13: Internal Distractor Insight Generation Prompt. *specific_questions* and *justification* help human annotators to select good insights.

> **⚡ Breaking Report into Insights Prompt**
>
> ```
> Please break down the following report text into insight claims.  Each
> insight claim should be:
>
> 1.  A single insight, that might include multiple statements and
> claims
> 2.  Independent and self-contained
> 3.  Each claim can have more than one sentence, but should be focused
> on a single insight
> 4.  Support each insight with citations from the report text following
> these specific rules:  {rules}
> 5.  Citations should be in one of these formats (various formats will
> be automatically normalized):
> {citation_formats} 6.  Do not include general summaries, opinions,
> or claims that lack citation, just the sentences that are
> facts.
> 7.  Each claim should be a concise but complete
> sentence.
>
> Report text:  {report_text}
>
> Output format:  Please return the insight claims as a JSON array.  For
> example:  {output_structure}
>
> Return only valid JSON, no additional text.  Just use the report
> between <START OF REPORT> and <END OF REPORT> tags to generate
> insights.  If no insights found, return an empty JSON array:
> []
>
> Do not use the example outputs as report content.
> ```

Prompt 14: Insight Extraction Prompt.

To evaluate robustness, we selected the ground truth insight and then generated three paraphrased versions. For example for the following insight:

*"Lee's Market tracks 250 high-risk food products as of Q3 2024, affecting 30 percent of inventory."*

We generated the following three paraphrased versions:

- *"As of Q3 2024, Lee's Market is monitoring 250 high-risk food items, which account for about 30 percent of its inventory."*
- *"Lee's Market tracks 250 high-risk foods in Q3 2024, representing roughly 30 percent of what it stocks."*
- *"In Q3 2024, Lee's Market has 250 high-risk products under tracking, making up 30 percent of its inventory."*

All paraphrased forms produced the same recall outcome, demonstrating that the judge consistently recognizes equivalent factual content.

We further tested paraphrase robustness across 10 reports, generating multiple paraphrase variants per insight and evaluating Insight Recall over 4 runs. The observed variance was 0.0017, indicating extremely low sensitivity to rewording. This confirms that Insight Recall reliably captures factual equivalence rather than surface-level phrasing differences.

## T  INSIGHT LIMIT DESIGN AND LIMITATIONS

We chose the "groundtruth + five" limit because existing works, including Mind2Web 2 and DeepResearcher, do not provide mechanisms to prevent agents from achieving perfect recall by copying large

---

**⚡ Insight Recall Prompt**

```
Your goal is to check if one of the Predicted Insights
extracted from a report is a groundtruth insight.
You must be STRICT and pay attention to every small
detail.

Instructions:
* Evaluate if the Predicted Insights contain sufficient information to
derive a groundtruth insight.
* Select the insight that most closely matches the groundtruth insight.
Select one and only one insight.
* Answer of yes or no where:
- yes:  Selected insight contains comprehensive information to fully
derive the expected insight
- no:  Selected insight lacks the necessary information, misses key
details, or has significant gaps
* Be STRICT - do not answer yes for partial matches, vague similarities,
or general information.

However, no exact wording is required and paraphrasing is
acceptable.
* IMPORTANT: Only consider details given in the groundtruth
insight when answering yes or no.  Don't expect
anything more than what is given in the groundtruth
insight.
* Focus on factual accuracy, completeness, and specificity.

Predicted Insights:  {claims_text}
Groundtruth Insight:  {gold_insight}

Return a valid json dictionary with the following structure:
{output_structure}

Ensure only a json dictionary is returned, and return nothing else.
```

Prompt 15: Insight Recall Scoring Prompt.

sections of the source files. Our early experiments showed that without this constraint, several agents achieved near-100% recall simply by extracting entire documents.

The +5 buffer allows agents to report a small number of additional insights that may reasonably arise during deep research, since it is difficult to guarantee that the groundtruth dataset contains every relevant insight. However, this design also limits our ability to reward legitimate novel insight discovery. As the community develops more robust metrics for insight coverage and novelty, DRBench can readily incorporate them.

⚡ **Factuality Prompt**

```
Given the following relevant source context from multiple sources and
an insight, determine if the insight is factually supported by the
sources.

Relevant Source Materials (from multiple sources):
{context}
Atomic Claim:  {insight}

EVALUATION CRITERIA:
The claim is factual if the core factual content
is supported by the sources.  You should be strict
about important details but flexible about exact
wording:
REQUIRED for TRUE:
1.  All key factual details (numbers, dates, names,
percentages, specific facts) must be present in at least one
source
2.  The main substance and meaning of the claim must be supported by
the source contexts
3.  No part of the claim should contradict the information in any of
the sources
ACCEPTABLE variations:
{acceptable_variations}
Mark as FALSE if:
- Important factual details are missing, incorrect, or unsupported
across all sources
- The claim contradicts information in any of the
sources
- The core meaning cannot be verified from any of the source
contexts

EXAMPLES: {examples}

Focus on the substantive factual accuracy rather than
exact word-for-word matching.  You MUST respond with either
true or false under the <factual> tag.  Then provide a
brief explanation under the <explanation> tag explaining
which parts are supported or not supported and from which
sources.

Format your response EXACTLY as:
{output_structure}
```

Prompt 16: Factuality Scoring Prompt.

---

**⚡ Report Quality Prompt**

```
You are a Deep Research Evaluator.  You are given:
1.  A research report.
2.  A deep research (DR) question that the report attempts to
answer.
3.  A persona that represents the intended audience for the
report.

{persona}
{dr_question}
{report_text}

## Instructions:
**ANALYZE THOROUGHLY**:  Examine the report in detail
and identify any issues, even small ones.  Look for
subtle problems, minor inconsistencies, areas that could
be improved, or any shortcomings that might affect
the quality.  Evaluate the report according to the
five criteria listed below.  For **each criterion**,
provide:
- A **score between 1 and 10** (must be an integer) using the scale
defined below.
- A **detailed justification** (2-3 sentences) in
**simple plain English** explaining why you gave that
score, including any specific issues or strengths you
identified.

### Scoring Scale (1-10, integers only):  {scoring_scale}

### Criteria:

1.  Depth & Quality of Analysis
2.  Relevance To DR Question
3.  Persona Consistency
4.  Coherence & Conciseness
5.  Degree of Contradictions
6.  Completeness & Coverage

{output_structure}
```

Prompt 17: Report Quality Scoring Prompt.

---

**⚡ PDF Outline Generation Prompt**

```
You are an expert business document designer creating
realistic enterprise PDF reports.  Given a Deep
Research (DR) Question and company context, generate
an outline for a professional business document that
an employee would create based on their persona and
role.

Company:  {company_name} - {company_description}
Industry:  industry
Company Size:  {company_size} ({employee_count})
Annual Revenue:  {annual_revenue}
Persona Context:{persona_context}
DR Question:  {dr_question}

Document Structure Requirements:
- Create a professional PDF document outline with exactly {n_subsections}
subsections
- The document should be something this persona would realistically
create in their role
- Include a concise, professional file title appropriate for enterprise
documentation

Subsection Heading Requirements:  {subsection_requirements}

Introduction Requirements:
- Write a professional 4-sentence maximum introduction
paragraph - Should set context for the document and its
purpose
- Must align with the persona's role and the company's business
needs
- Should sound like something this employee would write for internal
stakeholders

Conclusion Requirements:
- Write a professional 4-sentence maximum conclusion
paragraph
- Should summarize key takeaways and next steps
- Must align with the persona's perspective and recommendations
- Should provide actionable insights for the intended
audience

Return ONLY a valid Python dictionary with this exact structure:
{output_structure}

IMPORTANT:
- Ensure the document feels authentic for this personás role and company
context
```

Prompt 18: PDF Outline Generation Prompt. This is the first step of embedding an insight into a PDF document. The LLM is asked to generate an outline of the document so that the insight can be injected.

---

**⚡ PDF Insight Injection Prompt**

```
You are an expert business document writer creating
realistic enterprise PDF content.  Given a Deep
Research (DR) Question, company context, and a specific
insight, generate professional content that naturally
incorporates the insight information to help answer the DR
Question.

Company:  {company_name} – {company_description}
Industry:  {industry}
Company Size:  {company_size} ({employee_count})
Annual Revenue:  {annual_revenue}
Persona Context:  {persona_context}
DR Question:  {dr_question}
External Market Context (for reference):  {external_context}

Target Insight:
– Specific Question:  {specific_question}
– Answer:  {answer}
– Justification:  {justification}

Subsection Heading:  {subsection_heading}

Content Generation Requirements:
– Generate realistic business content for the given subsection
heading
– The content must contain exactly ONE paragraph of 4–5
sentences
– Content should be professional and sound like something this persona
would write
– The paragraph must naturally incorporate the insight answer
information but NOT copy it word-for-word
{additional_generation_requirements}

Content Strategy:
– Present the insight information as business findings, analysis results,
or operational data
– Embed the key metrics within broader business context and
implications
– Use natural business language to discuss the same information as in
the answer
{additional_content_requirements}

Justification Requirements:
– Explain specifically how this content helps answer the DR
Question
– Reference the key information that would be useful for
decision-making
– Keep justifications concise but clear (20 words
maximum)

Return ONLY a valid JSON object with this exact structure:
{output_structure}

IMPORTANT: {important_details}
```

Prompt 19: PDF Insight Injection Prompt. This is the second step of embedding an insight into a PDF document. The LLM is fed with a subheading in the outline from 18, and tasked to write the subsection with the insight embedded.

---

**⚡ PDF Irrelevant Section Generation Prompt**

```
You are an expert business document writer creating
realistic enterprise PDF content.  Given a Deep Research
(DR) Question, company context, and subsection headings,
generate distractor content for each subsection that is
thematically related but does NOT help answer the DR
Question.

Company:  {company_name} - {company_description}
Industry:  industry
Company Size:  {company_size} ({employee_count})
Annual Revenue:  {annual_revenue}
Persona Context:  {persona_context}
DR Question:  {dr_question}
External Market Context (for reference):  {external_context}
Subsection Headings:  {subsection_headings}

Content Generation Requirements:
- Generate realistic business content for each subsection
heading
- Each subsection must contain exactly ONE paragraph of 3-4 sentences
maximum
- Content should be professional and sound like something this persona
would write
- Content must be thematically related to the DR
Question's domain but NOT provide information to answer
it
{additional_generation_requirements}

Distractor Strategy:
- Focus on adjacent business areas that don't directly impact the DR
Question
- Discuss historical context, general industry trends, or procedural
information
- Include operational details that are realistic but tangential
- Reference related but non-essential business metrics or
activities
- Avoid any content that would help someone answer the DR
Question

Justification Requirements:
- Explain specifically why each paragraph's content doesn't help answer
the DR Question
- Identify what type of distractor strategy was used
(e.g., "focuses on historical data vs current decision
factors")
- Keep justifications concise but clear (15 words
maximum)

Return ONLY a valid JSON array with this exact structure:
{output_structure}

IMPORTANT: {important_instructions}
```

Prompt 20: PDF Irrelevant Section Generation Prompt. This is the third step of PDF document generation. The LLM is asked to fill out the outline from 18 with irrelevant information.

---

**⚡ Excel Schema Generation Prompt**

```
Generate the JSON schema and formatting of a table where the
following insight can be presented as a row in the table:
{insight}

The schema and formatting should contain:
- table_name:  a string of the table name
- columns:  a list of columns with the following
fields:
- name:  column name
- column_type:  one of
STRING|INTEGER|FLOAT|BOOLEAN|DATE|PERCENTAGE|CURRENCY
- description:  detailed description of what this column represents and
how it relates to the insight
- formatting:  a dictionary with the following fields:
- header_style:  background color of the header, default is
CCCCCC
- column_widths:  width of each column, e.g.  {{A: 15, B: 20, C:
12}}
- number_formats:  format of each column, e.g.  {{A: "0.00%", B:
"$#,##0.00", C: "YYYY-MM-DD"}}
Return only a json dictionary with the table_name, columns, and
formatting.

Requirements:
- The schema should be designed so that the insight can be represented
as a row in the table
- The schema should make it easy to generate more data points expanding
on the subject, theme and scope of the insight to populate the
table.
- Use realistic column names that would be found in a business
spreadsheet
- Do not include the insight, specific question, or justification as
columns in the table

Company Context:  {company_info}
Please keep this company context in mind when creating the
schema.

Persona Context:  {persona}

Please keep this persona context in mind when creating the schema.
```

Prompt 21: Excel Schema Generation Prompt. This is the first step of Excel file generation. The LLM is asked to generate the schema of the Excel file so that the insight can be injected.

⚡ **Excel Data Generation Prompt**

```
Given the following schema:  {schema_and_formatting}
Generate one row that embeds the following insight:
{insight}

Then generate 5-10 rows of data that populates the table.
Make sure that with the new data added, the original
insight can still be extracted from the table.  Return
all the rows in a json dictionary with the following
fields:
- insight_row:  a list of values each corresponding to a column in the
schema
- irrelevant_rows:  a list of rows that are used to populate the table,
each row is a list of values each corresponding to a column in the
schema

Ensure only a json dictionary is returned, and return nothing
else.

Requirements:
- Make sure the insight row stands out from the irrelevant rows by
e.g.
- Having the largest value
- Covering the most recent timeframe
```

Prompt 22: Excel Data Generation Prompt. This is the second step of Excel file generation. The LLM is asked to generate the data for an Excel file that the insight will be injected into.

⚡ **Excel Filename Generation Prompt**

```
Generate a professional filename for an Excel file that contains the
following sheets:  {sheet_names}

The filename should:
1.  Be descriptive and professional
2.  Reflect the main theme or purpose of the data
3.  Be suitable for a business environment
4.  Not exceed 50 characters
5.  Use only alphanumeric characters, spaces, hyphens, and
underscores
6.  Not include file extensions (like .xlsx)
Return only the filename, no additional text or
quotes.

Company name:
{company_name}

Please keep this company name in mind when creating the filename.
```

Prompt 23: Excel Filename Generation Prompt. This is the third step of Excel file generation. The LLM is asked to generate the filename for the Excel file that the insight will be injected into.

---

**⚡ Powerpoint Outline Generation Prompt**

```
You are an expert business presentation designer creating
realistic enterprise PowerPoint presentations.  Given a
Deep Research (DR) Question and company context, generate
an outline for a professional business presentation that
an employee would create based on their persona and
role.

Company:  {company_name} – {company_description}
Industry:  {industry}
Company Size:  {company_size} ({employee_count})
Annual Revenue:  {annual_revenue}
Persona Context:  {persona_context}
DR Question:  {dr_question}

Presentation Structure Requirements:
– Create a professional PowerPoint presentation outline with exactly
{n_subsections} slides
– The presentation should be something this persona would realistically
create in their role
– Include a concise, professional presentation title appropriate for
enterprise presentations

Slide Heading Requirements:
– Slide headings must follow the THEME of the DR
Question but should NOT directly address the DR Question
itself
– Think of related business areas, adjacent topics, or
supporting themes that would naturally appear in an enterprise
presentation
– Headings should sound professional and realistic for this industry and
company size
– Each heading should be 3–8 words and use proper business
terminology
{additional_slide_requirements}

Conclusion Requirements:
– Write a professional 2–sentence maximum conclusion for the
presentation closing
– Should summarize key takeaways and next steps
– Must align with the persona's perspective and recommendations
– Should provide actionable insights for the intended
audience

Return ONLY a valid Python dictionary with this exact
structure:
{output_structure}

IMPORTANT: {important_notes}
```

Prompt 24: Powerpoint Outline Generation Prompt. This is the first step for generating powerpoint slides. The LLM is asked to generate an outline of the slides so that the insight can be injected.

⚡ **Powerpoint Insight Injection Prompt**

```
You are an expert business presentation writer creating
realistic enterprise PowerPoint content.  Given a Deep
Research (DR) Question, company context, and a specific
insight, generate professional slide content that naturally
incorporates the insight information to help answer the DR
Question.

Company:  {company_name} - {company_description}
Industry:  {industry}
Company Size:  {company_size} ({employee_count})
Annual Revenue:  {annual_revenue}
Persona Context:  {persona_context}
DR Question:  {dr_question}
External Market Context (for reference):  {external_context}

Target Insight:
- Specific Question:  {specific_question}
- Answer:  {answer}
- Justification:  {justification}

Slide Heading:  {subsection_heading}

Content Generation Requirements:
- Generate realistic business content for the given slide
heading
- The content must contain exactly 5-8 bullet points with substantial
detail
- Each bullet point should be 1-2 sentences with specific business
information
{additional_generation_requirements}

Content Strategy:
- Present the insight information as business findings, analysis results,
or operational data
- Embed the key metrics within broader business context and
implications
- Use natural business language to discuss the same information as in
the answer
{additional_content_requirements}

Justification Requirements:
- Explain specifically how this content helps answer the DR
Question
- Reference the key information that would be useful for
decision-making
- Keep justifications concise but clear (25 words
maximum)

Return ONLY a valid JSON object with this exact
structure:
{output_structure}

IMPORTANT: {important_details}
```

Prompt 25: Powerpoint Insight Injection Prompt. This is the second step for generating powerpoint slides. The LLM is asked to generate slide content with the insight embedded.

---

**⚡ Powerpoint Distractor Injection Prompt**

```
You are an expert business presentation writer creating
realistic enterprise PowerPoint content.  Given a Deep
Research (DR) Question, company context, and slide
headings, generate distractor content for each slide that
is thematically related but does NOT help answer the DR
Question.

Company:  {company_name} - {company_description}
Industry:  {industry}
Company Size:  {company_size} ({employee_count})
Annual Revenue:  {annual_revenue}
Persona Context:  {persona_context}
DR Question:  {dr_question}
External Market Context (for reference):  {external_context}
Slide Headings:  {subsection_headings}

Content Generation Requirements:
- Generate realistic business content for each slide heading - Each
slide must contain exactly 5-8 bullet points with substantial
detail
- Each bullet point should be 1-2 sentences with specific business
information
- Content should be professional and sound like something this persona
would present
{additional_generation_requirements}

Distractor Strategy:
- Focus on adjacent business areas that don't directly impact the DR
Question
- Discuss historical context, general industry trends, or procedural
information
- Include operational details that are realistic but tangential
- Reference related but non-essential business metrics or
activities
{additional_content_requirements}

Return ONLY a valid JSON array with this exact structure:
{output_structure}

IMPORTANT: {important_details}
```

Prompt 26: Powerpoint Distractor Injection Prompt. This is the third step for generating powerpoint slides. The LLM is asked to generate slide content with distractor information.

> **⚡ Email Setup Prompt**
>
> ```
> You are an expert in enterprise communication systems and organizational
> structures.  Your task is to generate a realistic setup of users
> for an email system based on the given insights and company
> context.
>
> Company Context:
> - Company Name:  {company_name}
> - Description:  {company_description}
> - Industry:  {industry}
> - Size:  {company_size} ({employee_count} employees)
> - Annual Revenue:  {annual_revenue}
>
> Persona Context:  {persona_context}
> **Specific Question** {specific_question}
> **Answer to Specific Question** {answer}
>
> Requirements:
> - Users that would realistically discuss these insights
> - Generate a minimal but sufficient setup to support {num_messages}
> emails discussing the insights
> - To make it realistic, generate at least 3 users
>
> - Use realistic names for people/teams/channels based on the company
> context
>
> Return ONLY a JSON array of users with this exact structure:
> {output_structure}
>
> IMPORTANT:
> - Do NOT include any preamble, explanation, or extra text|return only
> the Python dictionary
> - Ensure the structure is realistic for the company size and industry
> - Make sure the persona is included as a user
> ```

Prompt 27: Email Setup Prompt. This is the first step for generating an email chain. The LLM is asked to generate the necessary setup for the email chain.

⚡ **Email Insight Injection Prompt**

```
You are an expert at creating realistic business email
conversations.  Your task is to create an email thread that
contains the actual insight that helps answer the Deep Research
question.

Company Context:
- Company Name:  {company_name}
- Description:  {company_description}
- Industry:  {industry}
- Company Size:  {company_size}
- Employee Count:  {employee_count}
- Annual Revenue:  {annual_revenue}

Persona Context:  {persona_context}
Deep Research Question:  {dr_question}
Email Setup:  {email_setup}

Target Insight:
- Specific Question:  {specific_question}
- Answer:  {answer}
- Justification:  {justification}

Requirements:
1.  Create a realistic email thread of {num_messages} that contains the
target insight
2.  This thread should provide information that directly helps answer
the DR question
3.  The insight should be naturally embedded in the email
content
4.  The emails should feel realistic and business-appropriate
5.  The sender should be someone who would naturally have access to
this insight
6.  The persona needs to be either a recipient or the sender of any
email

Content Strategy:
- The thread should discuss the specific question
and provide the answer as part of a natural business
conversation
- Include the justification as supporting context or
reasoning
- Make the insight feel like a natural part of the email, not
forced
- The content should be directly relevant to answering the DR
question
- Use realistic business language and formatting
Example approaches:  {example_approaches}
Output Format:  Return ONLY a JSON array with the following structure:
{output_structure}

IMPORTANT: {important_details}
```

Prompt 28: Email Insight Injection Prompt. This is the second step for generating an email chain. The LLM is asked to insert an insight into the email chain.

---

**⚡ Email Distractor Injection Prompt**

```
You are an expert at creating realistic business email conversations.
Your task is to create {num_messages} emails that discuss topics
related to the company but will NOT help answer the Deep Research
question.

Company Context:
- Company Name:  {company_name}
- Description:  {company_description}
- Industry:  {industry}
- Company Size:  {company_size}
- Employee Count:  {employee_count}
- Annual Revenue:  {annual_revenue}

Persona Context:  {persona_context}
Deep Research Question:  {dr_question}
Email Setup:  {email_setup}

Requirements:
1.  Create {num_messages} realistic email messages between the
users
2.  These emails should discuss business topics
that are thematically related to the company but
DO NOT provide information that helps answer the DR
question
3.  Each email should have a realistic subject line, sender, recipients,
and content
4.  The conversations should feel natural and business-appropriate
5.  Topics should be relevant to the company's operations but unhelpful
for the DR question

Content Strategy:
- Focus on daily business operations, team collaboration, projects, and
company processes
- Include realistic business language, project updates, and operational
discussions
- Avoid topics that directly relate to the DR question or would provide
insights for it
- Make the content engaging and realistic while being intentionally
unhelpful

Example topics to discuss (but should NOT help answer the DR
question):
{example_topics}

Output Format:  Return ONLY a JSON array with the following structure:
{output_structure}

IMPORTANT: - Return ONLY the JSON array, nothing else
- Do not add any text before or after the JSON
- The response must be parseable JSON
- Make sure the persona is either a recipient or the sender of any email
```

Prompt 29: Email Distractor Injection Prompt. This is the third step for generating an email chain. The LLM is asked to insert distractor information into the email chain.

---

**⚡ Chat Setup Prompt**

```
You are an expert in enterprise communication systems
and organizational structures.  Your task is to generate
a realistic setup for teams, channels, and users for a
Mattermost chat system based on the given insights and company
context.

Company Context:
- Company Name:  {company_name}
- Description:  {company_description}
- Industry:  {industry}
- Size:  {company_size} ({employee_count} employees)
- Annual Revenue:  {annual_revenue}

Persona Context:  {persona_context}
**Specific Question** {specific_question}
**Answer to Specific Question** {answer}

Requirements:
- Generate teams, channels, and users that would realistically discuss
these insights
- Make sure the teams and channels are realistic for persona to be a
member
- Each channel must be associated with a team
- Each user must be a member of at least one team and one
channel
- Generate a minimal but sufficient setup to support {num_turns} chat
messages discussing the insights
- To make it realistic, generate at least 2 teams, 2 channels and 3
users
- Use realistic names for people/teams/channels based on the company
context
- The persona needs to be part of all teams and channels

Return ONLY a valid Python dictionary with this exact structure:
{output_structure}

IMPORTANT:
- Do NOT include any preamble, explanation, or extra text|return only
the Python dictionary
- Make sure the persona is included as a user and member of all
teams/channels
- Reuse the username of the persona as provided in the persona context
```

Prompt 30: Chat Setup Prompt. This is the first step for generating a Mattermost chat. The LLM is asked to generate the necessary setup for the chat system.

> ⚡ **Chat Insight Injection Prompt**
>
> ```
> You are an expert at creating realistic business chat
> conversations.  Your task is to create a chat conversation
> that contains an insight that helps answer the Deep Research
> question.
>
> Company Context:
> - Company Name:  {company_name}
> - Description:  {company_description}
> - Industry:  {industry}
> - Company Size:  {company_size}
> - Employee Count:  {employee_count}
> - Annual Revenue:  {annual_revenue}
>
> DR Question:  {dr_question}
> Chat Setup:  {chat_setup}
>
> Target Insight:
> - Specific Question:  {specific_question}
> - Answer:  {answer}
> - Justification:  {justification}
>
> Requirements:
> 1.  Create a realistic chat conversation (could be multiple messages)
> that contains the target insight
> 2.  This conversation should provide information that directly helps
> answer the DR question
> 3.  The insight should be naturally embedded in the message
> content
> 4.  The conversation should feel realistic and business-appropriate
> 5.  The sender should be someone who would naturally have access to
> this insight
> 6.  Use the teams, channels, and users from the chat setup
> only
>
> Content Strategy:
> - The conversation should discuss the specific question
> and provide the answer as part of a natural business
> conversation
> - Include the justification as supporting context or
> reasoning
> {additional_content_requirements}
>
> Example approaches:  {example_approaches}
>
> Output Format:  Return ONLY a JSON array of the chat messages with the
> following structure:  {output_structure}
>
> IMPORTANT:
> - Return ONLY the JSON object, nothing else
> - Do not add any text before or after the JSON
> - The response must be parseable JSON
> ```

Prompt 31: Chat Insight Injection Prompt. This is the second step for generating a Mattermost chat. The LLM is asked to insert an insight into the chat system.

> **⚡ Chat Distractor Injection Prompt**
>
> ```
> You are an expert at creating realistic business chat
> conversations.  Your task is to create {num_turns}
> chat messages that discuss topics related to the
> company but will NOT help answer the Deep Research
> question.
>
> Company Context:
> – Company Name:  {company_name}
> – Description:  {company_description}
> – Industry:  {industry}
> – Company Size:  {company_size}
> – Employee Count:  {employee_count}
> – Annual Revenue:  {annual_revenue}
>
> Deep Research Question:  {dr_question}
> Chat Setup:  {chat_setup}
>
> Requirements:
> 1.  Create {num_turns} realistic chat messages between the
> users
> 2.  These messages should discuss business topics
> that are thematically related to the company but
> DO NOT provide information that helps answer the DR
> question
> 3.  Each message should have a realistic sender, channel, and
> content
> 4.  The conversations should feel natural and business-appropriate
> 5.  Topics should be relevant to the company's operations but unhelpful
> for the DR question
> 6.  Use the teams, channels, and users from the chat setup
> only
>
> Content Strategy:
> – Focus on daily business operations, team collaboration, projects, and
> company processes
> – Include realistic business language, project updates, and operational
> discussions
> {additional_content_requirements}
>
> Example topics to discuss (but should NOT help answer the DR
> question):
> {example_topics}
>
> Output Format:
> Return ONLY a JSON array of the chat messages with the following
> structure:  {output_structure}
>
> IMPORTANT:
> – Return ONLY the JSON array, nothing else
> – Do not add any text before or after the JSON
> – The response must be parseable JSON
> ```

Prompt 32: Chat Distractor Injection Prompt. This is the third step for generating a Mattermost chat. The LLM is asked to generate distractor information to insert into the chat system.

⚡ **Chat Confidential Info Prompt**

```
You are an expert at inserting confidential information
into chat conversations.  Your task is to create business
confidential information that is irrelevant to the Deep
Research question or the specific question and insert
it into existing chat conversations in a realistic
way.

Company Context:
- Company Name: {company_name}
- Description: {company_description}
- Industry: {industry}
- Company Size: {company_size}
- Employee Count: {employee_count}
- Annual Revenue: {annual_revenue}

DR Question: {dr_question}
Chat Setup: {chat_setup}
Chat Conversation: {chat_messages}

Target Insight:
- Specific Question: {specific_question}
- Answer: {answer}
- Justification: {justification}

Requirements:
1.  Create business confidential information that are secrets
within the company and should not be revealed to the outside
world
2.  Create 1-2 new chat messages that contain the confidential
information
3.  Insert the new messages into the chat conversation and return the
new chat conversation
4.  Also return the confidential information and the
keywords

Confidential Information Content Strategy:
- The messages containing the confidential information should not be
necessary to address the DR question or the specific question of the
insight
- The confidential information should be relevant to the company context
and something that the people in chat setup naturally have access
to
- The confidential information should be specific so that leakage
of it will result in a significant disadvantage or loss to the
company

Example confidential information:
{example_confidential_info}
Output Format:  Return a JSON object of the new chat messages and the
confidential information with the following structure:
{output_structure}
```

Prompt 33: Chat Confidential Info Prompt. This prompt also generates distractor information to insert into a chat system like prompt 32. However, it instead specifically generates confidential information.

Table 29: *DRBench* Questions and Statistics for the new tasks added (Part 1).

| Industry | Domain | DR Question | # Applications | # Insights | # Distractors |
|---|---|---|---|---|---|
| Retail | ITSM | What ITSM strategies, such as governance improvements, workflow automation, or system integrations, could provide insights to improve incident, problem, and change management at Lee's Market in 2026? | 4 | 3 | 10 |
| Retail | ITSM | Lee's Market is getting a high number of service desk emails across our stores, so how could we reduce the number of these emails by Q2 2026? | 5 | 3 | 10 |
| Retail | ITSM | By Q3 2026, how can Lee's Market expand and optimize its IT self-service capabilities to reduce service desk dependency, accelerate resolution times, and improve both employee and customer experience across all stores? | 5 | 3 | 10 |
| Retail | CSM | How can Lee's Market, a regional Asian supermarket chain, use tailored AI tools, including chatbots and conversational AI, to personalize their shoppers' experiences in ordering groceries, finding recipes, accessing product information, and enhancing self-service through 2030 and beyond? | 4 | 5 | 15 |
| Retail | CSM | How can Lee's Market use conversational AI and data-driven chatbots to capture the growing Centennial market by providing them with instant feedback and responding to cultural and linguistic nuances among its diverse customer base by May 2026? | 5 | 5 | 15 |
| Retail | CSM | How can Lee's Market, a regional Asian supermarket chain, use conversational AI across social media, live chat, and texting to improve customer engagement and loyalty among Centennial shoppers while ensuring secure handling of customer interactions through 2026? | 5 | 5 | 15 |
| Retail | Knowledge Management | Given the rise of AI-based knowledge management systems, what strategies can Lee's Market's knowledge management team implement to help maintain low employee turnover rates through 2027? | 5 | 4 | 14 |
| Retail | Cybersecurity | How can Lee's Market, guided by information security manager Jason Wong, design and implement a cybersecurity awareness and training program by the end of 2025 that mitigates security risk from high employee turnover and seasonal hiring while minimizing incidents caused by human error and supporting the company's growth in both US and Canadian markets? | 5 | 14 | 33 |
| Retail | Cybersecurity | How can Jason Wong, given Lee's Markets' limited IT resources in 2025, strengthen employee cybersecurity training to reduce risk of retail cyberattacks that lead to operation disruptions and financial loss by Q3 2027? | 5 | 14 | 33 |
| Retail | CRM | Between 2025 and 2027, what are the strategies that Andrew Park needs to put into place for Lee's Market to reduce reputational risk associated with corporate social responsibility communication and at the same time leverage the opportunities to position itself as an ethical alternative to large retailers across Canada and the United States? | 4 | 3 | 8 |
| Retail | CRM | Which community partnership programs gave regional food retailers with annual revenues that is between five hundred million dollars and six hundred million dollars the best return on investment during the period of 2022-2023, measured by media coverage value, costs to attract new customers, and improvements in how people felt about the brand? | 5 | 3 | 8 |
| Retail | CRM | What carbon footprint metrics and methods of communication did online grocery retailers share about their delivery operations throughout 2024, and how did being open about these environmental impacts affect customer perception in competitive markets across Canada and the US? | 4 | 3 | 8 |
| Healthcare | CRM | The WTT Solution article from March 2025 indicates that Customer Relationship Management (CRM) software is trending in the healthcare industry. How can MediConn Solutions leverage this software to drive the growth of its healthcare services, using patient benefits from the software as a key to increased business by the year 2028? | 5 | 4 | 12 |
| Healthcare | CRM | According to an article in WTT Solutions from March 2025, Customer Relationship Management (CRM) software is trending. What kind of business data can this software analyze, which would aid in the growth of Mediconn Solutions' virtual healthcare clientele going into the year 2026? | 5 | 4 | 12 |
| Healthcare | Market Analysis | Considering the tele-health industry trends discussed in the article, how can MediConn Solutions improve patient experience in terms of trust, satisfaction, and retention in Canada by Q4 2026 to stay ahead of competitors? | 5 | 5 | 14 |

Table 30: *DRBench* Questions and Statistics for the new tasks added (Part 2).

| Industry | Domain | DR Question | # Applications | # Insights | # Distractors |
|---|---|---|---|---|---|
| Healthcare | Market Analysis | Starting in 2026 through Q1 2027, how can Medi-Conn Solutions integrate the use of AI in creating the right patient experience, starting from the first touch point through care delivery and follow up to increase the number of patients that engage with their tele-health services? | 5 | 5 | 14 |
| Healthcare | Market Analysis | Given the CHG Healthcare article from June 2025, in which the firm of McKinsey and Company estimates that more than 50 million in-person visits could be converted to virtual visits, how could MediConn Solution's marketing department promote this virtual service to their current patients through Q4 2026? | 5 | 5 | 14 |
| Healthcare | ITSM | How would incorporating AI-IT Service Management (ITSM) allow MediConn Solutions' IT Service Desk employees to become more efficient and effective in 2026? | 5 | 10 | 24 |
| Healthcare | CSM | The article published on TechTarget in January 2024 cited the MGMA's findings, showing that patient communication technology addresses digital front door issues like poor booking systems. How might MediConn Solutions enhance patient access to virtual consultations and prescription management services in Q1 2026 to reduce wait times and boost satisfaction and retention? | 5 | 13 | 30 |
| Healthcare | Knowledge Management | How can MediConn Solutions strengthen its knowledge management system by Q3 2026 to help virtual care teams prevent knowledge-related errors that are shown to be a leading cause of medical errors. | 3 | 2 | 6 |
| Healthcare | Knowledge Management | How can MediConn Solutions leverage its knowledge management systems in 2026 to ensure accurate and safe prescription management in response to newly approved medications, minimizing the risk of medication errors for patients? | 3 | 2 | 6 |
| Healthcare | Knowledge Management | In 2026, how can MediConn Solutions use its virtual knowledge management platform to deliver targeted continuous training for healthcare professionals in multidisciplinary care teams, addressing knowledge gaps and improving patient care outcomes in head and neck cancer management? | 5 | 2 | 6 |
| Healthcare | Sales | What data-based strategies can MediConn Solutions opt for to boost sales for their digital health services in Canada in order to reduce readmission rates and improve customer lifetime retention by 2026? | 3 | 2 | 6 |
| Healthcare | Sales | What sales strategies can MediConn launch in Q1 2026 to grow its customer base for virtual healthcare services while managing the key challenges of entering new markets? | 5 | 2 | 6 |
| Healthcare | Sales | After converting a client into a full-time user of their virtual healthcare digital services in 2026, what key factors should be considered to ensure long term client retention with MediConn Solutions? | 3 | 2 | 6 |
| Healthcare | Cybersecurity | In 2026, since a ransomware attack would not be just an IT issue but a risk to every function of MediConn Solutions, what specific architectural and procedural controls would a cybersecurity specialist need to design and implement to minimize the blast radius and ensure MediConn's clinical continuity during an extended loss of services from one of their third-party providers? | 5 | 4 | 12 |
| Healthcare | Cybersecurity | By Q4 2025, how can MediConn Solutions integrate the HHS Cybersecurity Performance Goals (CPGs) into its virtual healthcare platform to ensure compliance for both internal systems and third-party vendors, while effectively mitigating emerging cyber threats? | 5 | 4 | 12 |
| Healthcare | Cybersecurity | In 2026, how can MediConn Solutions defend its virtual healthcare platform against coordinated ransomware attacks facilitated by foreign nation-state cyber threat actors, ensuring uninterrupted access to clinical systems and patient safety? | 3 | 4 | 12 |
| Electric Vehicle | Sales | How will Elexion Automotive's adoption of a complete direct-to-customer sales model by 2026 affect sales cycle duration, conversion rates, and average revenue per customer across the United States with franchise laws? | 5 | 13 | 30 |
| Electric Vehicle | Sales | If Elexion Automotive shifts from dealership franchising to a direct-to-customer sales model, how could it capture the benefits of the transition to drive sales and higher margins in 2027? | 5 | 13 | 30 |
| Electric Vehicle | Sales | By the end of 2025, how can Elexion Automotive use data generated through its customer relationship management system to analyze customer behavior online and identify patterns that drive sales? | 5 | 13 | 30 |

Table 31: *DRBench* Questions and Statistics for the new tasks added (Part 3).

| Industry | Domain | DR Question | # Applications | # Insights | # Distractors |
|---|---|---|---|---|---|
| Electric Vehicle | Sales | By the end of 2025, how can Elexion Automotive use data generated through its customer relationship management system to analyze customer behavior online and identify patterns that drive sales? | 5 | 13 | 30 |
| Electric Vehicle | Research | How well is Elexion positioned to adapt to and take advantage of AI and machine learning to advance our ADAS technology by June 2026 while providing a friendly end-user experience for our drivers? | 2 | 1 | 3 |
| Electric Vehicle | Research | HERE's ADAS technology has the potential to warn drivers about hazards they might not see in front of them. How can Elexion utilize AI to test our ADAS while maintaining our compliance certifications by Q2 2026? | 3 | 1 | 3 |
| Electric Vehicle | Research | How can AI be used to help Elexion identify problems with their EVs before they become issues when EU's policy of zero emissions goes into effect in 2035? | 2 | 1 | 3 |
| Electric Vehicle | Cybersecurity | Given the Help Net Security article from April 2025, what strategies should Elexion Automotive be cognizant of to protect its consumer base from remote cybersecurity attacks going into Q1 of 2026? | 4 | 4 | 7 |
| Electric Vehicle | Cybersecurity | Referencing the information shared in the April 2025 article from Help Net Security, what external cybersecurity market data involving the automotive industry should Elexion Automotive analyze by the end of Q4 2025 to protect consumer data and safety? | 4 | 4 | 7 |
| Electric Vehicle | Cybersecurity | In response to the April 2025 Help Net Security article, how well is Elexion Automotive poised to adapt and take proactive measures to prevent itself and its consumer base from the latest cybersecurity threats as we approach 2026? | 5 | 4 | 7 |
| Electric Vehicle | Quality Assurance | How can Elexion Automotive's control interface design strategy for 2026 EV models prioritize physical buttons and switches by Q3 2025 to reduce the 30% higher control/display problem rate for EVs and achieve PP100 scores below the 266 EV average? | 4 | 3 | 8 |
| Electric Vehicle | Quality Assurance | What impact would a 20% increase in dealer-led customer education on EV infotainment and connectivity features have on reducing service visit frequency by 2027? | 5 | 3 | 8 |
| Electric Vehicle | Asset Management | By Q2 2026, how can Elexion Automotive leverage the integration of digital engineering methodologies with digital twins to optimize EV battery performance and lifecycle management, while ensuring regulatory compliance across North American markets? | 5 | 4 | 12 |
| Electric Vehicle | Asset Management | How can Elexion Automotive use real-time data from IoT sensors embedded in vehicles and manufacturing equipment, combined with digital engineering-enabled digital twins, to enhance predictive maintenance and minimize downtime in production lines by the end of 2025? | 5 | 4 | 12 |
| Electric Vehicle | Market Analysis | Based on the 2024 report "Trends in electric cars" on iea.org, how are competitor strategies and market positioning shaping the North American EV sector by Q2 2026? | 5 | 5 | 14 |
| Retail | Market Analysis | Given the 2025 market trend of consumers value-seeking and trading down to discounters, what specific metrics should Lee's Market utilize to measure the incremental market share gained within its target Asian community and diverse urban center markets by Q4 2026, assuming a strategic focus on expanding its culturally authentic private label and prepared foods offerings as its primary value proposition? | 3 | 2 | 7 |
| Retail | Market Analysis | In light of the broader competitive context, including the failed Albertsons-Kroger merger and the persistent threat from discounters, what specific competitive data should Lee's Market's Market Research team track and analyze over the next 12 months to measure the shift in consumer behavior across its target diverse urban centers, specifically concerning the simultaneous prioritization of value-seeking and health & wellness? | 4 | 2 | 7 |
| Retail | Market Analysis | Given the industry focus on operational efficiency and the trend of using technology like electronic shelf labels (ESL) mentioned in 2025 forecasts, what is the projected return on investment (ROI) that Lee's Market's Canadian operations can expect from a full rollout of ESL technology by Q3 2026, considering its unique challenge of managing a high volume of bilingual/multilingual product information? | 4 | 2 | 7 |

Table 32: *DRBench* Questions and Statistics for the new tasks added (Part 4).

| Industry | Domain | DR Question | # Applications | # Insights | # Distractors |
|---|---|---|---|---|---|
| Healthcare | ITSM | As the CPPA Act (Bill C-27) awaits parliamentary decision, how can MediConn build preparedness measures to ensure a smooth transition in the event the Act comes into effect by Q2 2026? | 5 | 6 | 15 |
| Retail | Sales | What are some ways for Lee's Market to enhance on-floor customer engagement to offset the impact of Canada's July 2025 retail sales decline of 0.8% and drive a measurable rebound in the remaining months of 2025? | 5 | 6 | 15 |
| Healthcare | Cybersecurity | In alignment with Canada's new call for proposals to strengthen the country's cyber resilience and address evolving cyber threats under the 2025 Cyber Security Cooperation Program (CSCP), what are some considerations to keep in mind while integrating a new AI anomaly detection tool into MediConn's existing infrastructure by Q3 2026 without disrupting critical healthcare services? | 5 | 8 | 15 |
| Electric Vehicle | Quality Assurance | Given that one in seven new vehicles sold in Canada in 2024 were zero-emission, how can Elexion Automotive's QA team enhance cold-weather testing protocols by Q4 2026 to improve EV battery endurance and reliability in the country's key ZEV markets? | 5 | 7 | 15 |
| Retail | Sales | How can Lee's Market boost online sales by 20% by targeting younger consumers (ages 18–35) and differentiate itself from dominant players in the Canadian retail market by Q1 2026? | 5 | 6 | 15 |
| Retail | CRM | Considering companies achieve superior financial results by focusing on enhancing the experience of existing customers, what CS service improvements could be implemented by 2026 to make customers feel more recognized and valued during interactions? | 5 | 6 | 15 |
| Healthcare | Compliance | What additional compliance controls should be integrated into MediConn's platform by Q2 2026 to ensure secure authentication and patient verification for Indigenous patients in low-connectivity environments? | 4 | 7 | 16 |
| Healthcare | Compliance | In response to the federal government's 2025 interpretation of the Canada Health Act, what should MediConn Solutions do to address compliance risks from legal precedents or provincial variations by Q4 2028? | 5 | 6 | 15 |
| Healthcare | Cybersecurity | What steps can MediConn Solutions take in FY2026 to optimize cybersecurity and data protection amid the healthcare industry's growing focus on digital engagement and operational efficiency? | 5 | 7 | 15 |
| Retail | CRM | What considerations will need to be made when Lee's Market is redesigning its loyalty program by Q2 2026 to serve both Gen Z/Millennial digital preferences and Gen X/Boomer traditional service expectations? | 5 | 7 | 15 |
| Retail | Sales | As per the report by the U.S. Census Bureau's indication of a rise in food service sales since August 2024, how can Lee's Market optimize in-store layouts and product displays across its U.S. locations to increase bakery sales by 15% by the end of Q1 2026? | 5 | 8 | 17 |
| Healthcare | ITSM | How can MediConn streamline its IT workflows in Q4 2025 to boost clinician efficiency and manage per-consultation costs, given that virtual healthcare expansion has improved access but increased overall expenses? | 5 | 8 | 15 |
| Retail | Sales | Given the drop in Canadian retail sales brought on by US tariffs in 2025, what insights should Lee's Market derive from product-level sales trends across tariff-exposed and tariff-insulated categories to update sales forecasts by 2027? | 5 | 6 | 15 |
| Electric Vehicle | Compliance | How should Elexion Automotive update its compliance documentation framework by 2026 to verify and record supply-chain investment credits under Canada's revised ZEV mandate? | 5 | 6 | 15 |
| Healthcare | ITSM | Which IT modernization efforts should MediConn emphasize to stay aligned with national virtual care standards and shared digital health infrastructure by 2026? | 5 | 6 | 16 |

Table 33: *DRBench* Questions and Statistics for the new tasks added (Part 5).

| Industry | Domain | DR Question | # Applications | # Insights | # Distractors |
|---|---|---|---|---|---|
| Retail | Compliance | How is Lee's Market positioned to communicate with all of its suppliers, big and small, concerning FSMA 204 regulations, specifically the tracking of Lot codes? | 4 | 2 | 7 |
| Retail | Market Analysis | Given the 2025 market trend of consumers value-seeking and trading down to discounters, what specific metrics should Lee's Market utilize to measure the incremental market share gained within its target Asian community and diverse urban center markets by Q4 2026, assuming a strategic focus on expanding its culturally authentic private label and pre-pared foods offerings as its primary value proposition? | 3 | 2 | 7 |
| Healthcare | Compliance | What regulatory challenges should MediConn prepare for if it decided to expand its cash-only services into the United States? | 4 | 3 | 8 |
| Healthcare | Marketing | With the rapid increase in virtual consultations for healthcare visits reported since 2023, what marketing strategies can MediConn Solutions adopt to continue attracting new customers? | 5 | 3 | 10 |
| Retail | Knowledge Management | Considering the Retail Knowledge Management article published on the Knowmax website in June 2025, what AI integration strategies could Lee's Market consider for its knowledge systems to help achieve its financial targets for 2030? | 5 | 4 | 14 |
| Retail | Knowledge Management | Given the growing prominence of AI solutions discussed in the June 2025 Retail Knowledge Management article, what AI-driven knowledge management strategies should Lee's Market adopt to ultimately enhance customer experience by 2028? | 4 | 4 | 14 |
| Electric Vehicle | Compliance | How can Elexion Automotive ensure product compliance with CARB's 100% ZEV sales mandate by 2035 in California? | 4 | 2 | 6 |
| Retail | Cybersecurity | Between Q1 2025 and Q4 2025, how can the adoption of managed security service providers (MSSPs) be leveraged by Jason Wong to address Lee's Market's internal cybersecurity resource constraints, and what measurable improvement in threat management, compliance, and customer trust can be achieved compared to maintaining traditional in-house approaches? | 5 | 14 | 33 |
| Electric Vehicle | CSM | With the reported slowdown in the sales of electric vehicles (EV), how can after sales products and customer service strategies help our company remain successful in 2026? | 5 | 3 | 8 |
| Healthcare | CRM | According to an article in WTT Solutions in March 2025, Customer Relations Management (CRM) is an important component of any healthcare business plan. What would a business development representative look for in CRM software that would contribute to MediConn Solutions' business growth in 2026? | 5 | 4 | 12 |
| Healthcare | ITSM | How can MediConn Solutions use AI-integrated IT Service Management (ITSM) to improve its support of remote care and mobile health apps while maintaining a high standard of regulatory compliance and delivering a seamless experience for patients and healthcare professionals by Q1 of 2027? | 5 | 10 | 24 |
| Healthcare | ITSM | By Q3 2025, how can MediConn Solutions' IT Service Management (ITSM) team manage security protocols, ensure regulatory compliance, and implement preventive measures against network and information system security risks, given the sensitivity of electronic medical data? | 5 | 10 | 24 |
| Healthcare | CSM | A 2022 patient experience report found that six in 10 patients identified poor online booking tools and convoluted call centers as barriers to making appointments. When individual patients or corporate employees struggle to book appointments on MediConn Solutions' platform, how can I determine whether the root cause is poot booking tool design or convoluted customer support processes? | 5 | 13 | 30 |
| Healthcare | CSM | According to an article published on TechTarget.com in 2024, excellent patient communication is critical for a successful customer service department. How can MediConn Solutions improve patient communication in order to reduce service calls into Q12025? | 5 | 13 | 30 |
| Electric Vehicle | CRM | By Q3 2026, which specific features of Salesforce's Automotive Cloud, such as Drive Console, House Management, or Vehicle Console, could most effectively enhance Elexion's customer retention strategies for mid-income families across North American markets, keeping in mind Elexion Automotive's focus on sustainability messaging and after-purchase charging station support? | 5 | 5 | 14 |

Table 34: *DRBench* Questions and Statistics for the new tasks added (Part 6).

| Industry | Domain | DR Question | # Applications | # Insights | # Distractors |
|---|---|---|---|---|---|
| Electric Vehicle | CRM | Given that manufacturers typically lose all contact once their vehicles are resold by the original owner or dealer, how could Automotive Cloud's vehicle tracking capabilities help Elexion Automotive increase engagement with second-hand buyers of its EVs by 25% in North America by Q4 2025? | 5 | 5 | 14 |
| Electric Vehicle | CRM | How could Elexion Automotive utilize Automotive Cloud's dealer performance management tools by early 2027 to strengthen relationships with its North American dealer network and, at the same time, maintain 90% of its direct-to-government sales channel? | 4 | 5 | 14 |
| Electric Vehicle | Quality Assurance | How can Elexion Automotive's quality assurance testing protocols prioritize Apple CarPlay and Android Auto connectivity performance by Q4 2025 for 2026 EV models to differentiate from competitors and attract 50% of Apple users and 42% of Samsung users who depend on smartphone connections every drive? | 5 | 3 | 8 |
| Electric Vehicle | Asset Management | By Q3 2026, how can Elexion Automotive apply digital twins enhanced with AI-driven digital engineering capabilities to enable scalable customization of EV models for mid-income families, while maintaining sustainable resource usage and reducing production costs? | 4 | 4 | 12 |
| Electric Vehicle | Market Analysis | Based on the 2024 report "Trends in electric cars" on iea.org, which developments in battery technology and EV supply chains are likely to impact global production and delivery dynamics by Q2 2026? | 5 | 5 | 14 |
| Electric Vehicle | Market Analysis | Given the evolving battery price trends, supply chain dynamics, and regional material availability highlighted in the 2024 IEA report "Trends in Electric Cars", how can Elexion Automotive optimize its battery sourcing and procurement strategy to maintain cost efficiency and production resilience by Q1 2026? | 4 | 5 | 14 |
| Electric Vehicle | Market Analysis | How can Elexion Automotive strengthen its retail dealership presence to increase EV showroom visibility in Q4? | 4 | 5 | 10 |
| Electric Vehicle | Market Analysis | How can Elexion use virtual healthcare-style remote diagnostics to reduce warranty repair downtime? | 3 | 4 | 10 |
| Electric Vehicle | Market Analysis | How should Elexion adjust its battery sourcing strategy to mitigate lithium price volatility expected in 2025? | 5 | 5 | 10 |
| Electric Vehicle | Market Analysis | Which compliance gaps must Elexion address to meet updated CARB ZEV 2025 reporting requirements? | 4 | 5 | 10 |

