# OpenReview forum: "DRBench: A Realistic Benchmark for Enterprise Deep Research"
_ICLR.cc/2026/Conference — ICLR 2026 Poster_

### Official Review · Reviewer_9AoK · 2025-10-29

**Soundness:** 3
**Presentation:** 3
**Contribution:** 3
**Rating:** 6
**Confidence:** 4

**Summary:**

The paper proposes DRBench, a benchmark and reproducible enterprise environment for evaluating deep-research agents that must synthesize evidence from both public web sources and private organizational data real across applications. It provides 15 persona-grounded tasks spanning 10 domains with injected ground-truth insights and distractors, generated via a structured LLM+human-in-the-loop pipeline and anchored to dated, authoritative URLs. The evaluation framework scores reports along insight recall, factuality via per-claim citation verification using a fixed RAG pipeline, and multi-dimensional report quality via LLM-as-a-judge. A baseline DRBench Agent with planning variants is analyzed; results show adaptive planning boosts recall while lightweight planning better preserves factuality, and app-based environments are notably harder than local file access. The paper includes ablations across multiple backbone models, browser-only baselines, and a small human study validating metric alignment.

**Strengths:**

- The benchmark genuinely bridges public web retrieval with private enterprise data across heterogeneous formats and real apps, grounded in personas and company context; this goes beyond web-only research settings.
- The evaluation is thoughtfully designed—atomic insight extraction, strict per-claim citation checks with a controlled RAG pipeline, explicit distractor avoidance, and nuanced report quality scoring—plus an anti-gaming cap on evaluated insights.
- A clear, reproducible pipeline with human verification produces distractor-rich files and stable, dated public sources; the environment is containerized and well-documented, enhancing reproducibility.
- Ablations on planning strategies, backbone models, and local vs app-based settings expose concrete failure modes, offering actionable guidance for future agent design.
- DRBench fills a gap between deep research benchmarks and computer-use agent tests, with code/scripts promised and a setup that feels close to real enterprise workflows.

**Weaknesses:**

- The paper’s presentation could be slightly improved;
  - Figure 8 could be better presented.
  - The paper introduces “golden insight” (e.g., Sec. 6 and Prompt 15) without prior definition, seemingly synonymous with “ground-truth insights,” which creates confusion in the evaluation description. Please unify terminology and define the term at first occurrence
  - There are two versions of labels in Figure 4 that overlap with each other.
- Only one backend evaluation (GPT-4o) is conducted. The stability of the evaluation across different backends is not evaluated.

**Questions:**

- What do the stars in L948–L949 mean?
- MinEval selects only the retail domain. Given that stratifying the same total number of tasks across industries (retail, healthcare, EV) need not increase computational/evaluation cost, why not adopt stratified sampling or include at least one representative task per industry?
- Can the agent access public resources beyond the Task URL? If not, how do you limit the behavior of agents like OpenAI Deep Research?

---

> ### Author Response · Authors · 2025-11-27
>
> # Part (1/2) Response to Reviewer (4) 9AoK for DRBench
>
> We appreciate the reviewer’s recognition that DRBench bridges public web retrieval with private enterprise data across heterogeneous formats and real apps, that our evaluation design with atomic insights, citation verification, distractor avoidance, and structured report scoring is well motivated, and that our LLM plus human pipeline yields realistic, distractor rich tasks with strong reproducibility. We address the weaknesses and questions below.
>
>
> ###  *| R4.1) "The paper’s presentation could be slightly improved; Figure 8 could be better presented."*
>
> **Ans:** Thank you for the suggestion. We have revised Figure 8 by presenting the data more clearly in Table 23 instead of using a figure.
>
>
> ### *| R4.2) "The paper introduces “golden insight” without prior definition, seemingly synonymous with “ground-truth insights,” which creates confusion. Please unify terminology and define the term at first occurrence."*
>
> **Ans:** We appreciate this clarification. “Golden insight” and “ground truth insight” refer to the same concept. We have unified the terminology to “ground truth insights” throughout the paper and define it clearly at its first appearance.
>
> In essence, a ground truth insight is a correct fact that provides important information that helps move toward answering the deep research question, even if it only addresses part of it. For example, for a question about how a grocery chain should respond to FSMA 204, a ground truth insight might be “FSMA 204 requires end-to-end traceability for high-risk foods,” since finding this fact helps the agent build the final answer.
>
>
> ### *| R4.3) "There are two versions of labels in Figure 4 that overlap with each other."*
>
> **Ans:** Thank you for pointing this out. We have corrected the label positions in Figure 4 so that they do not overlap.
>
>
> ### *| R4.4) "Only one backend evaluation (GPT 4o) is conducted. The stability of the evaluation across different backends is not evaluated."*
>
> **Ans:** Thank you for pointing this out. We have performed an additional ablation where we compared evaluation results using GPT 4o and Llama 3.1 as the judging backbone.
>
> The relative performance trends remained consistent across both backends, indicating that our evaluation setup is stable. We added Appendix L, including Table 21, which presents the results and discusses the reasons behind this behavior. This point is also now addressed in Section 5.1. See Table below.
>
> | **Model** | **Planning** | **Judge** | **Insight Recall (%)** | **Factuality (%)** | **Distractor Avoidance (%)** | **Report Quality** |
> |----------|--------------|-----------|--------------------------|---------------------|-------------------------------|---------------------|
> | GPT-5 | Complex | GPT-4o | 39.63 | 65.17 | 92.86 | 93.42 |
> | GPT-5 | Complex | Llama-3.1-405B | 38.92 | 65.17 | 92.40 | 91.83 |
> |----------|--------------|-----------|--------------------------|---------------------|-------------------------------|---------------------|
> | DeepSeek Chat 3.1 | Complex | GPT-4o | 30.26 | 70.27 | 96.67 | 86.88 |
> | DeepSeek Chat 3.1 | Complex | Llama-3.1-405B | 29.34 | 70.27 | 96.67 | 85.10 |
> |----------|--------------|-----------|--------------------------|---------------------|-------------------------------|---------------------|
> | Qwen 2.5 72B | Complex | GPT-4o | 26.82 | 58.35 | 97.65 | 89.64 |
> | Qwen 2.5 72B | Complex | Llama-3.1-405B | 26.11 | 58.12 | 97.20 | 87.45 |
> |----------|--------------|-----------|--------------------------|---------------------|-------------------------------|---------------------|
> | GPT-4o | Complex | GPT-4o | 17.31 | 60.84 | 98.33 | 91.62 |
> | GPT-4o | Complex | Llama-3.1-405B | 17.31 | 60.84 | 98.33 | 89.82 |
> |----------|--------------|-----------|--------------------------|---------------------|-------------------------------|---------------------|
> | Llama 3.1 405B | Complex | GPT-4o | 20.16 | 69.75 | 97.90 | 91.26 |
> | Llama 3.1 405B | Complex | Llama-3.1-405B | 19.75 | 69.42 | 97.55 | 89.51 |
>
>
> ### *| R4.5) "What do the stars in L948–L949 mean?"*
>
> **Ans:** We apologize for the typo. The stars were simply formatting artifacts meant to bold the text between them. We have corrected this.
>
> **Part 2 of this Rebuttal continues below.**

---

> > ### Author Response · Authors · 2025-11-27
> >
> > # Part (2/2) Response to Reviewer (4) 9AoK for DRBench
> >
> > ### *| R4.6) "MinEval selects only the retail domain. Given that stratifying the same total number of tasks across industries need not increase cost, why not adopt stratified sampling or include at least one representative task per industry?"*
> >
> > **Ans:** We selected MinEval to focus on representing all three difficulty levels within a single industry. Each industry contains exactly two easy, two medium, and one hard tasks, and using one domain avoids confounding factors with cross industry variation.
> >
> >
> > ### *| R4.7) "Can the agent access public resources beyond the Task URL? If not, how do you limit the behavior of agents like OpenAI Deep Research?"*
> >
> > **Ans:** Yes, agents can access public resources beyond the Task URL through normal web search tools. The Task URL is not provided to the agent. It is only used to ground the task based on a realistic external knowledge.
> >
> > Therefore, the agent must discover it through its own search process based on the deep research question and it has the whole web as its search space.
> >
> > For example, for the deep research question "How can Lee’s Market leverage FSMA 204 regulations to enhance food safety and customer trust?", the agent needs to issue a targeted search query such as "FSMA 204 FDA requirements" or "FSMA 204 traceability rule" in order to retrieve the correct public information.

---

### Official Review · Reviewer_hv9E · 2025-10-30

**Soundness:** 3
**Presentation:** 3
**Contribution:** 2
**Rating:** 6
**Confidence:** 4

**Summary:**

The paper introduces DRBench, a benchmark for answering deep research questions that require synthesizing information from both Web pages and enterprise documents embedded in files or apps like emails. The 15 tasks are curated using LLMs with a human-in-the-loop approach that involves generating a company and persona, collecting relevant public URLs and extracting insights from them, generating questions based on the public insights, generating internal insights and distractors for those questions, and generating internal documents to embed those insights. The answers are evaluated based on insight recall, factuality, distractor avoidance and report quality measure using LLM-as-a-Judge. A baseline DRBench Agent (DRBA) is also developed that consists of research planning, action planning, adaptive action planning and report generation. Experimental results show that even SoTA models struggle with these tasks, particularly on insight recall.

**Strengths:**

1. Novel and challenging task that involves assimilating information from various sources and also interacting with apps like emails
2. Systematic pipeline for benchmark creation
3. Extensive experiments and analyses to measure the performance of many models on four comprehensive criteria. Results demonstrate benchmark complexity
4. Human evaluation to validate both benchmark creation and evaluation metrics

**Weaknesses:**

1. Missing relevant work "Benchmarking Deep Search over Heterogeneous Enterprise Data" by Choubey et al.
2. Very small dataset consisting of only 15 tasks
3. No agent identified public insights. There should be some analysis done to understand if that is due to lack of web indexing, retrieval approach, tool limitations, benchmark design, agent extraction etc. For example, are the public insights inaccessible by the search tool?

**Questions:**

1. Although some tables show (e.g., Table 11) show number of questions answered successfully, why is such an accuracy not reported for all tasks and models? This seems to be the most important evaluation criterion. For example, an agent might retrieve all insights but still not synthesize them into the correct answer. LLM-as-a-Judge could be used for this criterion too. Does the benchmark include gold answers for all questions?
2. Which model is used for LLM-as-a-Judge? This could impact results as the judge LLM is known to be biased towards models from its own family.

---

> ### Author Response · Authors · 2025-11-27
>
> # (Part 1/2) Response to Reviewer (3) hv9E for DRBench
>
> We appreciate the reviewer’s recognition of several key strengths of DRBench, including the novelty and difficulty of integrating public web information with enterprise documents and apps, the systematic pipeline for benchmark creation, the extensive experiments evaluating multiple models on four metrics, and the human evaluations used to validate both task construction and the metrics. We address the reviewer’s concerns below.
>
>
> ### | *R3.1) "Missing relevant work "Benchmarking Deep Search over Heterogeneous Enterprise Data" by Choubey et al."*
>
> **Ans:** We thank the reviewer for pointing out this relevant work.  We have added Choubey et al. in the related work section.
>
> To highlight the difference between ours and their work, their benchmark poses more closed-form, fact-retrieval queries such as finding an employee ID, identifying an unimplemented feature, or locating a specific demo URL within engineering artifacts.
>
> In contrast, DRBench questions require multi-step deep research for more high-level questions that require multiple supporting insights from both private enterprise files and public web sources grounded on company context, such as analyzing how FSMA 204 regulations affect retail operations or how to leverage AI driven analytics to improve ITSM processes.
>
>
> ### | *R3.2) "Very small dataset consisting of only 15 tasks"*
>
> **Ans:** We acknowledge that the original benchmark was limited in size. We address this by expanding it substantially from 15 tasks to 100, making it larger than existing benchmarks such as AgentCompany [A] and ResearcherBench [B].
>
> As shown in Table 6 in the revised paper (in Section 6: Results on the Full Benchmark), the expanded benchmark produces results consistent with the original results, which confirms that it scales reliably and offers sufficient coverage for evaluating deep research agents. These 85 new tasks are now added in the Appendix (see tables 30, 31, 32, 33, 34, and 35), and also are accessible via this link: https://drive.google.com/drive/folders/1t2AIdxAlIDM9-yjZ20pSMlIYfDff9E0L?usp=sharing
>
> Please note that unlike web-only deep research benchmarks, each task in our benchmark includes having to carefully create private data and realistic files with supporting and distractor facts, which presents a unique challenging evaluation setting.  We will consolidate the tables in the paper by Dec 3rd to include the 100 task results as the main results.
>
>
> ### | *R3.3) "No agent identified public insights. There should be some analysis done to understand if that is due to lack of web indexing, retrieval approach, tool limitations, benchmark design, agent extraction etc."*
>
> **Ans:** We acknowledge that our paper did not provide enough detail about why no agent was able to identify the public insights. We will update the paper with a clearer explanation.
> In summary, the lack of successfully retrieving external knowledge can be explained as follows. For example, for the question "How can Lee’s Market leverage FSMA 204 regulations to enhance food safety and customer trust," the FSMA 204 regulation does not appear in the private files. The agent must detect this and perform a targeted web search for FSMA 204 sources. Instead, we observed that agents produced broad queries such as "how grocery stores improve customer trust," "food safety best practices," and "ways to strengthen customer loyalty," none of which return FSMA 204 content from the web. We added this explanation to the revised paper Section 7 (Discussion).
>
>
> ### | *R3.4) "Although some tables show the number of questions answered successfully, why is such an accuracy not reported for all tasks and models? This seems to be the most important evaluation criterion. Does the benchmark include gold answers for all questions?"*
>
> **Ans:** The computed insight recall is in fact the accuracy, because it represents how many of the groundtruth insights tied to each question the agent successfully identified, and therefore how many questions it was able to answer. We have updated the paper in Section 5.1 to clarify this.
>
> **Part 2 of this Rebuttal continues below.**

---

> ### Author Response · Authors · 2025-11-27
>
> # (Part 2/2) Response to Reviewer (3) hv9E for DRBench
>
> ### | *R3.5) "Which model is used for LLM-as-a-Judge? This could impact results as the judge LLM is known to be biased towards models from its own family."*
>
> **Ans:** We use GPT 4o as the judge model, and we agree that this could introduce bias.
>
> To check this, we ran an additional ablation where we evaluated the same outputs using both GPT 4o and Llama 3.1 as the judging backbone. The relative performance rankings remained consistent across the two judges (see Table below), indicating that our evaluation is stable across backends (both commercial and opensource).
>
> We added Appendix L, including Table 21, which presents the results and discusses the reasons behind this behavior. This point is also now addressed in Section 5.1 and we will add this option to the released codebase.
>
>
> | **Model** | **Planning** | **Judge** | **Insight Recall (%)** | **Factuality (%)** | **Distractor Avoidance (%)** | **Report Quality** |
> |----------|--------------|-----------|--------------------------|---------------------|-------------------------------|---------------------|
> | GPT-5 | Complex | GPT-4o | 39.63 | 65.17 | 92.86 | 93.42 |
> | GPT-5 | Complex | Llama-3.1-405B | 38.92 | 65.17 | 92.40 | 91.83 |
> |----------|--------------|-----------|--------------------------|---------------------|-------------------------------|---------------------|
> | DeepSeek Chat 3.1 | Complex | GPT-4o | 30.26 | 70.27 | 96.67 | 86.88 |
> | DeepSeek Chat 3.1 | Complex | Llama-3.1-405B | 29.34 | 70.27 | 96.67 | 85.10 |
> |----------|--------------|-----------|--------------------------|---------------------|-------------------------------|---------------------|
> | Qwen 2.5 72B | Complex | GPT-4o | 26.82 | 58.35 | 97.65 | 89.64 |
> | Qwen 2.5 72B | Complex | Llama-3.1-405B | 26.11 | 58.12 | 97.20 | 87.45 |
> |----------|--------------|-----------|--------------------------|---------------------|-------------------------------|---------------------|
> | GPT-4o | Complex | GPT-4o | 17.31 | 60.84 | 98.33 | 91.62 |
> | GPT-4o | Complex | Llama-3.1-405B | 17.31 | 60.84 | 98.33 | 89.82 |
> |----------|--------------|-----------|--------------------------|---------------------|-------------------------------|---------------------|
> | Llama 3.1 405B | Complex | GPT-4o | 20.16 | 69.75 | 97.90 | 91.26 |
> | Llama 3.1 405B | Complex | Llama-3.1-405B | 19.75 | 69.42 | 97.55 | 89.51 |
>
> ## References
>
> - [A] Xu, Frank F., et al. "Theagentcompany: benchmarking llm agents on consequential real world tasks."
> - [B] Xu, Tianze, et al. "Researcherbench: Evaluating deep AI research systems on the frontiers of scientific inquiry."
> - [C] Gou, Boyu, et al. "Mind2Web 2: Evaluating Agentic Search with Agent as a Judge." NeurIPS, 2024.
> - [D] Feng, Yufan, et al. "DeepResearcher: Scaling Deep Research with LLM Agents." 2024.

---

### Official Review · Reviewer_qNkw · 2025-10-31

**Soundness:** 3
**Presentation:** 3
**Contribution:** 3
**Rating:** 6
**Confidence:** 4

**Summary:**

This paper introduces DRBench, a benchmark for evaluating deep-research agents that must integrate public web information with private, enterprise-like data (e.g., files, chats, emails) inside a realistic multi-app environment. It proposes three evaluation axes—Insight Recall & Distractor Avoidance, Factuality (via evidence-checked citations), and Report Quality—and presents a baseline DRBench Agent (DRBA) with variants (SRP/CRP/AAP). Experiments span 15 persona-grounded tasks across 10 domains, with analyses of planning strategies, backbone LLMs, and app-based vs. local file access.

**Strengths:**

The paper convincingly argues that prior deep-research benchmarks are predominantly web-only and do not measure whether agents surface the most salient enterprise insights or ground claims with citations.

Tasks require tool use across storage, chat, email, and documents, which distinguishes DRBench from web-only retro-search settings such as DeepResearchGym and Deep Research Bench, both of which rely on fixed corpora or “frozen web” for reproducibility rather than mixed private+public sources.

Multi-axis evaluation design. The insight recall vs. distractor avoidance split is well motivated; factuality uses RAG-style evidence checks; report quality is judged on structured dimensions. The methodology reflects current best practice in LLM-as-a-judge evaluations.

**Weaknesses:**

Evidence of external validity & task coverage. While the 15 tasks are persona-grounded, the coverage across industries and the depth of internal knowledge heterogeneity remains modest. Benchmarks such as DeepResearchGym and BrowseComp-Plus now report hundreds to thousands of instances or large curated corpora; DRBench’s small task count risks overfitting and limited statistical power.

LLM-as-judge reliance & bias. All key metrics (recall alignment, factuality judgments, report quality) ultimately depend on LLM judges. The paper would benefit from more thorough human-vs-LLM agreement studies and inter-rater reliability beyond the limited assessments reported.

How robust is Insight Recall to paraphrase or partial matches? Lack of evaluating the span-level alignment and evidence coverage per insight.

**Questions:**

Can you quantify LLM-judge and human agreement for each metric (beyond small samples), and report Fleiss’ κ or Krippendorff’s α per dimension?

What are the exact artifacts you will release (images, VM snapshots, container specs, synthetic email/chat generators, grading scripts)? Any non-redistributable components?

How robust is Insight Recall to paraphrase or partial matches? Do you evaluate span-level alignment and evidence coverage per insight?

---

> ### Author Response · Authors · 2025-11-27
>
> ## Part (1/2) Response to Reviewer 2 qNkw for DRBench
>
> We appreciate the reviewer’s recognition that DRBench addresses limitations of prior web only benchmarks, that our multi app enterprise setting meaningfully extends deep research evaluation, and that our multi axis metrics follow current best practices in LLM-as-a-judge evaluation. We address the concerns below.
>
>
> ### | *R2.1) "Evidence of external validity and task coverage. While the 15 tasks are persona grounded, the coverage across industries and the depth of internal knowledge heterogeneity remains modest."*
>
> **Ans:** We acknowledge that the original benchmark was limited in size. We address this by expanding it substantially from 15 tasks to 100, making it larger than existing benchmarks such as AgentCompany [A] and ResearcherBench [B].
>
> As shown in Table 6 in the revised paper (in Section 6: Results on the Full Benchmark), the expanded benchmark produces results consistent with the original results, which confirms that it scales reliably and offers sufficient coverage for evaluating deep research agents. These 85 new tasks are now added in the Appendix (see Tables 30, 31, 32, 33, 34, and 35), and also are accessible via this link: https://drive.google.com/drive/folders/1t2AIdxAlIDM9-yjZ20pSMlIYfDff9E0L?usp=sharing
>
> Please note that unlike web-only deep research benchmarks, each task in our benchmark includes having to carefully create private data and realistic files with supporting and distractor facts, which presents a unique challenging evaluation setting.  We will consolidate the tables in the paper by Dec 3rd to include the 100 task results as the main results.
>
>
> ### | *R2.2) "LLM as judge reliance and bias. All key metrics ultimately depend on LLM judges."*
>
> **Ans:** We agree that LLM-as-a-judge can introduce some noise, but for open ended outputs like evaluating insights it is currently the only practical evaluation method.
>
> Existing deep research benchmarks also rely on LLM as a judge, including Mind2Web 2 [C] and DeepResearchBench [D], because metrics like exact match or ROUGE do not work when many different paraphrases can express the same correct idea.
> Having said that, we address this concern by validating our LLM-as-a-judge setup with an additional human study.
>
> We asked humans to judge whether each predicted insight appeared in the ground truth insight list and compared their answers with the LLM-as-a-judge decisions. The agreement between the humans and the LLM-as-a-judge was over 91.3% with a Cohen's Kappa score of 0.683 which represents substantial agreement [E] across 5 evaluators and 275 insights.
>
> The results here show that our LLM-as-a-judge setup is reliable for this evaluation.
>
>
> ### | *R2.3) "How robust is Insight Recall to paraphrase or partial matches? Lack of evaluating the span level alignment and evidence coverage per insight."*
>
> **Ans:** Insight Recall is robust to paraphrasing because our judging prompt focuses on whether the key facts are present, not whether the wording matches exactly. To check this, we took the ground truth insight "Lee’s Market tracks 250 high-risk food products as of Q3 2024, affecting 30 percent of inventory." and rewrote it in three different ways. All three paraphrases produced the same recall result:
>
> - "As of Q3 2024, Lee’s Market is monitoring 250 high-risk food items, which account for about 30 percent of its inventory."
> - "Lee’s Market tracks 250 high-risk foods in Q3 2024, representing roughly 30 percent of what it stocks."
> - "In Q3 2024, Lee’s Market has 250 high-risk products under tracking, making up 30 percent of its inventory."
>
> We also ran different paraphrases of the insights on 10 reports and found that variance across 4 runs to be 0.0016 on insight recall.
> This confirms that the metric reliably recognizes paraphrased versions of the same insight. We have added these results at Appendix T and referred to it in Section 5.2.
>
>
> ### *R2.4) "Can you quantify LLM judge and human agreement ... and report Fleiss κ or Krippendorff α?"*
>
> **Ans:** We have computed the inter-annotator agreement for the human evaluation experiment mentioned in our R2.2 response. The agreement between the humans and the LLM-as-a-judge was over 91.3% with a Cohen's Kappa score of 0.683 which represents substantial agreement [E] across 5 evaluators and 275 insights.
>
> We have obtained a Fleiss κ of 0.75 and a Krippendorff α of 0.75 for the Insight Recall metric, which is considered **Substantial Agreement**.  And we have obtained a Fleiss κ of 0.853 and a Krippendorff α of 0.851 for the Distractor Recall metric which is **Almost Perfect Agreement** .
>
> **Part 2 of this rebuttal continues below**

---

> > ### Author Response · Authors · 2025-11-27
> >
> > # (Part 2/2) Response to Reviewer (2) hv9E for DRBench
> >
> > ### | *R2.5) "What are the exact artifacts you will release (images, VM snapshots, container specs, synthetic email or chat generators, grading scripts)? Any non redistributable components?"*
> >
> > **Ans:** We will release all components of DRBench, including the private style enterprise files, synthetic chats and emails, distractor injected documents, task definitions, and grading scripts. All data is synthetic and human refined, and there are no non redistributable components.
> >
> > ## References
> > - [A] Xu, Frank F., et al. "Theagentcompany: benchmarking llm agents on consequential real world tasks."
> > - [B] Xu, Tianze, et al. "Researcherbench: Evaluating deep AI research systems on the frontiers of scientific inquiry."
> > - [C] Gou, Boyu, et al. "Mind2Web 2: Evaluating Agentic Search with Agent as a Judge." NeurIPS, 2024.
> > - [D] Feng, Yufan, et al. "DeepResearcher: Scaling Deep Research with LLM Agents." 2024.

---

### Official Review · Reviewer_jNFK · 2025-11-01

**Soundness:** 2
**Presentation:** 3
**Contribution:** 2
**Rating:** 4
**Confidence:** 4

**Summary:**

This paper introduces DRBench, a benchmark for evaluating AI agents on multi-step, long horizon enterprise deep research tasks. It consists of 15 persona grounded tasks across 10 domains. The tasks comprises of retrieval and insights generation from public web content as well as private enterprise data (emails, chats, documents, spreadsheets) to answer business related queries. It proposes 4 evaluation metrics - Insight Recall, Factuality, Distractor Avoidance, Report Quality. It introduces DRBench baseline agent and evaluates it on the benchmark across multiple planning strategies and backbone models.

**Strengths:**

1. The paper tackles an important challenge of enterprise deep research which is a problem space that remains highly unexplored.
2. The inclusion of private datasources simulating real world applications (such as cloud storage, chat, file system etc) and containing diverse file formats creates a more realistic evaluation environment. The benchmark incorporates private datasources distributed across realistic enterprise data sources such as cloud storage, chat, file system etc containing diverse file formats, resulting in a highly authentic simulation environment.
3. The evaluation framework consists of multiple complimentary metrics that help in evaluating agentic systems across both precision and recall and quality of report.
4. The paper includes ablation studies across planning strategies, backbone llms and environmental settings (local vs app based).

**Weaknesses:**

1. A major limitation of the benchmark is its limited size. 15 tasks and 114 insights makes the benchmark significantly smaller which raises questions on statistical significance of the evaluation.
2. Extraction of atomic insights from the final report is a very important step in the evaluation method since 3 of the 4 metrics depend on it. However due to the use of llms, this step will be noisy which will lead to less reliable metric scores.
3. Having LLM-as-a-judge as the only method of evaluation raises question about the accuracy of evaluation since llms can halucinate and show biasness. Even though the authors talk about correlation with human preference, It does not really indicate how accurate the llm evaluations are.
4. Synthetic data generation, even though it makes the benchmark generation approach more scalable, raises questions about the internal enterprise data being realistic in nature. Combined with the fact that LLM is also used for evaluation, it can lead to more noise and biasness in evaluation results.
5. Even though the paper includes several ablation studies, it does not provide indepth analysis of why the results are the way they are. It simply states that certain model / approach is better than the other without trying to provide any explanation as to why it might be so. A main example of this is stating the fact that no agent managed to successfully source external knowledge without providing any explanation to why it happened.

**Questions:**

1. Results show relatively poor performance in insights recall metric across models and methods. Why is it so? What % of it is due to incorrect / noisy insights extraction?
2. How good is atomic insights decomposition? Is there any quantitative analysis done to measure the performance of insights decomposition as well as the different metrics scoring?
3. The decision to choose number of ground-truth insights plus five for calculating insights recall score seems arbitrary. Was some other methods for penalising copying all content into the generated report explored?

---

> ### Author Response · Authors · 2025-11-27
>
> # Part (1/3) Response to Reviewer (1) jNFK for DRBench
>
> We appreciate the reviewer’s recognition of DRBench’s framework as a realistic benchmark for enterprise deep research, including its focus on an important and largely unexplored problem space, the inclusion of private enterprise data within a realistic evaluation environment, and the evaluation metrics that evaluates precision, recall, factuality, and report quality.
>
> **Below we address each of your concerns:**
>
>
> ### |*R1.1) The benchmark size is small*
>
> **Ans:** We acknowledge that the original benchmark was limited in size. We address this by expanding it substantially from 15 tasks to **100**, making it larger than existing benchmarks such as AgentCompany [A] and ResearcherBench [B].
>
> As shown in Table 6 in the revised paper (in Section 6: Results on the Full Benchmark), the expanded benchmark produces results consistent with the original results, which confirms that it scales reliably and offers sufficient coverage for evaluating deep research agents. These 85 new tasks are now added in the Appendix (see Tables 30, 31, 32, 33, 34, and 35), and also are accessible via this link: https://drive.google.com/drive/folders/1t2AIdxAlIDM9-yjZ20pSMlIYfDff9E0L?usp=sharing
>
> Please note that unlike web-only deep research benchmarks, each task in our benchmark includes having to carefully create private data and realistic files with supporting and distractor facts, which presents a unique challenging evaluation setting.  We will consolidate the tables in the paper by Dec 3rd to include the 100 task results as the main results.
>
>
> ### | *R1.2) Synthetic data generation, even though it makes the benchmark generation approach more scalable, raises questions about the internal enterprise data being realistic in nature.*
>
> **Ans:** To ensure that the synthetic data is as realistic as possible, we used real data that was provided to humans as seed for concocting plausible, realistic enterprise deep research tasks. They started with our synthetic data generation method to generate an initial set, then refined them and made sure they were realistic, relevant, and high-quality.
>
> Enterprise experts then reviewed each task in detail (including the context, company information, persona, and all supporting and distractor facts) and approved or fine-tuned them as needed.  For transparency, the deep research tasks are included in the Appendix.
>
>
> ### | *R1.3) The decision to choose number of ground truth insights plus five for calculating insights recall score seems arbitrary.*
>
> **Ans:** We agree that setting the cutoff to "ground truth insights plus five" seems arbitrary.
>
> We chose this design because we have not found a method in existing works, including Mind2Web 2 [C] and DeepResearcher [D], that prevents agents from gaming the metric by copying large portions of the source files.
>
> Our plus five limit prevents this behaviour while still giving the agent room to report a few additional insights that can reasonably arise during deep research, since it is difficult to guarantee that the groundtruth dataset contains every relevant insight.
>
> Note that we do have distractor insights injected into the enterprise files. If an agent extracts these distractors, the distractor avoidance score decreases, so copying is naturally penalized. At the same time, agents may uncover insights that are relevant to the deep research question but are not explicitly labeled as supporting or distracting. The small plus five buffer allows space for these insights without resulting in a penalty.
>
> While this design might not be ideal, it provides a meaningful metric that is more difficult to game and most importantly is stable across evaluations. As the community makes progress in this open problem, new metrics can be readily incorporated to this benchmark. We emphasized this limitation in the paper in Section 5.1.
>
> **Part 2/3 of this rebuttal continues below**

---

> ### Author Response · Authors · 2025-11-27
>
> # Part (2/3) Response to Reviewer (1) jNFK for DRBench
>
>
> ### | *R1.4) Even though the paper includes several ablation studies, it does not provide indepth analysis of why the results are the way they are….*
>
> **Ans:** We acknowledge that our paper did not provide enough detail about why different models behave the way they do.
> We added these explanations to the revised paper in Section 7 (Discussion).
>
> - **To summarize here,** the results in Table 3 and Table 5 show that larger models such as GPT 5 achieve higher insight recall and better report quality. This is consistent with the qualitative examples in Table 12, where larger models retrieve both the numeric value and the contextual information more often than smaller models, which tend to repeat only the explicit numeric portions of an insight.
>
> - **Moreover,** the planning results in Table 3 help explain part of the performance differences. Complex planning increases insight recall for GPT 5 because it promotes stepwise extraction across multiple files. In the qualitative examples, the insights that require combining a number with a specific time period or business detail are recalled more reliably by GPT 5 with planning. Smaller models show limited improvement and lower factuality because they fail to reliably merge information from separate parts of the data. Table 5 also shows that weaker models experience larger drops when working in the app environment, which involves more navigation steps, while stronger models remain more stable.
>
> - **Digging deeper,** we find that the lack of successfully retrieving external knowledge can be explained as follows. For example, for the question **"How can Lee’s Market leverage FSMA 204 regulations to enhance food safety and customer trust,"** the FSMA 204 regulation does not appear in the private files. The agent must detect this and perform a targeted web search for FSMA 204 sources. Instead, we observed that agents produced broad queries such as **"how grocery stores improve customer trust,"** **"food safety best practices,"** and **"ways to strengthen customer loyalty,"** none of which return FSMA 204 content from the web.
>
> ### | *R1.5) What percent of it is due to incorrect or noisy insights extraction? Extraction of atomic insights is a noisy step. How good is atomic insights decomposition? Is there any quantitative analysis?*
>
> **Ans:** Please note that we use a similar atomic insight extraction approach as Mind2Web 2, which has shown strong evidence of reliability [C].
>
> That said, we recognize that this step can introduce some noise. This is why we emphasize that agents must produce their own list of atomic insights for the benchmark to evaluate them accurately across insight recall, distractor recall, and factuality.
>
> We also want to clarify a mistake in Section 5.1. The sentence stating that we "first decompose each report into atomic insights" was written incorrectly. In our actual setup, the agents provide the insight list directly, and that is what we use for all reported results.
>
> This means the quality of decomposition in our results comes entirely from the agent, not from our method.
>
> As suggested, we validate the extraction method to see how much noise may affect insight recall. We ran an experiment across the 15 tasks, focusing on 80 of the insights.
>
> For each task, we copied the relevant subsections (where the insight has been injected) from the enterprise source files word for word, aggregated them into a report, and applied the same insight extractor used by our agent.
>
> The results showed that 7.5% (6/80) of groundtruth insights could **not** be identified by the agent's extractor which  contributes to the decrease in insight recall.
>
> In addition, we also found that our insight extractor is consistent in the set of insights it extracts across runs.
>
> We ran the process three times, and for the same report it produced the same set of insights with only minor paraphrasing. These variations were small and did not affect evaluation, as our LLM-as-a-judge setup consistently computed the same result of 7.5% (6/80) of insights not being recalled.
>
> **Part 3/3 of this rebuttal continues below**

---

> > ### Author Response · Authors · 2025-11-27
> >
> > # Part (3/3) Response to Reviewer (1) jNFK for DRBench
> >
> > ### *R1.6) Having LLM-as-a-judge raises questions about the accuracy of evaluation.*
> >
> > **Ans:** We agree that **LLM-as-a-judge** can introduce some noise, but for open ended outputs it is currently the only practical evaluation method.
> >
> > Existing deep research benchmarks rely on LLM-as-a-judge, including **Mind2Web 2 [C]** and **DeepResearchBench [D]**, because exact match metrics cannot evaluate paraphrased insights.
> >
> > We validated our setup with a human study (**Appendix S**).
> >
> > Humans judged whether each predicted insight appeared in the ground truth list, and we compared their answers with the LLM-as-a-judge outcomes.
> >
> > Agreement was **91.3%**, with a **Cohen's Kappa of 0.683**, representing **substantial agreement [E]** across 5 evaluators and 275 insights, which indicates the LLM-as-a-judge setup is reliable for this evaluation.
> >
> > ## References
> > - [A] Xu, Frank F., et al. "Theagentcompany: benchmarking llm agents on consequential real world tasks."
> > - [B] Xu, Tianze, et al. "Researcherbench: Evaluating deep AI research systems on the frontiers of scientific inquiry."
> > - [C] Gou, Boyu, et al. "Mind2Web 2: Evaluating Agentic Search with Agent as a Judge." NeurIPS, 2024.
> > - [D] Feng, Yufan, et al. "DeepResearcher: Scaling Deep Research with LLM Agents." 2024.
> > - [E] Gwet, K. (2010). "Handbook of Inter-Rater Reliability (Second Edition)"

---

### Author Response · Authors · 2025-12-03
**DRBench Review & Rebuttal Summary**

Dear AC,

To summarize, we have addressed all reviewers’ feedback on DRBench in our rebuttal. Our current scores are 4, 6, 6, and 6 and the concerns raised by the reviewer who gave a 4 were straightforward to resolve. The revised rebuttal provides clear clarifications, added experiments and strengthened analysis that directly address every point.

Here are the four key updates:
1) **We have expanded and strengthened the benchmark** by increasing it from 15 to 100 tasks which makes it larger than existing benchmarks such as AgentCompany and ResearcherBench. The results scale reliably and the expanded benchmark offers sufficient coverage for evaluating deep research agents.
2) **We have added new human evaluations and experiments to validate** the metrics by running a detailed human study that compared human judgments with the LLM-as-a-judge decisions. The agreement exceeded 91.3 percent which shows strong alignment between human and LLM evaluations and confirms that our judging setup is reliable.
3) **We have verified the consistency of the atomic insight extraction process** by showing that extraction noise is very low  and that the extractor captures almost all insights from a report while producing stable results across repeated runs which supports the reliability of the insight recall scoring.
4) **We have improved the clarity and structure of the paper** by adding missing related work, unifying terminology, correcting figures and formatting and adding in-depth analysis that explains model behavior and clarifies why certain patterns lead to failure cases and success cases including why stronger models benefit more from complex planning.

Thank you for your time.

---

### Meta-Review · Area_Chair_srt3 · 2026-01-09

**Summary:**

Reviewer qNkw and Reviewer 9AoK view DRBench as filling an evaluation gap by combining public web retrieval with private enterprise heterogeneous data across apps. It's also shipped with well-motivated multi-axis metrics and a containerized environment.
Reviewer hv9E similarly highlights the realism and difficulty of the setting and the systematic benchmark construction pipeline.

Reviewer jNFK’s main concerns are the originally small benchmark size, potential noise from atomic insight extraction, and reliance on LLM-as-a-judge for multiple metrics.
In the rebuttal, the authors report expanding from **15** to **100** tasks and add a larger human-vs-LLM agreement study (91.3% agreement, κ reported) and additional checks on insight extraction consistency and judge-backend stability.
These updates address the main evaluation and scale concerns raised across reviews ([jNFK, qNkw, 9AoK, hv9E]).
Remaining concerns are largely about how broadly DRBench covers enterprise scenarios and whether the proposed metrics fully capture end-to-end answer correctness beyond insight recall.

**Reviewer Concerns:**

**Addressed**

- Benchmark size / statistical power: jNFK, qNkw, and hv9E questioned the small size (15 tasks). The rebuttal reports expansion to 100 tasks with consistent trends on the full benchmark.
- LLM-as-a-judge reliability: jNFK and qNkw asked for stronger human validation and inter-rater reliability. The rebuttal adds a larger human study and reports agreement metrics.
- Judge-backend stability: 9AoK asked for stability across different judge backends. The rebuttal adds GPT-4o vs Llama-3.1 judge comparisons with consistent trends.
- Missing related work and presentation issu: hv9E and 9AoK flagged missing citations and clarity/figure issues, which the rebuttal states are fixed.
- Why agents fail on public insights: jNFK and hv9E requested deeper analysis; the rebuttal adds qualitative explanation of query formulation failures

**Outstanding**

- Breadth of external validity and enterprise coverage: qNkw still notes that coverage and heterogeneity depth may remain modest even with improvements.
- End-to-end “answer correctness” vs insight-based metrics: hv9E questions whether insight recall alone reflects whether questions are answered correctly in a holistic sense. The rebuttal clarifies their stance but the concern remains partly open.
- Robustness of Insight Recall to partial matches / span-level grounding: qNkw asked about span-level alignment and evidence coverage per insight.  The rebuttal provides paraphrase robustness checks, but span-level grounding remains limited.

**Reviewer Scores:**

Reviewer jNFK: Likely increases, as the benchmark size, evaluation reliability, and analysis concerns are directly addressed.

Reviewer qNkw: Unchanged. Core concerns about external validity and metric coverage are only partially resolved.

Reviewer hv9E: Unchanged. While clarification and additional analysis were added, concerns about end-to-end correctness beyond insight recall remain.

Reviewer 9AoK: Unchanged. Judge-backend stability and presentation issues are addressed, but no shift in overall stance is indicated.

---

### Decision · Program_Chairs · 2026-01-26

Accept (Poster)